# Diversity-Driven Offline Multi-Objective Optimization via Nested Pareto Set Learning

**Yiyi Zhu** [* 1] **Yaolin Wen** [* 1] **Xiang Xia** [* 1] **Xin An** [1] **Hanyi Si** [1] **Xiang Shu** [2] **Yangde Fu** [1] **Liang Dou** [1] **Hong Qian** [1 3]

## Abstract

Multi-objective optimization (MOO) has emerged as a powerful approach to solving complex optimization problems involving multiple objectives. In many practical scenarios, function evaluations are unavailable or prohibitively expensive, necessitating optimization solely based on a fixed offline dataset. In this setting, known as offline MOO, the goal is to find out the Pareto set without access to the true objective functions. This setting suffers from the out-of-distribution (OOD) issue, where the surrogate model is not accurate for unseen designs. Due to the OOD issue, surrogate errors may cause the optimizer to select solutions that do not lie on the true Pareto front and are biased toward its extremes. To address this, this paper proposes Diversity-driven Offline Multi-Objective Optimization (DOMOO), which aims to find out a diverse and high-quality set of solutions. First, DOMOO incorporates an accumulative risk control module that estimates the potential risk of candidate solutions and alleviates the OOD issue between the training data and the generated solutions. In addition, a nested Pareto set learning (PSL) strategy is proposed to jointly learn preference and PSL parameters, then optimize them, enabling adaptation to diverse Pareto front geometries. To further enhance solution quality, we design a diversity-driven selection strategy that extracts a representative and well-distributed set of final solutions. To achieve this diversity-driven selection strategy, we propose $\text{IGD}_{\text{offline}}$, a tailored indicator for the offline setting that considers both diversity and convergence,

and avoids the bias of hypervolume indicator. Extensive experiments on synthetic and real-world benchmarks show that DOMOO achieves the best average rank across tasks in both convergence and diversity among the compared methods.

## 1. Introduction

Multi-objective optimization (MOO) is widely used in fields ranging from neural architecture search (Lu et al., 2020) to antenna structure design (Yu et al., 2019), where practitioners must balance conflicting goals, for example, developing a drug (Nicolaou & Brown, 2013) that is both highly effective and minimally toxic. MOO seeks to discover the complete set of Pareto optimal solutions, where no objective can be improved without degrading others (Lin et al., 2022). Many existing methods rely on surrogate models to approximate the true objectives. However, to maintain the accuracy of the surrogates, they typically require actively querying new function evaluations with the true objectives during training (Li et al., 2025). In many real-world applications, such as protein engineering and molecular design (Xue et al., 2024), evaluating true objective functions can be prohibitively expensive or hazardous (Yuan et al., 2025), making function evaluations difficult. Fortunately, these domains often provide available historical data (i.e., offline dataset) in the form of solutions and their corresponding true objective function values. This motivates the offline MOO setting, where the goal is to recommend a set of solutions that represent the best trade-offs among multiple objectives, using only an offline dataset without any active evaluation.

A common approach to solving offline MOO is to train surrogate models (e.g., Gaussian processes or deep neural networks) on the offline dataset. Then, optimization algorithms (e.g., evolutionary algorithms) explore the solution space under the guidance of surrogate models to identify solutions expected to perform well (Xue et al., 2024; Yuan et al., 2025). However, the trained surrogates are susceptible to the out-of-distribution (OOD) issue, often producing unreliable predictions for solutions that lie far from the train-

---

[*]Equal contribution [1]Shanghai Institute of AI for Education, and School of Computer Science and Technology, East China Normal University, Shanghai, China [2]Ant Group, Hangzhou, China [3]Shanghai Innovation Institute, Shanghai, China. Correspondence to: Hong Qian <hqian@cs.ecnu.edu.cn>.

*Proceedings of the 43rd International Conference on Machine Learning*, Seoul, South Korea. PMLR 306, 2026. Copyright 2026 by the author(s).

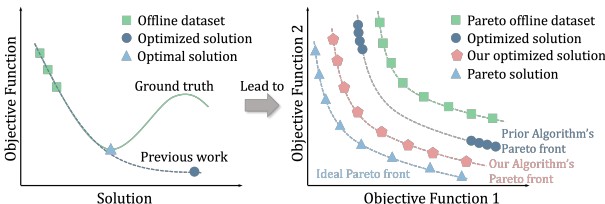

*Figure 1.* **Motivation illustration.** The left figure illustrates the OOD issue in offline single-objective optimization, while the right figure highlights that OOD can lead to reduced diversity and convergence in offline multi-objective optimization.

ing distribution (Lu et al., 2023; Brookes et al., 2019; Chen et al., 2023; Yun et al., 2024). As shown in the left part of Figure 1, we present an offline single-objective optimization example for visualization. In this setting, the surrogate model trained on an offline dataset tends to underestimate the true objective far from the dataset. As a result, the optimizer selects solutions that appear promising under the surrogate but perform poorly under the true objective due to the OOD issue. In the multi-objective setting, the OOD issue can cause the surrogates to underestimate a few solutions, making them incorrectly dominate many others. This leads to *a severely imbalanced Pareto front (as illustrated by the dark blue solutions in the right part of Figure 1), where most solutions are eliminated and the diversity, as well as convergence, drops sharply* (Xue et al., 2024).

Despite its significance, the OOD issue in offline MOO remains largely underexplored. Although several methods have been proposed to address OOD in single-objective offline settings (Qi et al., 2022; Kumar & Levine, 2020; Trabucco et al., 2021), for instance, by incorporating conservatism into surrogate models to intentionally lower the predictions of potentially overestimated OOD solutions (Yu et al., 2021), these techniques cannot be directly applied to MOO due to the intricate structure of Pareto dominance among multiple objectives. As a result, they often exhibit poorer diversity in their solutions when naively extended to the multi-objective case.

Moreover, existing online MOO methods, such as multi-objective Bayesian optimization (Ozaki et al., 2024) and evolutionary algorithms (Li et al., 2015), are typically immune to the OOD issue in their native setting, as they can actively query new data. However, when these methods are directly applied to the offline scenario, where no additional data can be obtained, they often suffer from severe OOD-induced errors, leading to degraded optimization performance. This highlights the urgent need for principled methods that explicitly address OOD issue in offline MOO.

**Contribution.** To address the aforementioned problem in offline MOO, this paper proposes Diversity-Driven Offline Multi-Objective Optimization (DOMOO), a Nested Pareto Set Learning (NPSL) framework designed to improve the diversity and convergence of the candidate solutions. Our

key contributions are:

- **Nested PSL with Risk Control.** We propose a NPSL framework that simultaneously learns preference-conditioned mappings and optimizes preference vectors with accumulative risk control. To tackle OOD uncertainty, DOMOO embeds risk suppression within the preference update mechanism, ensuring a robust balance between diversity and reliability.

- **Diversity-Driven Selection With IGD$_{\text{offline}}$.** We design a diversity-driven solution selection strategy with a novel IGD$_{\text{offline}}$ indicator tailored for the offline setting. This indicator replaces the unavailable true Pareto front with a shifted offline reference, avoiding the bias of hypervolume toward extreme solutions and enabling reliable diversity evaluation without active queries.

- **Strong Empirical Performance.** Extensive experiments on synthetic and real-world benchmarks show that DOMOO achieves the best average rank across tasks in both convergence and diversity among the compared methods.

The subsequent sections present the related work and preliminaries, describe the proposed DOMOO method, show the empirical results, and conclude the paper.

## 2. Related Work

Offline single-objective optimization methods alleviating the OOD issue fall into three types: forward approaches (e.g., COMs (Trabucco et al., 2021), NEMO (Fu & Levine, 2021), COOREM (Zhu et al., 2025)), generative models (e.g., MIN (Kumar & Levine, 2020), CbAS (Brookes et al., 2019)), and trajectory-based methods (e.g., BONET (Mashkaria et al., 2023), PGS (Chemingui et al., 2024)). These methods respectively focus on surrogate robustness, distribution learning with regularization, and leveraging synthetic trajectories to explore quality solutions beyond the offline dataset. While these methods mitigate the OOD issue, extending them to the multi-objective setting is challenging as it requires balancing diversity and convergence across conflicting objectives. Benchmarking efforts such as Design-Bench (Trabucco et al., 2022) and SOO-Bench (Qian et al., 2025) have provided standardized evaluation protocols for offline single-objective optimization; however, no comparable benchmark framework existed for the multi-objective case until Off-MOO-Bench (Xue et al., 2024).

**Offline Multi-Objective Optimization.** Offline MOO typically adopts three main approaches: evolutionary algorithms, Bayesian optimization, and deep neural network-based methods. Population-based search strategies are com-

monly used in evolutionary algorithms, where a trained surrogate model serves as an oracle to guide the optimization process. Representative methods following this paradigm include DDMOEA/GAN (Zhang et al., 2024), MS-RV (Yang et al., 2020), and IBEA-MS (Liu et al., 2023). Similarly, Bayesian optimization also employs a surrogate model as an oracle, but selects candidate solutions via acquisition functions and updates the selection iteratively. Various methods and enhancements have been proposed under the multi-objective Bayesian optimization (MOBO) framework, including MOBO-qNEHVI (Daulton et al., 2021), MOBO-qParEGO (Knowles, 2006), MOBO-JES (Hvarfner et al., 2022), and so on. Unlike the previous two categories, which struggle to effectively address the OOD issue, neural network-based methods can mitigate this problem by replacing traditional surrogate models with those adopted in forward approaches from offline single-objective optimization (e.g., COMs (Trabucco et al., 2021), IOMs (Qi et al., 2022), Tri-Mentoring (Chen et al., 2023)), and extending them using multiple models (Xue et al., 2024) to handle offline MOO. While these methods achieve strong convergence properties, they do not consider how to maintain solution diversity across the Pareto front (PF).

**Pareto Set Learning.** PSL is a recently proposed model-based approach that learns a mapping from preference vectors to Pareto optimal solutions by training a neural network. PSL-MOBO (Lin et al., 2022), which is the first method to integrate PSL with MOBO, enables efficient approximation of black-box PFs by learning a preference-conditioned solution generator based on surrogate models. EPS (Ye et al., 2024) combines evolutionary algorithms with PSL, enabling faster convergence and broader PF coverage through adaptive evolution of preference vectors. CDM-PSL (Li et al., 2025) introduces diffusion models into Pareto set learning for MOBO, achieving improved solution quality and diversity under limited evaluations through conditional sampling and entropy-based guidance. However, PSL-MOBO heavily relies on Gaussian process surrogates, which were primarily developed for online evaluation. When applied to offline optimization, they often encounter severe OOD issues.

## 3. Preliminaries

### 3.1. Offline Multi-Objective Optimization

In offline MOO, the goal is to optimize multiple conflicting objectives simultaneously using only a fixed, static dataset $\mathcal{D} = \{(\boldsymbol{x}_i, \boldsymbol{y}_i)\}_{i=1}^N$, where $\boldsymbol{x}_i \in \mathcal{X} \subset \mathbb{R}^D$ denotes a candidate solution and $\boldsymbol{y}_i$ is the associated objective vector. The MOO problem can be formally stated as $\min_{\boldsymbol{x} \in \mathcal{X}} \boldsymbol{f}(\boldsymbol{x}) = (f_1(\boldsymbol{x}), f_2(\boldsymbol{x}), \ldots, f_M(\boldsymbol{x}))$, where $\boldsymbol{f} : \mathcal{X} \to \mathbb{R}^M$ is composed of $M$ individual objective functions.

**Definition 3.1** (**Pareto-Optimal Solution** (Ehrgott, 2005)).

A solution $\boldsymbol{x}^* \in \mathcal{X}$ is called Pareto-optimal if there exists no other solution $\boldsymbol{x}' \in \mathcal{X}$ such that $\forall i \in \{1, 2, \ldots, M\}$, $f_i(\boldsymbol{x}') \leq f_i(\boldsymbol{x}^*)$, with at least one strict inequality, i.e., $\exists j \in \{1, 2, \ldots, M\}$ such that $f_j(\boldsymbol{x}') < f_j(\boldsymbol{x}^*)$.

**Definition 3.2** (**Pareto Set and Pareto Front** (Li et al., 2015)). The set of all Pareto-optimal solutions is called Pareto set, denoted by $\mathcal{M}_{\text{ps}}$, and its image under the mapping $\boldsymbol{f}$, $\boldsymbol{f}(\mathcal{M}_{\text{ps}}) = \{\boldsymbol{f}(\boldsymbol{x}) \mid \boldsymbol{x} \in \mathcal{M}_{\text{ps}}\}$ is called the Pareto front.

However, in MOO no single solution can optimize all objectives concurrently and trade-offs among conflicting objectives are inevitable (Qian et al., 2013; Bian et al., 2025). Therefore, the primary goal in offline MOO can be viewed as the pursuit of the Pareto solutions (i.e., solutions for which no other solution can improve some objectives without causing detriment to at least one other objective, as defined in Definition 3.1) and the effective approximation of the Pareto front (as Definition 3.2).

### 3.2. Pareto Set Learning for Offline MOO

In MOO, the preference $\boldsymbol{\lambda}$ reflects the relative importance or priority of each objective. To learn a connection from all valid preferences $\Lambda = \{\boldsymbol{\lambda} \in \mathbb{R}_+^M \mid \sum \lambda_i = 1\}$ to their corresponding Pareto solutions, Pareto set learning (PSL) (Lin et al., 2022) trains a Pareto set model through scalarization methods, which bridge preferences and Pareto solutions by transforming the multi-objective problem into a single-objective one for each preference. Specifically, PSL (Lin et al., 2022) uses the scalarization based on the augmented Tchebycheff approach (Kaliszewski, 1987):

$$g_{\text{tch\_aug}}(\boldsymbol{x} \mid \boldsymbol{\lambda}) = \max_{1 \leq i \leq M} \{\lambda_i (f_i(\boldsymbol{x}) - (z_i^* - \varepsilon))\} + \rho \sum_{i=1}^M \lambda_i f_i(\boldsymbol{x}), \quad \forall \boldsymbol{\lambda} \in \Lambda, \tag{1}$$

where $\boldsymbol{z}^* = (z_1^*, \cdots, z_M^*)$ is the ideal vector for the objective $\boldsymbol{f}(\boldsymbol{x})$, defined as $z_i^* = \min_{\boldsymbol{x} \in \mathcal{D}} f_i(\boldsymbol{x})$ for each $i = 1, \ldots, M$, $\varepsilon$ is a small positive scalar and $\rho$ is a small positive scalar that depends on the problem and the current solution location.

During the training process, for each sampled preference $\boldsymbol{\lambda}$, the Pareto set model outputs a solution $h_\phi(\boldsymbol{\lambda})$ and is optimized to minimize the scalarized objective $g_{\text{tch\_aug}}(h_\phi(\boldsymbol{\lambda})|\boldsymbol{\lambda})$ over all valid preferences: $\phi^* = \arg\min_\phi \mathbb{E}_{\boldsymbol{\lambda} \sim \Lambda} g_{\text{tch\_aug}}(\boldsymbol{x} = h_\phi(\boldsymbol{\lambda})|\boldsymbol{\lambda})$. However, in offline MOO, solutions cannot be evaluated during the optimization process. Therefore, $M$ surrogate models $\hat{f}_i$ are built for each objective based on the offline dataset $\mathcal{D}$ to predict solutions when calculating Equation 1. With the trained Pareto set model $h_{\phi^*}$, we can obtain the Pareto set: $\mathcal{M}_{\text{ps}} = \{\boldsymbol{x} = h_{\phi^*}(\boldsymbol{\lambda}) \mid \boldsymbol{\lambda} \in \Lambda\}$, where $h_{\phi^*}(\boldsymbol{\lambda}) = \arg\min_{\boldsymbol{x} \in \mathcal{X}} g_{\text{tch\_aug}}(\boldsymbol{x} \mid \boldsymbol{\lambda}), \forall \boldsymbol{\lambda} \in \Lambda$.

### 3.3. Energy Model

In offline MOO, the objective function cannot be evaluated during the optimization process, so $M$ surrogate models are constructed for each objective given the offline dataset $\mathcal{D}$ to predict the objective values for any candidate solution. However, most existing surrogate models typically ignore OOD risk, which can lead to performance degradation or unsafe decisions in high-stakes applications. Therefore, explicit risk modeling and suppression are necessary in offline multi-objective optimization. To mitigate the negative impact of OOD solutions, ARCOO (Lu et al., 2023) introduces the energy model $E_{\boldsymbol{\omega}}$ to assign an energy value $E_{\boldsymbol{\omega}}(\boldsymbol{x})$ to each solution $\boldsymbol{x}$, which is realized as a neural network that maps solutions $\boldsymbol{x} \in \mathbb{R}^D$ to their associated energy $E_{\boldsymbol{\omega}}(\boldsymbol{x}) \in \mathbb{R}$. The energy model is trained via contrastive divergence with Langevin dynamics negative sampling, and a risk suppression factor $R(\boldsymbol{x})$ is computed to dynamically weight optimization updates, suppressing OOD solutions while emphasizing in-distribution (ID) ones. The detailed formulation and training process for the energy model $E_{\boldsymbol{\omega}}(\boldsymbol{x})$ and the risk suppression factor $R(\boldsymbol{x})$ are deferred to Appendix A.

## 4. The Proposed Method

In this section, we first provide an overview of the proposed method diversity-driven offline multi-objective optimization (DOMOO), followed by a detailed description of the nested Pareto set learning with accumulative risk control, and diversity-driven solution selection strategy, respectively.

### 4.1. Methodology Overview

Offline MOO struggles to alleviate the OOD issue, which results in a severely imbalanced PF (i.e., solutions cluster in high-density regions, failing to cover the entire PF), damaging both the diversity and convergence of the solutions. To alleviate this issue, we propose DOMOO, a risk-aware offline MOO method via nested Pareto set learning. We provide the framework of our algorithm in Figure 2 and the corresponding pseudo-code in Appendix B. Specifically, we first train $M$ surrogate models for each objective. Based on these surrogate models, we perform nested Pareto set learning with accumulative risk control to obtain a Pareto set model. Finally, candidate solutions are generated by both the trained Pareto set model and the trained surrogate model, and then the proposed diversity-driven solution selection strategy is employed, resulting in a solution set with balanced diversity and convergence.

### 4.2. Nested Pareto Set Learning with Risk Control

As described in Section 3.2, PSL (Lin et al., 2022) trains a Pareto set model to map any valid preference $\boldsymbol{\lambda} \in \Lambda = \{\boldsymbol{\lambda} \in \mathbb{R}_+^M \mid \sum \lambda_i = 1\}$ to its corresponding Pareto solution

via scalarization. However, in offline settings, the OOD issue can mislead the Pareto set model by promoting solutions with unreliably estimated high performance, creating an unexpected diversity on the PF. To mitigate the OOD issue, we propose a nested Pareto set learning approach with risk control. This approach addresses the OOD-induced diversity loss by jointly optimizing the Pareto set model parameters and preferences in a nested manner, where the inner loop preference optimization explores underrepresented regions of the PF and incorporates risk control, while the outer loop model optimization improves solution quality under these risk-guided preferences.

While ARCOO (Lu et al., 2023) provides a principled risk control framework for single-objective offline optimization, extending it to the multi-objective setting is non-trivial for two fundamental reasons. First, **preference-risk coupling**: in Pareto set learning, the optimization landscape is explicitly conditioned on preference vectors, and OOD risk does not act uniformly over the solution space; it varies across objectives and interacts with preference gradients. Simply bounding the scalarized error without modeling how risk shifts preference dynamics is insufficient to prevent Pareto front distortion. Second, **multi-objective dominance structure**: surrogate overestimation in one objective can artificially dominate others under the Pareto dominance criterion, collapsing solution diversity. Our extension shows that, by incorporating the risk suppression factor $R(\boldsymbol{x})$ into the preference gradient update (Eq. (2)), DOMOO effectively damps optimization steps toward unreliable OOD regions while preserving gradient flow toward well-supported trade-offs.

**Surrogate Model Training.** In offline MOO, the true objective functions are inaccessible during optimization. Therefore, before the nested Pareto set learning begins, we construct $M$ surrogate models $\hat{f}_1, \ldots, \hat{f}_M$ from the offline dataset $\mathcal{D}$, one for each objective. The complete surrogate model is then given by $\hat{\boldsymbol{f}}(\boldsymbol{x}) = (\hat{f}_1(\boldsymbol{x}; \boldsymbol{\theta}_1^*), \ldots, \hat{f}_M(\boldsymbol{x}; \boldsymbol{\theta}_M^*))$.

**Modeling and Suppressing Accumulative Risk.** In offline optimization, the risk of OOD is non-negligible, and neglecting this risk may result in performance degradation (Lu et al., 2023). Therefore, explicit risk modeling and suppression are necessary in offline MOO to mitigate OOD risk. Specifically, as shown in Figure 2(a), an energy model $E_{\boldsymbol{\omega}}$ is trained following ARCOO (Lu et al., 2023) to measure the risk of solutions and then a risk suppression factor is computed as $R(\boldsymbol{x}) = c(E_{\tilde{Q}} - E_{\boldsymbol{\omega}}(\boldsymbol{x}))/(E_{\tilde{Q}} - E_{\tilde{P}})$, where $E_{\tilde{Q}} = \mathbb{E}_{\boldsymbol{x}' \sim \tilde{Q}}[E_{\boldsymbol{\omega}}(\boldsymbol{x}')]$, $E_{\tilde{P}} = \mathbb{E}_{\boldsymbol{x}' \sim \tilde{P}}[E_{\boldsymbol{\omega}}(\boldsymbol{x}')]$ and $c$ denotes the initial momentum (consistent with ARCOO). The $\tilde{P}$ is the empirical distribution over the high-quality batch of solutions in the offline dataset. The $\tilde{Q}$ is the high-risk distribution sampled by Langevin dynamics starting

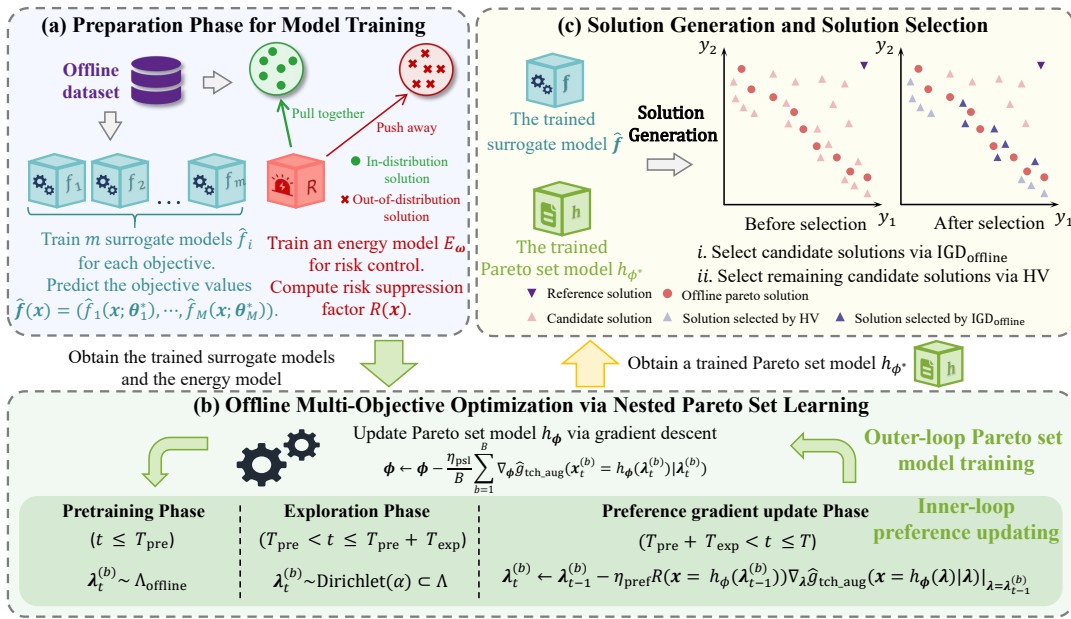

*Figure 2.* The framework of diversity-driven offline multi-objective optimization via nested Pareto set learning. **(a)** Surrogate models are trained for each objective and energy model is trained for risk control. **(b)** A nested Pareto set learning process with risk control is conducted to obtain a Pareto set model. **(c)** Candidate solutions are generated and then sequentially selected using the IGD$_{\text{offline}}$ indicator to ensure diversity, followed by the HV indicator to guarantee convergence.

from $\widetilde{P}$. For more details about the energy model $E_{\omega}$, please refer to the Section 3.3.

**Nested Pareto Set Learning.** The nested Pareto set learning process consists of three phases: pretraining, exploration, and preference gradient update. In each iteration, the preferences are updated first (inner loop), and then the Pareto set model $h_{\phi}$ is trained to convergence using these updated preferences as targets (outer loop).

In the pretraining phase, we leverage the offline PF $(\boldsymbol{X}_{\text{off}}, \boldsymbol{Y}_{\text{off}})$ to provide a better initialization for the subsequent training process. Specifically, during pretraining, we sample preferences from the offline preferences $\Lambda_{\text{offline}} = \left\{ \boldsymbol{\lambda}_{\text{off}}^{(i)} = \boldsymbol{\lambda}_{\text{off}}^{(i)'} / \left\| \boldsymbol{\lambda}_{\text{off}}^{(i)'} \right\|_1 \right\}_{i=1}^{n}$, where $n$ is the number of solutions in the offline PF and $\boldsymbol{\lambda}_{\text{off}}^{(i)'} = (1/(y_{\text{off},1}^{(i)} - z_1^*), \cdots, 1/(y_{\text{off},M}^{(i)} - z_M^*))$. Here, $\boldsymbol{z}^* = (z_1^*, \cdots, z_M^*)$ is the ideal vector for the objective $\boldsymbol{f}(\boldsymbol{x})$ and $\boldsymbol{y}_{\text{off}}^{(i)}$ is the objective vector of the $i$-th solution in the offline PF. By sampling preferences in this way, the pretraining process leverages the structure of the offline PF, providing a better initialization for the subsequent training stages and enabling the Pareto set model to start closer to the optimal solution distribution.

Then, in the exploration phase, the preferences are sampled from the valid preference $\Lambda_t = \{\boldsymbol{\lambda}_t^{(b)} \sim \text{Dirichlet}(\alpha) \subset \Lambda\}_{b=1}^{B}$, where $B$ is the batch size of the solutions in each iteration. Dirichlet$(\alpha)$ with $\alpha = \mathbf{1}_M$, where $\mathbf{1}_M$ denotes the $M$-dimensional all-ones vector, is defined on the sim-

plex $\{\boldsymbol{\lambda} \in \mathbb{R}_+^M \mid \sum \lambda_i = 1\}$, enabling diverse trade-off sampling and preventing overfitting to a narrow set of preferences (Navon et al., 2021). This stage serves as a pure exploration phase, enabling the model to be trained over the entire preference space and thus improving its generalization across different preferences.

Finally, in preference gradient update phase, preferences are adaptively updated using gradient information. To mitigate OOD risk, we incorporate the explicit risk modeling and suppression into the preference update. The preference gradient update phase with accumulative risk control is defined as follows:

$$\boldsymbol{\lambda}_t^{(b)} = \boldsymbol{\lambda}_{t-1}^{(b)} - \eta_{\text{pref}} R\Big(\boldsymbol{x} = h_{\phi}\Big(\boldsymbol{\lambda}_{t-1}^{(b)}\Big)\Big)$$
$$\cdot \nabla_{\boldsymbol{\lambda}} \hat{g}_{\text{tch\_aug}}(\boldsymbol{x} = h_{\phi}(\boldsymbol{\lambda}) \mid \boldsymbol{\lambda})\Big|_{\boldsymbol{\lambda}_{t-1}^{(b)}}, b = 1, 2, \ldots, B,$$

(2)

where $\eta_{\text{pref}}$ is the learning rate for preference optimization, $R(\boldsymbol{x})$ is a risk suppression factor (Lu et al., 2023) that controls the OOD risk of solution $\boldsymbol{x}$ and $\hat{g}_{\text{tch\_aug}}(\cdot)$ is the augmented Tchebycheff scalarization with the trained surrogate models. Specifically, the augmented Tchebycheff scalarization is defined as: $\hat{g}_{\text{tch\_aug}}(\boldsymbol{x} \mid \boldsymbol{\lambda}) = \max_{1 \leq i \leq M} \{\lambda_i(\hat{f}_i(\boldsymbol{x}; \boldsymbol{\theta}_i^*) - (z_i^* - \varepsilon))\} + \rho \sum_{i=1}^{M} \lambda_i \hat{f}_i(\boldsymbol{x}; \boldsymbol{\theta}_i^*)$, in which $\hat{f}_i(\cdot; \boldsymbol{\theta}_i^*)$ denotes the trained surrogate model for the $i$-th objective. Although Eq. (2) minimizes $\hat{g}_{\text{tch\_aug}}$, its gradient is evaluated on $\boldsymbol{x} = h_{\phi}(\boldsymbol{\lambda})$ from the current model. Hence, preferences leading to poor solutions produce larger gradients and are updated more,

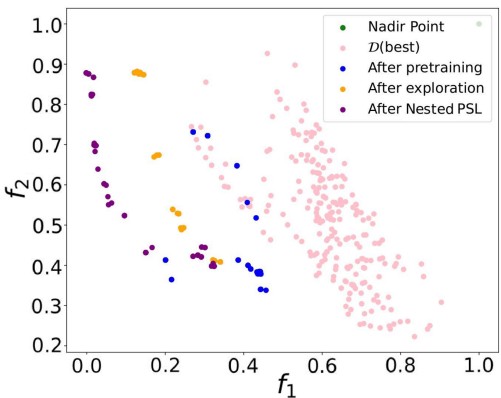

*Figure 3.* **Visualization of the nested PSL.** The solutions generated by DOMOO after each preference update phases in IN-1K/MOP7 task are visualized.

implicitly shifting exploration toward underrepresented regions. The outer loop (Eq. (3)) then improves solution quality under these updated preferences, jointly balancing diversity and quality.

As shown in Figure 3, it can be found that after the exploration and preference gradient update phases, the solutions generated by the Pareto set model become more uniformly distributed, better ensuring the diversity of the solution set.

After updating the preferences, gradient descent is used to train the Pareto set model $h_\phi$ with the trained surrogate model $\hat{\boldsymbol{f}}(\cdot)$, where $\eta_{\text{psl}}$ is the learning rate for the PSL model:

$$\phi = \phi - \frac{\eta_{\text{psl}}}{B} \sum_{b=1}^{B} \nabla_\phi \hat{g}_{\text{tch\_aug}} \left( \boldsymbol{x}_t^{(b)} = h_\phi \left( \boldsymbol{\lambda}_t^{(b)} \right) \mid \boldsymbol{\lambda}_t^{(b)} \right) .$$
(3)

Through the nested Pareto set learning approach, we obtain the trained Pareto set model $h_{\phi^*}$, which can effectively adapt to diverse PF geometries and approximate the Pareto set powerfully. Notably, the candidate solutions are guided by risk-modulated preferences from Eq. (2) throughout training, so the final selection stage can focus on the diversity–convergence trade-off.

### 4.3. Diversity-Driven Solution Selection Strategy

After the nested Pareto set learning, we have obtained a practical Pareto set model $h_{\phi^*}$ that can easily approximate the Pareto set with the valid preferences $\Lambda$. However, in real offline MOO scenarios, the deployment of solution sets is often constrained by scale limitations, e.g., only limited solutions can be evaluated. Therefore, how to select the optimal subset from the learned Pareto solution set becomes a key challenge. In this paper, we propose a diversity-driven solution selection strategy by combining two indicators: offline inverse generation distance (IGD$_{\text{offline}}$) and hypervolume (HV), to better balance diversity and convergence.

The traditional inverse generation distance (IGD) assumes access to the true PF to evaluate how well a solution set covers it.

In offline MOO, however, the true front is not observable since no additional evaluations are permitted. Therefore, we adapt IGD to the offline regime by replacing the unknown true front with an offline PF estimated from the dataset and by introducing a shift to form a stricter reference. The full definition is given by

$$\text{IGD}_{\text{offline}} = \frac{1}{n} \sum_{i=1}^{n} \min_{1 \leq j \leq |\boldsymbol{X}_{\text{cand}}|} \left\| \boldsymbol{y}_{\text{off}}^{(i)} - \beta y' \mathbf{1}_M - \hat{\boldsymbol{y}}_{\text{cand}}^{(j)} \right\|_2 ,$$
(4)

where $n$ is the number of solutions in the offline PF, $\boldsymbol{y}_{\text{off}}^{(i)}$ is the objective vector of the $i$-th solution in the offline PF, $|\boldsymbol{X}_{\text{cand}}|$ denotes the number of candidate solutions in $\boldsymbol{X}_{\text{cand}}$ and $\hat{\boldsymbol{y}}_{\text{cand}}^{(j)}$ is the objective vector of the $j$-th solution in the candidate solutions $\boldsymbol{X}_{\text{cand}}$ predicted by the surrogate model. Here, $\beta$ is a scaling factor and $y'$ is a shift value, defined as $y' = \max_{1 \leq i \leq n} \min_{1 \leq m \leq M} y_{\text{off},m}^{(i)}$, where $y_{\text{off},m}^{(i)}$ denotes the $m$-th objective value of the $i$-th solution in the offline PF. The shift value $y'$ is introduced to construct a more challenging reference front, allowing a stricter evaluation of optimization performance in terms of convergence and diversity. All objective values are min-max normalized per objective based on the offline dataset, ensuring fair scale-invariant comparisons. It is worth noting that the construction of IGD$_{\text{offline}}$ does not favor solutions that stay close to the offline data, as the reference front is normalized and shifted toward the ideal point, encouraging exploration and broad Pareto-front coverage rather than conservative interpolation.

Before performing the solution selection strategy, the trained Pareto set model $h_{\phi^*}$ is employed to generate $K$ candidate solutions $\boldsymbol{X}_{\text{ps}} = \{\boldsymbol{x}_{\text{ps}}^{(k)} = h_{\phi^*}(\boldsymbol{\lambda}_{\text{ps}}^{(k)})\}_{k=1}^{K}$, where $\boldsymbol{\lambda}_{\text{ps}}^{(k)} \sim \text{Dirichlet}(\alpha) \subset \Lambda$. To further enhance the diversity of the candidate solution set, we combine the $K$ solutions generated by our trained Pareto set model $h_\phi^*$ with another $K$ solutions produced by the surrogate model $\hat{\boldsymbol{f}}$, thereby obtaining the complete candidate solutions $\boldsymbol{X}_{\text{cand}}$.

**Diversity-Driven Solution Selection.** To address the diversity challenge posed by the HV indicator in offline settings, which is demonstrated in Appendix I, we select solutions based on both IGD$_{\text{offline}}$ and HV indicators. The core issue is that, due to limited offline data, surrogates often extrapolate a spurious Pareto front wider than the true one in OOD regions. HV's marginal-volume mechanism then uniformly picks solutions along this illusory front, thereby selecting many tightly clustered, low-quality solutions despite acceptable surrogate scores. In contrast, IGD$_{\text{offline}}$ penalizes deviation from the offline Pareto front, acting as a conservative filter against OOD artifacts.

*Table 1.* Comparison of average HV ranks achieved by different offline MOO methods across different tasks in Off-MOO-Bench (Xue et al., 2024). For each task, the top three methods are highlighted using **(1st)**, (2nd), and (3rd) formatting. $\mathcal{D}$(best) denotes the best subset in the offline dataset (i.e., with the highest HV), and the last column reports the average rank across all tasks. "N/A" marks cases where a method cannot complete on a task, either because it fails to return feasible solutions or because it exceeds practical limits in runtime or GPU memory on that task.

| Methods | Synthetic | MO-NAS | MORL | Sci-Design | RE | Average Rank |
|---|---|---|---|---|---|---|
| $\mathcal{D}$(best) | $11.79 \pm 0.71$ | $13.37 \pm 0.37$ | $6.80 \pm 0.24$ | $11.55 \pm 0.61$ | $14.46 \pm 0.44$ | $12.73 \pm 0.48$ |
| End-to-End | $7.46 \pm 0.78$ | $6.09 \pm 0.40$ | $4.10 \pm 0.37$ | $8.25 \pm 1.30$ | $6.77 \pm 0.78$ | $6.81 \pm 0.54$ |
| End-to-End + GradNorm | $9.96 \pm 0.72$ | $11.71 \pm 0.71$ | $12.30 \pm 0.51$ | $9.88 \pm 1.20$ | $11.55 \pm 0.44$ | $10.97 \pm 0.39$ |
| End-to-End + PcGrad | $\underline{6.85 \pm 0.61}$ | $7.23 \pm 1.11$ | $10.70 \pm 0.51$ | $7.17 \pm 1.05$ | $8.22 \pm 1.03$ | $7.50 \pm 0.48$ |
| Multi Head | $6.88 \pm 1.07$ | $\underline{6.03 \pm 0.50}$ | $10.20 \pm 0.60$ | $9.55 \pm 1.25$ | $6.28 \pm 0.81$ | $6.83 \pm 0.62$ |
| Multi Head + GradNorm | $10.90 \pm 1.01$ | $13.19 \pm 1.25$ | $12.20 \pm 0.68$ | $10.12 \pm 0.86$ | $12.22 \pm 1.17$ | $11.89 \pm 0.97$ |
| Multi Head + PcGrad | $7.94 \pm 0.98$ | $6.76 \pm 0.63$ | $8.70 \pm 0.51$ | $7.20 \pm 1.14$ | $9.49 \pm 1.04$ | $7.98 \pm 0.61$ |
| Multiple Models | $\underline{6.24 \pm 0.58}$ | $6.63 \pm 0.88$ | $7.40 \pm 0.49$ | $9.18 \pm 1.58$ | $6.37 \pm 0.70$ | $\underline{6.67 \pm 0.37}$ |
| Multiple Models + COMs | $9.40 \pm 0.44$ | $6.63 \pm 0.59$ | $\mathbf{1.90 \pm 0.37}$ | $\mathbf{5.12 \pm 0.91}$ | $10.72 \pm 0.38$ | $8.30 \pm 0.22$ |
| Multiple Models + RoMA | $9.90 \pm 1.02$ | $6.91 \pm 0.22$ | $6.10 \pm 0.37$ | $7.30 \pm 1.43$ | $9.45 \pm 0.97$ | $8.56 \pm 0.52$ |
| Multiple Models + IOM | $7.36 \pm 0.95$ | $\underline{5.96 \pm 1.09}$ | $\underline{3.50 \pm 0.45}$ | $\underline{5.72 \pm 0.36}$ | $\underline{6.38 \pm 1.30}$ | $\underline{6.41 \pm 0.68}$ |
| Multiple Models + ICT | $9.38 \pm 0.77$ | $9.53 \pm 0.71$ | $9.10 \pm 3.12$ | $6.60 \pm 1.25$ | $6.80 \pm 1.10$ | $8.50 \pm 0.60$ |
| Multiple Models + Tri-Mentoring | $9.44 \pm 0.67$ | $10.49 \pm 0.55$ | $8.50 \pm 2.07$ | $9.93 \pm 0.76$ | $6.40 \pm 0.45$ | $8.93 \pm 0.20$ |
| MOBO | $10.23 \pm 1.03$ | $\mathbf{5.02 \pm 0.12}$ | N/A | $\underline{5.93 \pm 2.10}$ | $8.91 \pm 0.82$ | $7.62 \pm 0.50$ |
| MOBO-$q$ParEGO | $10.50 \pm 0.97$ | $12.80 \pm 0.85$ | N/A | $12.10 \pm 1.59$ | $8.76 \pm 0.31$ | $10.84 \pm 0.19$ |
| MOBO-JES | $15.81 \pm 0.47$ | N/A | N/A | N/A | $12.02 \pm 1.06$ | $13.91 \pm 0.55$ |
| ParetoFlow | $9.18 \pm 1.55$ | $11.31 \pm 0.65$ | $9.83 \pm 1.31$ | $13.58 \pm 2.95$ | $9.04 \pm 0.66$ | $10.19 \pm 0.98$ |
| DOMOO (**ours**) | $\mathbf{3.89 \pm 0.56}$ | $6.65 \pm 0.17$ | $\underline{3.60 \pm 0.86}$ | $6.83 \pm 1.28$ | $\mathbf{3.26 \pm 0.53}$ | $\mathbf{4.63 \pm 0.38}$ |

Notably, IGD$_{\text{offline}}$ and HV are complementary indicators: IGD$_{\text{offline}}$ emphasizes diversity and the uniform coverage of the PF, whereas HV focuses more on solution quality. Therefore, combining IGD$_{\text{offline}}$ with HV allows us to better balance diversity and convergence while mitigating the limitations of using HV alone.

Therefore, we first utilize the IGD$_{\text{offline}}$ indicator to greedily select up to 128 solutions from the candidates $X_{\text{cand}}$. The budget of 128 solutions for the IGD$_{\text{offline}}$ stage is not chosen heuristically but selected through hyperparameter analysis: varying this budget from 0 to 256 (see Appendix K) shows stable performance with a clear peak around 128, confirming that this split effectively balances diversity preservation and convergence refinement. This encourages the selection of solutions that cover different regions of the offline PF, thereby enhancing the diversity of the solutions. Subsequently, we select the remaining solutions from the candidates $X_{\text{cand}}$ using the HV indicator, which maximizes the hypervolume in the objective space, serving as a convergence-oriented filling. With the diversity-driven strategy combining IGD$_{\text{offline}}$ for screening and HV for filling, we obtain the final solution set with 256 solutions, which effectively balances between convergence and diversity.

## 5. Experiment

In this section, we conduct a comprehensive empirical evaluation of DOMOO against a series of existing offline MOO approaches across multiple benchmark tasks. We begin by outlining the experimental setup, encompassing tasks, compared methods, training settings, and evaluation metrics. Subsequently, we report the experimental results, perform an ablation study and a hyper-parameter analysis. The exper-

iments are designed to answer the four significant questions:

**Q1:** Can DOMOO achieve better performance than other offline MOO methods in terms of convergence?

**Q2:** Can DOMOO balance diversity and convergence?

**Q3:** How do core modules contribute to diversity and convergence?

**Q4:** How do hyper-parameters affect the diversity of the solution set obtained by DOMOO?

The four questions are answered sequentially in this section. The full implementation and codes are available at https://github.com/YaolinWen/DOMOO.

**Benchmark and Tasks.** We evaluate DOMOO on Off-MOO-Bench (Xue et al., 2024), which includes five categories of offline multi-objective tasks: Synthetic functions, MO-NAS, MORL, Sci-Design, and RE. These tasks span diverse domains, objective dimensionalities, and optimization difficulties, providing a comprehensive testbed for offline MOO. Task details are provided in Appendix C.

**Compared Methods.** Our evaluation primarily adopts the baselines from Off-MOO-Bench (Xue et al., 2024), encompassing both DNN-based and GP-based approaches. To broaden the scope of method categories, we additionally evaluate ParetoFlow (Yuan et al., 2025), a recent flow-based generative method. Detailed descriptions of all baselines are provided in Appendix D.

**Evaluation.** Following Off-MOO-Bench (Xue et al., 2024), we evaluate each method by generating 256 solutions and querying the true objective functions. We report HV (Yuan et al., 2025), which measures the dominated volume with

*Table 2.* Comparison of average IGD$_{offline}$ ranks. Details are the same as Table 1.

| Methods | Synthetic | MO-NAS | MORL | Sci-Design | RE | Average Rank |
|---|---|---|---|---|---|---|
| $\mathcal{D}$(best) | 9.85 ± 0.96 | 12.83 ± 1.87 | 6.80 ± 0.24 | 9.62 ± 2.70 | 6.65 ± 0.73 | 9.71 ± 1.17 |
| End-to-End | 7.80 ± 1.38 | **3.89 ± 0.78** | 4.30 ± 0.60 | 8.68 ± 1.72 | 9.72 ± 1.44 | 7.12 ± 1.02 |
| End-to-End + GradNorm | 10.79 ± 1.23 | 10.93 ± 2.06 | 12.30 ± 0.51 | 9.25 ± 0.68 | 11.29 ± 0.62 | 10.90 ± 1.17 |
| End-to-End + PcGrad | 6.80 ± 1.09 | 6.54 ± 1.37 | 10.20 ± 0.24 | 6.97 ± 1.25 | 9.94 ± 0.93 | 7.71 ± 0.83 |
| Multi Head | 7.71 ± 1.62 | 4.00 ± 0.70 | 7.50 ± 0.45 | 9.25 ± 2.27 | 8.74 ± 0.72 | 7.24 ± 0.96 |
| Multi Head + GradNorm | 10.99 ± 1.45 | 12.68 ± 1.42 | 12.10 ± 0.49 | 8.95 ± 2.09 | 10.42 ± 1.50 | 11.10 ± 1.41 |
| Multi Head + PcGrad | 7.55 ± 1.37 | 7.53 ± 1.18 | 9.40 ± 0.58 | 6.25 ± 1.27 | 9.33 ± 0.82 | 7.96 ± 0.66 |
| Multiple Models | 6.58 ± 0.93 | 4.95 ± 0.44 | 6.10 ± 0.37 | 11.03 ± 1.95 | 9.43 ± 0.97 | 7.20 ± 0.44 |
| Multiple Models + COMs | 9.40 ± 0.56 | 6.90 ± 0.53 | 5.40 ± 0.66 | 6.10 ± 1.07 | 9.72 ± 0.71 | 8.38 ± 0.27 |
| Multiple Models + RoMA | 9.90 ± 0.75 | 7.30 ± 0.53 | **2.60 ± 0.58** | 7.97 ± 3.25 | 8.75 ± 0.92 | 8.40 ± 0.44 |
| Multiple Models + IOM | 7.00 ± 0.77 | 7.68 ± 2.00 | 7.50 ± 0.45 | 6.45 ± 0.87 | 5.90 ± 0.90 | 6.63 ± 0.52 |
| Multiple Models + ICT | 8.99 ± 0.58 | 9.11 ± 0.81 | 7.70 ± 3.19 | 7.40 ± 2.16 | 7.86 ± 0.82 | 8.55 ± 0.57 |
| Multiple Models + Tri-Mentoring | 9.63 ± 0.48 | 9.52 ± 1.04 | 8.90 ± 2.42 | 11.62 ± 2.19 | 8.20 ± 0.63 | 9.32 ± 0.24 |
| MOBO | 8.99 ± 1.40 | 6.58 ± 0.92 | N/A | 4.95 ± 2.17 | 7.68 ± 1.00 | 7.41 ± 0.90 |
| MOBO-$q$ParEGO | 9.20 ± 0.92 | 12.15 ± 0.63 | N/A | 8.30 ± 2.01 | 5.29 ± 0.36 | 9.07 ± 0.50 |
| MOBO-JES | 14.92 ± 0.80 | N/A | N/A | N/A | 10.01 ± 2.49 | 11.68 ± 1.99 |
| ParetoFlow | 8.82 ± 3.19 | 9.60 ± 0.70 | 5.00 ± 2.94 | **3.21 ± 1.67** | **2.92 ± 0.25** | 8.20 ± 3.45 |
| DOMOO (ours) | **4.95 ± 0.70** | 7.14 ± 0.48 | 6.70 ± 1.54 | 7.55 ± 1.21 | 6.67 ± 0.47 | **6.27 ± 0.23** |

respect to a reference point (i.e., Nadir Point in Figure 3), where a higher HV indicates better performance. To address the bias of HV toward extreme solutions in offline settings, we also report IGD$_{offline}$ as in Section 4.3.

### 5.1. Experimental Settings

### 5.2. The Performance of DOMOO

**About Superiority and Convergence (To Q1).** Table 1 reports the average HV rank of all compared offline MOO methods. Detailed results at 100$^{th}$ and 50$^{th}$ percentiles are provided in Appendix G. We make the following observations: **(1)** As shown in Table 1, DOMOO achieves the best average rank across all tasks, verifying its effectiveness and convergence. **(2)** End-to-End, Multi-Head, and Multiple Models consistently outperform $\mathcal{D}$(best), highlighting the effectiveness of learned surrogates and generative models in discovering solutions beyond the offline dataset. **(3)** GP-based methods often tend to exhibit relatively less competitive. This is partly because they are primarily designed for online optimization and may struggle in offline settings. Moreover, their high computational cost and long runtime make them impractical for complex tasks, sometimes leading to failure to produce any solution within the time budget (i.e., N/A in the Table 1). **(4)** Although DOMOO performs worse on a few extremely discrete tasks (e.g., C-10/MOP1, C-10/MOP2, IN-1K/MOP5), this is mainly because these tasks require very high-dimensional one-hot encodings, resulting in extremely sparse inputs that are difficult for neural Pareto-set models to learn. The core challenge lies in the mismatch between continuous optimization and one-hot discrete spaces: valid data lie only on sparse vertices, while intermediate regions are largely OOD. Although the risk model mitigates this issue, its effectiveness is limited by the vast OOD space. Importantly, NAS tasks do not exhibit such extreme sparsity, as their discrete operations have low cardinality and structured choices; therefore, DOMOO still

ranks among the top methods on most NAS subtasks. **(5)** On several real-world tasks (e.g., MORL and Sci-Design), methods such as MultipleModels+COMs or MultipleModels+IOM occasionally achieve higher surrogate-based HV scores than DOMOO. We attribute this to a difference in design philosophy: these baselines incorporate conservatism primarily at the per-objective surrogate level, which reduces but does not eliminate the risk of surrogate overestimation under the multi-objective Pareto dominance structure, where overestimation in a single objective can still cause a solution to incorrectly dominate others. DOMOO, in contrast, integrates risk suppression directly into the preference gradient update (Eq. (2)), providing preference-dependent, multi-objective-aware risk modulation. The Pareto set model is then optimized via standard gradient descent (Eq. (3)) under these risk-guided preferences. It trades slight surrogate HV for higher true reliability. In real-world applications such as molecular design and robotics control, deploying solutions with falsely predicted high efficacy can lead to costly experimental failures or safety risks; thus, robustness to surrogate errors often outweighs nominal surrogate scores. Importantly, DOMOO achieves the best average rank across all benchmarks, indicating superior holistic performance rather than overfitting to specific tasks.

Consequently, baseline methods relying on evolutionary search are less impacted by such discrete optimization tasks. In a nutshell, the results verify that DOMOO can handle offline MOO tasks well and achieves superior optimization performance compared to other offline MOO methods, which answers Q1.

**About Diversity (To Q2).** Table 2 reports the average IGD$_{offline}$ ranks based on the 100$^{th}$ percentile. Detailed results, including the 50$^{th}$ percentile, are provided in the Appendix H.1 and Appendix H.2. We make the following observations: **(1)** As shown in Table 2, DOMOO achieves the best average ranks on most tasks, highlighting its strong

*Table 3.* Ablation Study on the HV and IGD$_{\text{offline}}$ Indicator Performance of DOMOO.

| Metric | Methods | DTLZ3 | IN-1K/MOP7 | MO-Hopper | Regex | RE24 |
|---|---|---|---|---|---|---|
| HV | *w.o. ARC* | $10.61 \pm 0.02$ | $4.45 \pm 0.06$ | $5.49 \pm 0.52$ | $5.72 \pm 0.27$ | $4.84 \pm 0.00$ |
| | *w.o. NPSL* | $10.62 \pm 0.02$ | $3.89 \pm 0.43$ | $5.44 \pm 0.60$ | $4.98 \pm 0.33$ | $4.84 \pm 0.00$ |
| | *w.o. PSMG* | $10.61 \pm 0.02$ | $4.31 \pm 0.15$ | $5.87 \pm 0.00$ | $3.68 \pm 0.21$ | $4.83 \pm 0.00$ |
| | *w.o. SMG* | $9.72 \pm 0.45$ | $3.82 \pm 0.15$ | $6.30 \pm 0.11$ | $6.11 \pm 0.33$ | $4.83 \pm 0.00$ |
| | *w.o. DDSS* | $10.62 \pm 0.01$ | $4.43 \pm 0.07$ | $5.36 \pm 0.51$ | $5.25 \pm 0.35$ | $4.83 \pm 0.01$ |
| | DOMOO | $\mathbf{10.63 \pm 0.01}$ | $\mathbf{4.48 \pm 0.08}$ | $\mathbf{6.43 \pm 0.24}$ | $\mathbf{6.52 \pm 0.11}$ | $\mathbf{4.84 \pm 0.00}$ |
| IGD$_{\text{offline}}$ | *w.o. ARC* | $0.15 \pm 0.01$ | $0.38 \pm 0.03$ | $0.76 \pm 0.11$ | $1.08 \pm 0.04$ | $0.02 \pm 0.02$ |
| | *w.o. NPSL* | $0.15 \pm 0.01$ | $0.53 \pm 0.09$ | $0.78 \pm 0.11$ | $1.04 \pm 0.03$ | $0.01 \pm 0.02$ |
| | *w.o. PSMG* | $0.16 \pm 0.03$ | $\mathbf{0.34 \pm 0.03}$ | $0.65 \pm 0.00$ | $1.09 \pm 0.01$ | $0.03 \pm 0.02$ |
| | *w.o. SMG* | $0.24 \pm 0.03$ | $0.51 \pm 0.01$ | $0.61 \pm 0.03$ | $0.88 \pm 0.04$ | $0.02 \pm 0.02$ |
| | *w.o. DDSS* | $0.15 \pm 0.01$ | $0.38 \pm 0.04$ | $0.78 \pm 0.10$ | $1.08 \pm 0.04$ | $0.02 \pm 0.02$ |
| | DOMOO | $\mathbf{0.14 \pm 0.01}$ | $0.38 \pm 0.03$ | $\mathbf{0.58 \pm 0.07}$ | $\mathbf{0.88 \pm 0.04}$ | $\mathbf{0.01 \pm 0.02}$ |

solution diversity. **(2)** We observe that on RE tasks, most methods outperform the offline dataset in terms of HV, yet many perform worse when evaluated by IGD$_{\text{offline}}$. This discrepancy highlights practical limitations of HV in offline settings: inaccurate reference-point estimation and model-induced errors can make HV fail to faithfully reflect the diversity of the solution set. IGD$_{\text{offline}}$ penalizes unbalanced distributions, providing a more informative assessment of overall coverage. Overall, the results indicate that DOMOO makes a better trade-off between the convergence and diversity of the solutions, which answers Q2.

### 5.3. Ablation Study

**About the Contribution of Key Modules (To Q3).** We evaluate the contribution of DOMOO's components by comparing the full method against five ablated variants: w.o. ARC (removing accumulative risk control), w.o. NPSL (removing nested Pareto set learning), w.o. PSMG (removing Pareto set model generation), w.o. SMG (removing surrogate model generation), and w.o. DDSS (removing diversity-driven solution selection, replacing it with standard HV selection). Detailed configurations for these variants are provided in Appendix J. The results are reported in Table 3. We observe that removing w.o. NPSL leads to the most significant performance drop, confirming the importance of preference-conditioned solution refinement. Excluding the surrogate model (w.o. SMG) also notably degrades solution quality. Furthermore, the drops in w.o. PSMG and w.o. DDSS verify their roles in maintaining solution diversity, while w.o. ARC proves the necessity of risk-aware optimization. In summary, all modules contribute meaningfully to DOMOO's overall performance, which answers Q3.

### 5.4. Hyper-Parameter Analysis

**About the Impact of Hyper-Parameters in DOMOO (To Q4).** We found that the chosen hyper-parameters, the exploration steps in nested Pareto set learning $T_{\text{exp}}$, the scaling factor $\beta$ in IGD$_{\text{offline}}$, $K$ in DDSS and the risk ratio of the energy models, verify robust performance across most ex-

periments, with only a few discrete problems necessitating fine-tuning. For further details, refer to Appendix K.

## 6. Conclusion and Discussion

This paper focuses on achieving better diversity while maintaining satisfactory convergence of the solution set in offline MOO and proposes DOMOO, an offline MOO method via Nested Pareto Set Learning. DOMOO integrates nested Pareto set learning with risk control and the proposed diversity-driven solution selection strategy to efficiently generate diverse and reliable solutions in offline MOO.

Despite its promising performance, DOMOO has several limitations that point to directions for future work. First, DOMOO is relatively less effective on highly discrete tasks with high-cardinality one-hot encodings, where the continuous optimization paradigm struggles with the sparse, vertex-only structure of the search space. A promising remedy is to adopt more compact latent representations (e.g., VAEs) or to develop hybrid architectures that can directly operate on categorical variables. Second, the risk control mechanism relies on the energy model's ability to coarsely distinguish ID from OOD regions; while our sensitivity analysis shows robustness to model capacity, developing calibration methods with formal guarantees on risk estimation quality is an important direction. Third, extending DOMOO to higher-dimensional objective spaces remains an open challenge due to the increased complexity of Pareto front representation and the growing difficulty of preference sampling in high-dimensional simplices. Finally, while DOMOO shows strong performance across benchmarks, further validation on a broader range of real-world deployment scenarios, particularly in safety-critical applications, is needed.

## Acknowledgments

We would like to thank the anonymous reviewers for their constructive comments. This work is supported by the National Key Research and Development Program of China

under Grant 2024YFC3308503, the National Natural Science Foundation of China under Grant 62476091, and Ant Group.

## Impact Statement

This paper presents work whose goal is to advance the field of Machine Learning. There are many potential societal consequences of our work, none of which we feel must be specifically highlighted here.

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

## A. Energy Model Training and Risk Suppression Factor

### A.1. Train the Energy Model

To train the energy model to identity low-risk and high-risk solutions, ARCOO employs Contrastive Divergence (CD) ([Hinton, 2002](#)):

$$\mathcal{L}_{\text{CD}}(\boldsymbol{\omega}) = \mathbb{E}_{\boldsymbol{x} \sim \mathcal{P}}[E_{\boldsymbol{\omega}}(\boldsymbol{x})] - \mathbb{E}_{\boldsymbol{x} \sim \mathcal{Q}}[E_{\boldsymbol{\omega}}(\boldsymbol{x})] \,, \tag{5}$$

where $\mathcal{P}$ denotes the low-risk distribution and $Q$ denotes the high-risk distribution.

Before training the energy model, the high-risk distribution $\mathcal{Q}$ is still unfulfilled. Since $\mathcal{Q}$ is intended to represent OOD solutions that are prone to overestimation, ARCOO adopts Markov Chain Monte Carlo (MCMC) methods ([Geyer, 1992](#); [Welling & Teh, 2011](#)) with Langevin dynamics $LD_{\boldsymbol{\psi}}$ ([Nijkamp et al., 2019](#); [Du & Mordatch, 2019](#)) kernel to sample such solutions. Let $\mathcal{Q} = LD_{\boldsymbol{\psi}}(\mathcal{P}; K_{\text{LD}})$, $\boldsymbol{x}_0 \sim \mathcal{P}$, $\boldsymbol{x}_k \sim \mathcal{Q}^k$, and $\mathcal{Q}^k$ is sampled as:

$$\boldsymbol{x}_k \leftarrow \boldsymbol{x}_{k-1} + \eta \nabla_{\boldsymbol{x}} \hat{f}_{\boldsymbol{\psi}}(\boldsymbol{x}_{k-1}) + \boldsymbol{\alpha}_k, \quad k = 1, \ldots, K_{\text{LD}} \,, \tag{6}$$

where $\alpha_{k,i}$ denotes the $i$-th element of the $\boldsymbol{\alpha}_k$, sampled independently as $\alpha_{k,i} \sim \mathcal{N}(0, \eta)$ and $K_{\text{LD}}$ is the total number of steps. Starting from a sample $\boldsymbol{x}_0$ drawn from the low-risk distribution $\mathcal{P}$, the Langevin dynamics $LD_{\boldsymbol{\psi}}(\mathcal{P}; K_{\text{LD}})$ performs $K_{\text{LD}}$ iterations of noisy gradient ascent to approximate a distribution $\mathcal{Q}$ that concentrates on overestimated OOD solutions.

### A.2. Risk Suppression Factor

After training the energy model $E_{\boldsymbol{\omega}}$, we use the output of the energy model $E_{\boldsymbol{\omega}}(\boldsymbol{x})$, to compute a risk suppression factor $R(\boldsymbol{x})$, defined as follows:

$$R(\boldsymbol{x}) = \frac{c \left( E_{\tilde{Q}} - E_{\boldsymbol{\omega}}(\boldsymbol{x}) \right)}{E_{\tilde{Q}} - E_{\tilde{P}}} \,, \tag{7}$$

where $E_{\widetilde{Q}} = \mathbb{E}_{\boldsymbol{x}' \sim \widetilde{Q}}[E_{\boldsymbol{\omega}}(\boldsymbol{x}')]$, $E_{\widetilde{P}} = \mathbb{E}_{\boldsymbol{x}' \sim \widetilde{P}}[E_{\boldsymbol{\omega}}(\boldsymbol{x}')]$ and $c$ denotes the initial momentum. The $\widetilde{P}$ represents the empirical distribution over the high-quality batch of solutions in the offline dataset. The $\widetilde{Q}$ represents the high-risk distribution sampled by Langevin dynamics starting from $\widetilde{P}$. With the risk suppression factor, we can suppress the risk to a corresponding level in each iteration of nested Pareto set learning.

## B. Pseudo-Code of DOMOO

The pseudo-code of DOMOO is shown in Algorithm 1. The algorithm aims to solve offline multi-objective optimization problems and obtain a solution set with satisfactory diversity and convergence. Given an offline dataset $\mathcal{D}$, DOMOO trains the surrogate objectives $\hat{\boldsymbol{f}}$ and learns a Pareto set model that maps diverse preference vectors to corresponding Pareto-optimal solutions. The inputs to the algorithm include the offline dataset $\mathcal{D}$, valid preferences $\Lambda$, offline preferences $\Lambda_{\text{offline}}$, the number of objectives $M$, total optimization steps $T$, the number of pretraining steps $T_{\text{pre}}$, the number of exploration steps $T_{\text{exp}}$, number of candidate solutions $K$, and batch size $B$.

At the beginning, DOMOO trains $M$ surrogate models $\hat{f}_i$ for each objective using the offline dataset $\mathcal{D}$ and initializes a Pareto set model $h_{\boldsymbol{\phi}}$, as shown in lines 1-2. The Pareto set learning is performed in a nested manner. In the inner loop (lines 5-14), DOMOO updates the preferences depending on the current phase. During the pretraining phase ($t \leq T_{\text{pre}}$), preferences are sampled directly from the offline preference set $\Lambda_{\text{offline}}$, providing a better initialization for the subsequent training stages, as shown in line 7. In the exploration phase ($T_{\text{pre}} < t \leq T_{\text{pre}} + T_{\text{exp}}$), preferences are sampled from the Dirichlet distribution over the preference set $\Lambda$, i.e., $\boldsymbol{\lambda}_t^{(b)} \sim \text{Dirichlet}(\alpha) \subset \Lambda$, enabling the model to be trained over the entire preference space, as shown in line 9. In the later stage ($t > T_{\text{pre}} + T_{\text{exp}}$), preferences are updated via gradient descent according to Equation 2 to guide the Pareto set model to focus on regions where its performance is lacking, as shown in lines 11-13.

In the outer loop (lines 15-19), the updated preferences are used to train the Pareto set model via gradient descent according to Equation 3. After the nested Pareto set learning, diverse preferences are sampled again, and the trained Pareto set model generates candidate solutions (lines 21-22). These candidate solutions are merged with solutions generated by the surrogate model to form a comprehensive candidate set (line 23).

---

**Algorithm 1** Diversity-Driven Offline Multi-Objective Optimization via Nested Pareto Set Learning

---

**Input:** Offline dataset $\mathcal{D}$, valid preferences $\Lambda$, offline preferences $\Lambda_{\text{offline}}$, objective number $M$, total steps $T$, pretraining steps $T_{\text{pre}}$, exploration steps $T_{\text{exp}}$, candidate number $K$, batch size $B$

**Procedure:**

1: Train surrogate model $\hat{\boldsymbol{f}}(\boldsymbol{x}) = (\hat{f}_1(\boldsymbol{x}; \boldsymbol{\theta}_1^*), \cdots, \hat{f}_M(\boldsymbol{x}; \boldsymbol{\theta}_M^*))$ using $\mathcal{D}$

2: Initialize Pareto set model $h_{\boldsymbol{\phi}} : \boldsymbol{\lambda} \mapsto \boldsymbol{x}$

3: **/\* Nested Pareto Set Learning \*/**

4: **for** $t = 1$ to $T$ **do**

5:     **/\* Inner-Loop Preference Update \*/**

6:     **if** $t \leq T_{\text{pre}}$ **then**

7:         Sample preferences $\Lambda_t = \{\boldsymbol{\lambda}_t^{(b)} \sim \Lambda_{\text{offline}}\}_{b=1}^B$                    $\triangleright$ Pretraining phase

8:     **else if** $T_{\text{pre}} < t \leq T_{\text{pre}} + T_{\text{exp}}$ **then**

9:         Sample preferences $\Lambda_t = \{\boldsymbol{\lambda}_t^{(b)} \sim \text{Dirichlet}(\alpha) \subset \Lambda\}_{b=1}^B$        $\triangleright$ Exploration phase

10:     **else**

11:         Generate $\boldsymbol{X}_{t-1} = \{\boldsymbol{x}_{t-1}^{(b)} = h_{\boldsymbol{\phi}}(\boldsymbol{\lambda}_{t-1}^{(b)})\}_{b=1}^B$        $\triangleright$ Preference gradient update phase

12:         Evaluate objective values via the surrogate model

13:         Find preference vectors $\Lambda_t = \{\boldsymbol{\lambda}_t^{(b)}\}_{b=1}^B$ via gradient descent according to Equation 2

14:     **end if**

15:     **/\* Outer-Loop Set Model Update \*/**

16:     Generate $\boldsymbol{X}_t = \{\boldsymbol{x}_t^{(b)} = h_{\boldsymbol{\phi}}(\boldsymbol{\lambda}_t^{(b)})\}_{b=1}^B$

17:     Evaluate objective values via the surrogate model

18:     Update Pareto set model parameters $\boldsymbol{\phi}$ via gradient descent according to Equation 3

19: **end for**

20: **/\* Candidate Solution Generation \*/**

21: Sample diverse candidate preferences $\Lambda_{\text{ps}} = \{\boldsymbol{\lambda}_{\text{ps}}^{(k)} \sim \text{Dirichlet}(\alpha) \subset \Lambda\}_{k=1}^K$

22: Generate $K$ candidates via the trained Pareto set model $h_{\boldsymbol{\phi}^*}$: $\boldsymbol{X}_{\text{ps}} = \{\boldsymbol{x}_{\text{ps}}^{(k)} = h_{\boldsymbol{\phi}^*}(\boldsymbol{\lambda}_{\text{ps}}^{(k)})\}_{k=1}^K$

23: Merge $\boldsymbol{X}_{\text{ps}}$ with the $K$ solutions generated by the surrogate model $\hat{\boldsymbol{f}}(\cdot)$ and obtain the final $\boldsymbol{X}_{\text{cand}}$

24: **/\* Solution Selection based on Two Indicators \*/**

25: Use the $\text{IGD}_{\text{offline}}$ indicator to select the solutions greedy from $\boldsymbol{X}_{\text{cand}}$ for initial screening

26: Use the HV indicator to select remaining solutions from $\boldsymbol{X}_{\text{cand}}$ for final filling

27: **return** the solution set of the selected Pareto solutions

---

Finally, DOMOO selects the final set of Pareto solutions using a two-stage selection strategy: it first applies the $\text{IGD}_{\text{offline}}$ indicator to select solutions and then uses the HV indicator to fill the remaining solutions, as shown in lines 24-27. This selection mechanism ensures both diversity and convergence of the final solution set.

## C. Task Descriptions

In this section, We describe a set of tasks included in the benchmark, explaining their information in detail[1]. We benchmark our method on Off-MOO-Bench tasks (Xue et al., 2024), including diverse real-world and synthetic tasks. We focus on five distinct task categories[2]. An overview of the tasks is provided in Table 4.

• **Synthetic Function (Synthetic):** This task comprises 16 subtasks, each with 2-3 objectives, aiming to identify potential solutions across the offline dataset. All synthetic problems feature continuous solution spaces. Table 5 provides detailed information about each problem, including the shape of the Pareto front and the reference point.

• **Multi-Objective Neural Architecture Search (MO-NAS):** This task involves 14 subtasks, each aiming to optimize 2-3

---

[1]In this study, we focus on tasks with up to three objectives. This choice is motivated by the significantly increased complexity and computational cost associated with high-dimensional Pareto fronts. To ensure fair comparison and reproducibility under a limited computational budget, we do not evaluate tasks with more than three objectives. Extending our method to higher-dimensional objective spaces is left for future work.

[2]We communicated with the original authors and used updated benchmark data to complete the experimental results for all tasks, rather than relying on those reported in the original paper. As a result, some discrepancies may exist.

*Table 4.* Properties of the tasks.

| Task Name | Dataset size | Dimensions | # Objectives | Search space |
|-----------|--------------|------------|--------------|--------------|
| Synthetic | 60000 | 2-30 | 2-3 | Continuous |
| MO-NAS | 9735-60000 | 5-34 | 2-3 | Categorical |
| MORL | 8571 | 9734 | 2 | Continuous |
|  | 4500 | 10184 | 2 | Continuous |
| Sci-Design | 49001 | 32 | 3 | Continuous |
|  | 42048 | 4 | 2 | Sequence |
|  | 4937 | 4 | 2 | Sequence |
|  | 48000 | 4 | 2 | Sequence |
| RE | 60000 | 3-6 | 2-6 | Continuous & Mixed |

*Table 5.* Problem information and reference point for synthetic functions.

| Name | $D$ | $M$ | Type | Pareto Front Shape | Reference Point |
|------|-----|-----|------|--------------------|-----------------|
| DTLZ1 | 7 | 3 | Continuous | Linear | (558.21, 552.30, 568.36) |
| DTLZ2 | 10 | 3 | Continuous | Concave | (2.77, 2.78, 2.93) |
| DTLZ3 | 10 | 3 | Continuous | Concave | (1703.72, 1605.54, 1670.48) |
| DTLZ4 | 10 | 3 | Continuous | Concave | (3.03, 2.83, 2.78) |
| DTLZ5 | 10 | 3 | Continuous | Concave (2d) | (2.65, 2.61, 2.70) |
| DTLZ6 | 10 | 3 | Continuous | Concave (2d) | (9.80, 9.78, 9.78) |
| DTLZ7 | 10 | 3 | Continuous | Disconnected | (1.10, 1.10, 33.43) |
| ZDT1 | 30 | 2 | Continuous | Convex | (1.10, 8.58) |
| ZDT2 | 30 | 2 | Continuous | Concave | (1.10, 9.59) |
| ZDT3 | 30 | 2 | Continuous | Disconnected | (1.10, 8.74) |
| ZDT4 | 10 | 2 | Continuous | Convex | (1.10, 300.42) |
| ZDT6 | 10 | 2 | Continuous | Concave | (1.07, 10.27) |
| Omnitest | 2 | 2 | Continuous | Convex | (2.40, 2.40) |
| VLMOP1 | 1 | 2 | Continuous | Concave | (4.0, 4.0) |
| VLMOP2 | 6 | 2 | Continuous | Concave | (1.10, 1.10) |
| VLMOP3 | 2 | 3 | Continuous | Disconnected | (9.07, 66.62, 0.23) |

objectives in neural architecture design, including prediction error, parameter count, edge GPU latency, and so on. Detailed information of these search spaces $\mathcal{X}$ can be found in Table 6.

• **Multi-Objective Reinforcement Learning (MORL)**: This task encompasses two subtasks: (a) MO-Swimmer: This task involves finding a control policy in a 9,734-dimensional space to optimize both speed and energy efficiency for a robot. (b) MO-Hopper: This task involves finding a control policy in a 10,184-dimensional space to optimize 2 objectives related to running and jumping for a single-legged robot.

• **Scientific Design (Sci-Design)**: This task includes four representative subtasks: (a) Molecule design-optimization in a pretrained 32-dimensional latent space to improve activity against GSK3$\beta$ and JNK3; (b) Regex-maximizing bigram frequencies in protein sequences; (c) ZINC-optimizing molecular properties (logP and QED) on a small-scale dataset; (d) RFP-large-scale optimization of red fluorescent protein variants for solvent-accessible surface area and stability.

• **Real-World Application (RE)**: The task includes many real-world multi-objective engineering design problems, such as four bar truss design, pressure vessel design, disc brake design, and so on.

*Table 6.* An overview of the search spaces in MO-NAS tasks.

| Search space $\mathcal{X}$ | Type | $D$ | $|\mathcal{X}|$ |
|---|---|---|---|
| NAS-Bench-101 | micro | 26 | 423K |
| NAS-Bench-201 | micro | 6 | 15.6K |
| NATS | macro | 5 | 32.8K |
| DARTS | micro | 32 | $\sim 10^{21}$ |
| ResNet50 | macro | 25 | $\sim 10^{14}$ |
| Transformer | macro | 34 | $\sim 10^{14}$ |
| MNV3 | macro | 21 | $\sim 10^{20}$ |

## D. Details of Compared Methods

In line with Off-MOO-Bench (Xue et al., 2024), our evaluation primarily includes two categories of methods-deep neural network (DNN)-based and Gaussian process (GP)-based approaches. Additionally, to broaden the scope of comparison, we also evaluate a flow-based generative modeling technique.

**DNN-Based Methods.** These methods employ surrogate DNN models combined with evolutionary algorithms for solution optimization. We evaluate three configurations: (a) End-to-End Model (E2E): Directly outputs an m-dimensional objective vector for a given design $x$, enhanced by multi-task learning (Chen et al., 2018; Yu et al., 2020) for improved objective performance. (b) Multi-Head Model (MH): uses multi-task learning by a single surrogate model, with the same enhancements as the E2E model. (c) Multiple Models (MM): Maintains $m$ independent surrogates, each trained with OOD mitigating techniques, such as COMs (Trabucco et al., 2021), RoMA (Yu et al., 2021), IOM (Qi et al., 2022), ICT (Yuan et al., 2023), and Tri-mentoring (Chen et al., 2023). Following the original study (Xue et al., 2024), we adopt NSGA-II (Deb et al., 2002) as the default evolutionary algorithm.

**GP-Based Methods.** Bayesian optimization computes an acquisition function to guide the selection of solutions, which are then evaluated using a surrogate model. We consider three representative techniques: hypervolume-based qNEHVI (Daulton et al., 2021), scalarization-based qParEGO (Knowles, 2006), and information-theoretic JES (Hvarfner et al., 2022).

**Generative Methods.** We additionally include ParetoFlow (Yuan et al., 2025) as a comparison. It is a flow-based preference-conditioned generator that employs classifier-guided generation and thus trains one surrogate predictor per objective, while conditioning the flow-based generator on uniformly sampled preference weights to produce solutions along the Pareto front.

## E. Training Details

For fair comparison, we adopt the same experimental settings as in the Off-MOO-Bench (Xue et al., 2024). In our method, the predictor network is a multilayer perceptron (MLP) with the following architecture:

$$\text{Input} \rightarrow \text{MLP}(2048) \rightarrow \text{LeakyReLU} \rightarrow \text{MLP}(2048) \rightarrow \text{LeakyReLU} \rightarrow \text{MLP}(1).$$

We use mean squared error (MSE) as the loss function and optimize the network using Adam with a learning rate of $\eta = 0.001$ and exponential learning rate decay $\gamma = 0.98$. The model is trained on the offline dataset for 100 epochs with a batch size of 256. Additionally, we apply data pruning to alleviate model collapse on certain tasks.

For the energy-based model, we use a separate MLP with the following architecture:

$$\text{Input} \rightarrow \text{MLP}(512) \rightarrow \text{LeakyReLU} \rightarrow \text{MLP}(512) \rightarrow \text{LeakyReLU} \rightarrow \text{MLP}(1).$$

The energy-based model is trained using the Adam optimizer with the same learning rate. The energy head is updated via contrastive loss, where negative samples are generated using Langevin dynamics. This model is trained for 50 epochs with a batch size of 256.

We adopt task-specific hyper-parameters for different categories in the Off-MOO-Bench. For MO-NAS tasks, the energy model uses $K = 64$ Langevin steps, the Pareto set model is pre-trained for 100 steps, followed by 400 steps of optimization with randomly sampled preferences and 400 steps of nested PSL optimization. For MORL tasks, due to the extremely high-dimensional input space, we use a smaller configuration with $K = 8$ Langevin steps, 100 pre-training steps, and only 5

steps each for random preference optimization and nested PSL. For all other tasks, we set $K = 42$ for the energy model, and perform 200 steps of pre-training, 200 steps of random preference optimization, and 100 steps of nested PSL.

## F. Computational Cost

All experiments are conducted on a workstation equipped with an Intel(R) Xeon(R) Gold 6354 CPU (3.00GHz) and an NVIDIA RTX 3090 GPU. The total computational cost of our method consists of five main components: training the surrogate model, training the energy model, initializing the Pareto set model, training the Pareto set model, and performing data selection. The corresponding runtime (measured in seconds) is provided in Table 7. Our method is efficient, completing most tasks within 10 minutes.

*Table 7.* Time cost of DOMOO.

| Task | ZDT2 | C-10/MOP1 | MO-Hopper | Zinc | RE23 |
|---|---|---|---|---|---|
| training the surrogate model | 51.55 | 24.70 | 50.23 | 79.15 | 45.60 |
| training the energy model | 397.64 | 87.69 | 36.46 | 308.84 | 364.98 |
| initializing the Pareto set model | 0.41 | 0.49 | 0.30 | 0.30 | 0.42 |
| training the Pareto set model | 3.36 | 3.37 | 0.21 | 3.85 | 3.20 |
| data selection | 30.20 | 26.77 | 41.78 | 17.50 | 57.64 |
| Overall time cost (second) | 483.16 | 143.02 | 128.98 | 409.64 | 471.85 |

*Table 8.* The runtime for each method to complete model training and optimization on the C-10/MOP1 and MO-Hopper tasks (unit: minutes).

| Tasks | C-10/MOP1 | MO-Hopper |
|---|---|---|
| End2End | 1.20 | 1.17 |
| Multihead | 1.24 | 1.17 |
| Multiple Models | 1.70 | 1.80 |
| MOBO | 0.12 | 33.68 |
| DOMOO (ours) | 2.38 | 2.15 |

As shown in Table 8, DOMOO takes longer than some baseline methods due to the additional cumulative risk control module for handling OOD issues. Although DOMOO includes additional components such as the energy model (Table 7), the overall runtime remains moderate. As shown in Table 8, DOMOO is only slightly slower than lightweight surrogate-based baselines-typically within one minute-while remaining competitive or even faster than several existing methods. More importantly, in offline optimization the quality of the obtained Pareto set is far more critical than marginal differences in runtime, since no additional evaluations or online interactions are permitted. The modest overhead introduced by the risk-control module therefore represents a reasonable and practical trade-off.

## G. HV Experiment Results

### G.1. The $100^{th}$ Percentile Results

As shown in Table 9, Table 10, Table 11, Table 12, and Table 13, we report the $100^{th}$ percentile hypervolume results with 256 solutions. DOMOO consistently performs well across tasks. Methods within one standard deviation of the best are highlighted in **bold**.

*Table 9.* Hypervolume results for synthetic functions with 256 solutions and $100^{th}$ percentile evaluations. For each task, algorithms within one standard deviation of having the highest performance are **bolded**.

| Methods | DTLZ1 | DTLZ2 | DTLZ3 | DTLZ4 | DTLZ5 | DTLZ6 | DTLZ7 | OmniTest | VLMOP1 | VLMOP2 | VLMOP3 | ZDT1 | ZDT2 | ZDT3 | ZDT4 | ZDT6 |
|---|---|---|---|---|---|---|---|---|---|---|---|---|---|---|---|---|
| $\mathcal{D}$(best) | 10.6 | 9.91 | 10 | **10.76** | 9.35 | 8.88 | 8.56 | 4.53 | 0.08 | 1.78 | 45.65 | 4.17 | 4.68 | 5.15 | **5.46** | 4.61 |
| End-to-End | 10.63 ± 0.01 | 8.40 ± 1.10 | 10.28 ± 0.32 | 6.98 ± 1.19 | 7.63 ± 1.02 | 8.77 ± 1.20 | 10.68 ± 0.06 | **4.78 ± 0.01** | **0.32 ± 0.00** | 4.23 ± 0.02 | 45.92 ± 0.02 | 4.84 ± 0.01 | 5.66 ± 0.01 | 5.50 ± 0.14 | 4.91 ± 0.13 | **4.78 ± 0.00** |
| End-to-End + GradNorm | 10.64 ± 0.00 | 8.10 ± 1.15 | 10.40 ± 0.27 | 8.01 ± 1.91 | 7.00 ± 1.25 | **9.96 ± 0.43** | 10.73 ± 0.02 | 4.57 ± 0.32 | 0.31 ± 0.00 | 2.14 ± 0.86 | 44.45 ± 1.49 | 4.72 ± 0.03 | 5.51 ± 0.04 | 5.34 ± 0.13 | 4.75 ± 0.48 | 4.66 ± 0.05 |
| End-to-End + PcGrad | 10.64 ± 0.00 | 9.33 ± 1.19 | 10.46 ± 0.24 | 8.76 ± 0.99 | 9.23 ± 0.51 | 9.53 ± 0.31 | 10.66 ± 0.05 | **4.78 ± 0.01** | **0.32 ± 0.00** | 4.23 ± 0.02 | **45.93 ± 0.00** | **4.85 ± 0.00** | 5.64 ± 0.04 | 5.54 ± 0.05 | 3.73 ± 0.26 | 3.91 ± 1.05 |
| Multi Head | **10.65 ± 0.00** | 8.52 ± 1.26 | 10.52 ± 0.16 | 7.15 ± 1.12 | 8.04 ± 0.81 | 8.38 ± 1.31 | 10.66 ± 0.09 | 4.78 ± 0.00 | **0.32 ± 0.00** | **4.24 ± 0.01** | **45.93 ± 0.00** | 4.80 ± 0.04 | 5.55 ± 0.10 | 5.58 ± 0.08 | 4.62 ± 0.30 | **4.78 ± 0.00** |
| Multi Head + GradNorm | 10.64 ± 0.00 | 7.59 ± 1.82 | 10.46 ± 0.24 | 8.14 ± 1.03 | 8.45 ± 0.75 | 9.27 ± 0.99 | 10.18 ± 0.54 | 4.00 ± 0.84 | 0.05 ± 0.11 | 3.34 ± 0.51 | 42.08 ± 5.22 | 4.60 ± 0.19 | 5.36 ± 0.17 | 5.62 ± 0.18 | 3.76 ± 0.39 | 3.91 ± 0.99 |
| Multi Head + PcGrad | 10.64 ± 0.00 | 9.52 ± 0.60 | 10.52 ± 0.09 | 6.46 ± 1.42 | 9.10 ± 0.52 | 9.48 ± 0.44 | 10.62 ± 0.02 | 4.78 ± 0.00 | 0.31 ± 0.01 | 4.21 ± 0.03 | **45.93 ± 0.00** | 4.80 ± 0.09 | 5.57 ± 0.08 | 5.56 ± 0.04 | 4.22 ± 0.54 | 3.67 ± 1.32 |
| Multiple Models | **10.65 ± 0.00** | 8.19 ± 1.41 | 10.61 ± 0.02 | 8.07 ± 0.99 | 8.19 ± 1.04 | 8.52 ± 1.48 | 10.71 ± 0.08 | 4.78 ± 0.00 | **0.32 ± 0.00** | **4.24 ± 0.01** | **45.93 ± 0.00** | 4.80 ± 0.02 | 5.55 ± 0.07 | 5.55 ± 0.18 | 5.25 ± 0.22 | 4.73 ± 0.05 |
| Multiple Models + COMs | 10.64 ± 0.00 | 9.63 ± 0.55 | 10.39 ± 0.11 | 8.03 ± 0.54 | 9.32 ± 0.17 | 8.97 ± 0.28 | 9.94 ± 0.17 | 4.78 ± 0.00 | 0.31 ± 0.01 | 4.18 ± 0.03 | 45.92 ± 0.02 | 4.53 ± 0.04 | 5.11 ± 0.06 | 5.52 ± 0.03 | 5.01 ± 0.10 | 2.70 ± 0.70 |
| Multiple Models + RoMA | 10.64 ± 0.00 | 9.87 ± 0.26 | 10.37 ± 0.29 | 8.71 ± 0.54 | 7.08 ± 0.24 | 9.72 ± 0.07 | 10.57 ± 0.02 | 3.96 ± 0.32 | **0.32 ± 0.00** | 1.44 ± 0.00 | 41.21 ± 4.41 | 4.84 ± 0.00 | 5.59 ± 0.02 | **5.91 ± 0.01** | 3.99 ± 0.20 | 2.08 ± 0.51 |
| Multiple Models + IOM | 10.64 ± 0.00 | 9.81 ± 0.20 | 10.49 ± 0.11 | 9.46 ± 0.82 | **9.57 ± 0.37** | 9.54 ± 0.19 | 10.61 ± 0.06 | 4.78 ± 0.00 | 0.31 ± 0.01 | 3.93 ± 0.32 | **45.93 ± 0.00** | 4.64 ± 0.03 | 5.56 ± 0.06 | 5.64 ± 0.02 | 5.05 ± 0.19 | 4.75 ± 0.02 |
| Multiple Models + ICT | 10.64 ± 0.00 | 9.63 ± 0.55 | 10.18 ± 0.34 | 9.22 ± 1.07 | 8.65 ± 0.76 | 8.94 ± 0.66 | 10.42 ± 0.10 | 4.77 ± 0.02 | **0.32 ± 0.00** | 4.08 ± 0.10 | 45.92 ± 0.01 | 4.81 ± 0.02 | 5.55 ± 0.06 | 5.42 ± 0.18 | 4.37 ± 0.14 | 4.23 ± 0.53 |
| Multiple Models + Tri-Mentoring | 10.50 ± 0.28 | 9.17 ± 0.86 | 10.30 ± 0.37 | 8.92 ± 1.12 | 7.70 ± 1.14 | 9.33 ± 1.00 | 10.07 ± 0.15 | 4.76 ± 0.02 | **0.32 ± 0.00** | 3.97 ± 0.40 | 45.92 ± 0.01 | 4.78 ± 0.01 | 5.55 ± 0.17 | 5.13 ± 0.14 | 5.01 ± 0.12 | 3.40 ± 1.05 |
| MOBO | **10.65 ± 0.00** | 10.24 ± 0.09 | 10.35 ± 0.03 | 10.59 ± 0.01 | 9.23 ± 0.00 | 9.38 ± 0.18 | 10.34 ± 0.03 | 4.78 ± 0.00 | **0.32 ± 0.00** | 2.77 ± 0.03 | N/A | 4.34 ± 0.01 | 5.01 ± 0.00 | 5.32 ± 0.01 | 4.56 ± 0.08 | 3.18 ± 0.05 |
| MOBO-qParEGO | **10.65 ± 0.00** | 10.00 ± 0.02 | 10.17 ± 0.01 | 9.48 ± 0.70 | 9.38 ± 0.16 | 8.96 ± 0.31 | 10.19 ± 0.02 | 4.78 ± 0.00 | **0.32 ± 0.00** | 3.59 ± 0.15 | **45.93 ± 0.00** | 4.35 ± 0.02 | 5.08 ± 0.06 | 5.27 ± 0.02 | 5.01 ± 0.07 | 2.73 ± 0.25 |
| MOBO-JES | N/A | N/A | N/A | N/A | N/A | N/A | N/A | 4.70 ± 0.06 | 0.30 ± 0.00 | N/A | N/A | 4.01 ± 0.04 | 4.94 ± 0.07 | 5.10 ± 0.05 | 4.39 ± 0.08 | 2.73 ± 0.25 |
| ParetoFlow | 10.61 ± 0.02 | **10.30 ± 0.11** | 10.36 ± 0.09 | 10.46 ± 0.21 | 9.53 ± 0.31 | 9.62 ± 0.07 | 9.04 ± 0.10 | 4.78 ± 0.00 | N/A | 4.22 ± 0.00 | N/A | 4.19 ± 0.05 | **5.94 ± 0.36** | 5.21 ± 0.12 | 4.97 ± 0.12 | 4.50 ± 0.05 |
| DOMOO (ours) | **10.65 ± 0.00** | 9.91 ± 0.18 | **10.63 ± 0.01** | 9.49 ± 0.33 | 9.35 ± 0.19 | 9.41 ± 0.93 | **10.73 ± 0.03** | 4.78 ± 0.00 | **0.32 ± 0.00** | 4.21 ± 0.02 | **45.93 ± 0.00** | **4.85 ± 0.00** | 5.70 ± 0.00 | 5.62 ± 0.11 | 5.29 ± 0.16 | 4.75 ± 0.02 |

*Table 10.* Hypervolume results for MO-NAS with 256 solutions and $100^{th}$ percentile evaluations. For each task, algorithms within one standard deviation of having the highest performance are **bolded**.

| Methods | C-10/MOP1 | C-10/MOP2 | C-10/MOP3 | C-10/MOP8 | C-10/MOP9 | IN-1K/MOP1 | IN-1K/MOP2 | IN-1K/MOP3 | IN-1K/MOP4 | IN-1K/MOP5 | IN-1K/MOP6 | IN-1K/MOP7 | IN-1K/MOP8 | NasBench201-Test |
|---|---|---|---|---|---|---|---|---|---|---|---|---|---|---|
| $\mathcal{D}$(best) | 4.72 | 10.42 | 9.21 | 4.38 | 9.64 | 4.36 | 4.45 | 9.86 | 4.15 | 4.3 | 9.15 | 3.7 | 9.13 | 9.89 |
| End-to-End | 4.75 ± 0.01 | 10.46 ± 0.01 | 10.19 ± 0.02 | 4.64 ± 0.09 | 10.21 ± 0.16 | 4.53 ± 0.08 | 4.54 ± 0.03 | 9.98 ± 0.03 | **4.58 ± 0.10** | 4.60 ± 0.05 | 10.00 ± 0.24 | 4.04 ± 0.31 | 9.38 ± 0.11 | 10.19 ± 0.10 |
| End-to-End + GradNorm | 4.64 ± 0.04 | 10.43 ± 0.02 | 9.19 ± 0.11 | 4.22 ± 0.16 | 9.92 ± 0.32 | 4.19 ± 0.23 | 4.40 ± 0.06 | 8.42 ± 0.28 | 4.50 ± 0.06 | 4.57 ± 0.02 | 9.59 ± 0.26 | 4.10 ± 0.14 | 8.35 ± 0.16 | 9.06 ± 1.63 |
| End-to-End + PcGrad | 4.75 ± 0.01 | 10.46 ± 0.03 | 10.17 ± 0.01 | 4.61 ± 0.04 | **10.29 ± 0.07** | 4.50 ± 0.05 | 4.51 ± 0.06 | 9.97 ± 0.09 | 4.36 ± 0.14 | 4.55 ± 0.06 | 9.74 ± 0.13 | 4.06 ± 0.11 | 9.50 ± 0.07 | 10.15 ± 0.11 |
| Multi Head | 4.75 ± 0.01 | 10.47 ± 0.03 | 10.07 ± 0.03 | 4.59 ± 0.05 | 10.09 ± 0.21 | 4.61 ± 0.04 | 4.51 ± 0.03 | 10.02 ± 0.04 | 4.54 ± 0.05 | **4.65 ± 0.08** | **10.02 ± 0.17** | 4.22 ± 0.17 | 9.51 ± 0.08 | 10.14 ± 0.02 |
| Multi Head + GradNorm | 4.46 ± 0.25 | 10.15 ± 0.23 | 9.36 ± 0.19 | 4.02 ± 0.17 | 8.67 ± 1.27 | 4.28 ± 0.17 | 3.98 ± 0.37 | 8.72 ± 1.07 | 4.41 ± 0.11 | 4.49 ± 0.08 | 9.63 ± 0.36 | 2.99 ± 0.59 | 6.23 ± 1.96 | 9.97 ± 0.24 |
| Multi Head + PcGrad | 4.75 ± 0.02 | 10.47 ± 0.01 | 10.05 ± 0.11 | 4.64 ± 0.03 | 10.28 ± 0.10 | 4.47 ± 0.04 | 4.55 ± 0.03 | 10.02 ± 0.01 | 4.40 ± 0.05 | 4.61 ± 0.01 | 9.75 ± 0.13 | 3.97 ± 0.07 | 9.28 ± 0.27 | **10.23 ± 0.08** |
| Multiple Models | 4.75 ± 0.01 | 10.44 ± 0.01 | 10.08 ± 0.06 | 4.64 ± 0.04 | 10.14 ± 0.13 | 4.43 ± 0.22 | 4.51 ± 0.04 | 9.97 ± 0.05 | 4.53 ± 0.06 | 4.61 ± 0.03 | 9.78 ± 0.20 | 4.17 ± 0.20 | 9.57 ± 0.05 | 10.08 ± 0.19 |
| Multiple Models + COMs | **4.76 ± 0.02** | 10.44 ± 0.01 | 10.14 ± 0.02 | 4.61 ± 0.07 | 10.03 ± 0.22 | 4.60 ± 0.03 | 4.54 ± 0.02 | 10.03 ± 0.05 | 4.43 ± 0.10 | 4.54 ± 0.04 | 9.90 ± 0.15 | 4.07 ± 0.05 | 9.52 ± 0.11 | 10.18 ± 0.07 |
| Multiple Models + RoMA | 4.75 ± 0.01 | 10.46 ± 0.01 | 10.17 ± 0.02 | 4.32 ± 0.05 | 9.84 ± 0.11 | 4.55 ± 0.07 | 4.56 ± 0.04 | 9.94 ± 0.01 | 4.53 ± 0.05 | 4.63 ± 0.05 | 9.91 ± 0.12 | 4.37 ± 0.09 | 9.39 ± 0.11 | 10.00 ± 0.28 |
| Multiple Models + IOM | 4.73 ± 0.02 | 10.38 ± 0.05 | 10.09 ± 0.04 | **4.68 ± 0.02** | 10.24 ± 0.12 | 4.63 ± 0.03 | 4.59 ± 0.03 | **10.06 ± 0.02** | 4.42 ± 0.06 | 4.58 ± 0.03 | 9.71 ± 0.10 | 4.18 ± 0.18 | **9.69 ± 0.04** | 10.20 ± 0.10 |
| Multiple Models + ICT | 4.73 ± 0.02 | 10.43 ± 0.16 | 9.82 ± 0.24 | 4.28 ± 0.24 | 9.54 ± 0.34 | 4.42 ± 0.07 | 4.42 ± 0.07 | 9.80 ± 0.15 | 4.53 ± 0.07 | 4.50 ± 0.05 | 9.99 ± 0.11 | 3.98 ± 0.13 | 9.01 ± 0.56 | 10.20 ± 0.13 |
| Multiple Models + Tri-Mentoring | 4.73 ± 0.02 | **10.49 ± 0.05** | 10.18 ± 0.01 | 4.29 ± 0.09 | 8.87 ± 0.33 | 4.42 ± 0.12 | 4.26 ± 0.10 | 9.47 ± 0.22 | 4.39 ± 0.05 | 4.47 ± 0.06 | 9.80 ± 0.14 | 4.08 ± 0.22 | 9.49 ± 0.14 | 9.37 ± 0.24 |
| MOBO | 4.76 ± 0.01 | 10.49 ± 0.02 | **10.22 ± 0.00** | 4.61 ± 0.02 | 10.24 ± 0.07 | 4.67 ± 0.02 | 4.56 ± 0.02 | 10.05 ± 0.01 | 4.39 ± 0.04 | 4.56 ± 0.03 | 9.69 ± 0.02 | 4.18 ± 0.08 | 9.65 ± 0.02 | N/A |
| MOBO-qParEGO | 4.75 ± 0.01 | 10.45 ± 0.07 | 8.55 ± 0.18 | 4.46 ± 0.04 | 9.98 ± 0.09 | 4.22 ± 0.04 | 4.17 ± 0.06 | 9.27 ± 0.03 | 4.09 ± 0.02 | 4.32 ± 0.10 | 9.18 ± 0.15 | 4.10 ± 0.02 | 9.09 ± 0.05 | N/A |
| MOBO-JES | N/A | N/A | N/A | N/A | N/A | N/A | N/A | N/A | N/A | N/A | N/A | N/A | N/A | N/A |
| ParetoFlow | 4.74 ± 0.03 | 10.46 ± 0.01 | 9.44 ± 0.10 | 4.46 ± 0.04 | 9.76 ± 0.00 | 4.36 ± 0.03 | 4.33 ± 0.06 | 9.79 ± 0.05 | 4.32 ± 0.07 | N/A | | 3.82 ± 0.02 | 9.19 ± 0.00 | N/A |
| DOMOO (ours) | 4.74 ± 0.01 | 10.42 ± 0.01 | 10.01 ± 0.09 | 4.65 ± 0.03 | 10.19 ± 0.04 | **4.68 ± 0.04** | **4.59 ± 0.04** | 9.96 ± 0.06 | 4.46 ± 0.10 | 4.58 ± 0.04 | 9.48 ± 0.27 | **4.48 ± 0.08** | 9.55 ± 0.02 | 10.16 ± 0.06 |

*Table 11.* Hypervolume results for MORL with 256 solutions and $100^{th}$ percentile evaluations. For each task, algorithms within one standard deviation of having the highest performance are **bolded**.

| Methods | MO-Swimmer | MO-Hopper |
|---|---|---|
| $\mathcal{D}$(best) | 3.64 | 5.67 |
| End-to-End | 3.62 ± 0.00 | 6.04 ± 0.00 |
| End-to-End + GradNorm | 2.96 ± 0.00 | 5.69 ± 0.00 |
| End-to-End + PcGrad | 3.43 ± 0.00 | 5.63 ± 0.00 |
| Multi Head | 3.29 ± 0.00 | 5.83 ± 0.00 |
| Multi Head + GradNorm | 3.37 ± 0.00 | 4.77 ± 0.00 |
| Multi Head + PcGrad | 3.08 ± 0.00 | 6.06 ± 0.00 |
| Multiple Models | 3.49 ± 0.00 | 5.87 ± 0.00 |
| Multiple Models + COMs | **3.87 ± 0.00** | 6.15 ± 0.00 |
| Multiple Models + RoMA | 3.50 ± 0.00 | 6.03 ± 0.00 |
| Multiple Models + IOM | 3.59 ± 0.00 | 6.24 ± 0.00 |
| Multiple Models + ICT | 3.45 ± 0.26 | 5.73 ± 0.34 |
| Multiple Models + Tri-Mentoring | 3.42 ± 0.18 | 5.86 ± 0.14 |
| MOBO | N/A | N/A |
| MOBO-qParEGO | N/A | N/A |
| MOBO-JES | N/A | N/A |
| ParetoFlow | 3.41 ± 0.08 | 5.65 ± 0.00 |
| DOMOO (ours) | 3.61 ± 0.00 | **6.43 ± 0.24** |

*Table 12.* Hypervolume results for RE with 256 solutions and $100^{th}$ percentile evaluations. For each task, algorithms within one standard deviation of having the highest performance are **bolded**.

| Methods | RE21 | RE22 | RE23 | RE24 | RE25 | RE31 | RE32 | RE33 | RE34 | RE35 | RE36 | RE37 | MO-Portfolio |
|---|---|---|---|---|---|---|---|---|---|---|---|---|---|
| $\mathcal{D}$(best) | 4.1 | 4.78 | 4.75 | 4.6 | 4.79 | 10.6 | 10.56 | 10.56 | 9.3 | 10.08 | 7.61 | 5.57 | 4.24 |
| End-to-End | **4.60 ± 0.00** | **4.84 ± 0.00** | **4.84 ± 0.01** | 4.65 ± 0.22 | **4.84 ± 0.01** | 10.55 ± 0.20 | **10.65 ± 0.00** | 10.61 ± 0.01 | 10.10 ± 0.01 | 10.38 ± 0.06 | 10.19 ± 0.07 | 6.67 ± 0.05 | 4.43 ± 0.03 |
| End-to-End + GradNorm | 4.57 ± 0.02 | **4.84 ± 0.00** | 4.12 ± 0.79 | 3.94 ± 1.06 | 4.80 ± 0.03 | 8.52 ± 4.26 | 10.64 ± 0.01 | 10.57 ± 0.02 | 9.80 ± 0.11 | 10.35 ± 0.01 | 0.02 ± 0.00 | 6.56 ± 0.03 | 4.41 ± 0.01 |
| End-to-End + PcGrad | **4.60 ± 0.00** | **4.84 ± 0.00** | 4.84 ± 0.00 | 4.43 ± 0.18 | 4.82 ± 0.01 | **10.65 ± 0.00** | 10.61 ± 0.02 | 10.59 ± 0.03 | 10.11 ± 0.01 | 10.55 ± 0.02 | 10.06 ± 0.10 | 6.68 ± 0.04 | 4.45 ± 0.03 |
| Multi Head | **4.60 ± 0.00** | **4.84 ± 0.00** | 4.84 ± 0.00 | **4.84 ± 0.00** | 4.84 ± 0.00 | **10.65 ± 0.00** | 10.64 ± 0.00 | 10.62 ± 0.00 | 10.11 ± 0.00 | 10.41 ± 0.08 | 10.16 ± 0.08 | 6.70 ± 0.02 | 4.39 ± 0.07 |
| Multi Head + GradNorm | 4.30 ± 0.40 | 4.26 ± 0.74 | 3.96 ± 1.06 | 3.94 ± 1.07 | 4.72 ± 0.10 | 10.08 ± 0.48 | 10.56 ± 0.12 | 9.31 ± 1.65 | 10.01 ± 0.11 | 10.23 ± 0.38 | 7.87 ± 3.39 | 6.11 ± 0.97 | 4.26 ± 0.12 |
| Multi Head + PcGrad | 4.59 ± 0.01 | 3.87 ± 1.94 | 4.84 ± 0.00 | 3.07 ± 0.53 | 4.79 ± 0.06 | **10.65 ± 0.00** | 10.63 ± 0.00 | 10.61 ± 0.01 | 10.11 ± 0.01 | 10.56 ± 0.03 | 9.73 ± 0.21 | 6.69 ± 0.04 | 4.33 ± 0.08 |
| Multiple Models | **4.60 ± 0.00** | **4.84 ± 0.00** | 4.83 ± 0.02 | 4.82 ± 0.03 | 4.64 ± 0.24 | **10.65 ± 0.00** | 10.60 ± 0.02 | 10.62 ± 0.00 | 10.11 ± 0.00 | 10.56 ± 0.01 | 10.23 ± 0.04 | 6.74 ± 0.01 | 4.59 ± 0.29 |
| Multiple Models + COMs | 4.36 ± 0.06 | 4.82 ± 0.01 | 4.83 ± 0.02 | 4.83 ± 0.00 | 4.83 ± 0.01 | 10.62 ± 0.02 | 10.64 ± 0.01 | 10.62 ± 0.01 | 9.94 ± 0.11 | 10.54 ± 0.02 | 9.37 ± 0.24 | 6.32 ± 0.07 | 3.64 ± 0.71 |
| Multiple Models + RoMA | 4.57 ± 0.00 | 4.83 ± 0.02 | 4.83 ± 0.01 | 3.85 ± 1.00 | 4.83 ± 0.01 | **10.65 ± 0.00** | **10.65 ± 0.00** | 10.57 ± 0.04 | 9.92 ± 0.01 | 10.56 ± 0.02 | 9.93 ± 0.11 | 6.67 ± 0.02 | 4.41 ± 0.09 |
| Multiple Models + IOM | 4.59 ± 0.00 | **4.84 ± 0.00** | 4.83 ± 0.01 | 4.82 ± 0.01 | 4.84 ± 0.00 | **10.65 ± 0.00** | 10.64 ± 0.00 | 10.60 ± 0.03 | 10.11 ± 0.00 | 10.58 ± 0.01 | 10.06 ± 0.20 | 6.70 ± 0.01 | 4.52 ± 0.30 |
| Multiple Models + ICT | **4.60 ± 0.00** | **4.84 ± 0.00** | 4.73 ± 0.17 | 4.65 ± 0.22 | 4.84 ± 0.00 | **10.65 ± 0.00** | 10.64 ± 0.00 | 10.61 ± 0.01 | 10.09 ± 0.01 | 10.56 ± 0.01 | 10.15 ± 0.12 | 6.73 ± 0.01 | 4.56 ± 0.23 |
| Multiple Models + Tri-Mentoring | **4.60 ± 0.00** | **4.84 ± 0.00** | 4.73 ± 0.06 | 4.83 ± 0.00 | 4.84 ± 0.00 | **10.65 ± 0.00** | 10.63 ± 0.01 | 10.61 ± 0.02 | 10.05 ± 0.05 | 10.45 ± 0.25 | 10.02 ± 0.11 | 6.73 ± 0.01 | 4.54 ± 0.17 |
| MOBO | 4.37 ± 0.06 | **4.84 ± 0.00** | 4.84 ± 0.00 | 4.83 ± 0.00 | 4.84 ± 0.00 | 10.20 ± 0.00 | **10.65 ± 0.00** | 10.63 ± 0.00 | 9.71 ± 0.00 | 10.57 ± 0.01 | **10.26 ± 0.00** | 6.78 ± 0.00 | 0.99 ± 1.23 |
| MOBO-qParEGO | 4.58 ± 0.01 | **4.84 ± 0.00** | 4.84 ± 0.00 | 4.83 ± 0.00 | 4.84 ± 0.00 | **10.65 ± 0.00** | 10.64 ± 0.00 | 10.59 ± 0.01 | 9.12 ± 0.05 | 10.50 ± 0.01 | 10.14 ± 0.00 | 6.61 ± 0.07 | 2.77 ± 2.49 |
| MOBO-JES | 4.51 ± 0.02 | **4.84 ± 0.00** | 4.84 ± 0.00 | 4.82 ± 0.00 | 4.84 ± 0.00 | N/A | N/A | 10.53 ± 0.04 | 9.41 ± 0.00 | 10.55 ± 0.00 | N/A | N/A | 0.00 ± 0.00 |
| ParetoFlow | 4.36 ± 0.20 | 4.78 ± 0.09 | N/A | N/A | N/A | 10.63 ± 0.08 | **11.17 ± 0.00** | **10.78 ± 0.15** | **10.70 ± 0.16** | N/A | 8.43 ± 0.22 | **6.92 ± 0.61** | 4.12 ± 0.10 |
| DOMOO (ours) | **4.60 ± 0.00** | **4.84 ± 0.00** | 4.84 ± 0.00 | **4.84 ± 0.00** | 4.84 ± 0.00 | **10.65 ± 0.00** | 10.64 ± 0.01 | 10.63 ± 0.00 | 10.12 ± 0.00 | **10.59 ± 0.01** | 10.21 ± 0.06 | 6.76 ± 0.00 | **6.33 ± 0.07** |

*Table 13.* Hypervolume results for scientific design with 256 solutions and $100^{th}$ percentile evaluations. For each task, algorithms within one standard deviation of having the highest performance are **bolded**.

| Methods | Molecule | Regex | RFP | ZINC |
|---|---|---|---|---|
| $\mathcal{D}$(best) | 2.91 | 3.96 | 4.06 | 4.52 |
| End-to-End | 2.64 ± 0.12 | 3.76 ± 0.27 | 4.49 ± 0.28 | 4.72 ± 0.05 |
| End-to-End + GradNorm | 0.59 ± 0.72 | 4.72 ± 0.22 | 4.45 ± 0.31 | 4.64 ± 0.08 |
| End-to-End + PcGrad | 2.22 ± 0.57 | 4.61 ± 0.27 | 4.37 ± 0.31 | 4.76 ± 0.01 |
| Multi Head | 2.47 ± 0.12 | 3.98 ± 0.00 | 4.43 ± 0.29 | 4.67 ± 0.04 |
| Multi Head + GradNorm | 2.64 ± 0.33 | 3.93 ± 0.18 | 4.52 ± 0.31 | 4.56 ± 0.02 |
| Multi Head + PcGrad | 2.37 ± 0.40 | 4.56 ± 0.22 | 4.58 ± 0.22 | 4.72 ± 0.06 |
| Multiple Models | 2.59 ± 0.14 | 3.98 ± 0.00 | 4.11 ± 0.04 | 4.74 ± 0.04 |
| Multiple Models + COMs | 3.01 ± 0.09 | 4.61 ± 0.27 | 4.36 ± 0.29 | 4.74 ± 0.04 |
| Multiple Models + RoMA | 2.86 ± 0.72 | 4.61 ± 0.27 | 4.26 ± 0.26 | 4.66 ± 0.01 |
| Multiple Models + IOM | **3.14 ± 0.19** | 4.83 ± 0.00 | 4.28 ± 0.26 | 4.66 ± 0.01 |
| Multiple Models + ICT | 2.83 ± 0.04 | 4.63 ± 0.24 | **4.58 ± 0.25** | 4.68 ± 0.03 |
| Multiple Models + Tri-Mentoring | 1.83 ± 0.30 | 4.72 ± 0.22 | 4.22 ± 0.25 | 4.62 ± 0.05 |
| MOBO | 3.03 ± 0.64 | **6.77 ± 0.04** | 4.05 ± 0.01 | **4.77 ± 0.00** |
| MOBO-qParEGO | N/A | 6.47 ± 0.00 | 3.93 ± 0.03 | 4.61 ± 0.05 |
| MOBO-JES | N/A | N/A | N/A | N/A |
| ParetoFlow | 2.86 ± 0.81 | 3.26 ± 0.00 | 4.35 ± 0.11 | N/A |
| DOMOO (ours) | 2.78 ± 0.13 | 6.52 ± 0.11 | 4.24 ± 0.29 | 4.71 ± 0.06 |

## G.2. The $50^{th}$ Percentile Results

As shown in Table 14, we report the $50^{th}$ percentile HV average ranks with 256 solutions. As shown in Table 15, Table 16, Table 17, Table 18, and Table 19, we report the $50^{th}$ percentile hypervolume results with 256 solutions. DOMOO consistently performs well across tasks. Methods within one standard deviation of the best are highlighted in **bold**.

*Table 14.* Comparison of average HV ranks at the $50^{th}$ percentile achieved by different offline MOO methods across different tasks in Off-MOO-Bench (Xue et al., 2024). For each task, the top three methods are highlighted using **(1st)**, (2nd), and (3rd) formatting. $\mathcal{D}(\text{best})$ denotes the best subset in the offline dataset (i.e., with the highest HV), and the last column reports the average rank across all tasks.

| Methods | Synthetic | MO-NAS | MORL | Sci-Design | RE | Average Rank |
|---|---|---|---|---|---|---|
| $\mathcal{D}(\text{best})$ | $9.45 \pm 0.25$ | $9.06 \pm 0.35$ | $\mathbf{1.10 \pm 0.20}$ | $\mathbf{2.25 \pm 0.45}$ | $12.23 \pm 0.40$ | $9.15 \pm 0.23$ |
| End-to-End | $6.83 \pm 1.01$ | $5.94 \pm 0.36$ | $8.80 \pm 0.60$ | $8.30 \pm 0.81$ | $6.08 \pm 0.64$ | $6.58 \pm 0.30$ |
| End-to-End + GradNorm | $11.73 \pm 1.05$ | $12.99 \pm 1.35$ | $12.90 \pm 0.58$ | $11.65 \pm 1.15$ | $11.32 \pm 0.34$ | $12.02 \pm 0.85$ |
| End-to-End + PcGrad | $6.88 \pm 0.82$ | $7.56 \pm 1.21$ | $7.20 \pm 0.51$ | $8.60 \pm 2.35$ | $7.51 \pm 0.63$ | $7.39 \pm 0.55$ |
| Multi Head | $6.45 \pm 0.61$ | $6.23 \pm 0.45$ | $4.70 \pm 0.24$ | $7.95 \pm 1.08$ | $5.85 \pm 0.60$ | $6.28 \pm 0.32$ |
| Multi Head + GradNorm | $10.74 \pm 0.72$ | $13.27 \pm 0.94$ | $13.00 \pm 0.55$ | $7.42 \pm 1.34$ | $12.25 \pm 1.19$ | $11.69 \pm 0.69$ |
| Multi Head + PcGrad | $7.83 \pm 1.10$ | $7.56 \pm 1.19$ | $8.80 \pm 0.40$ | $9.88 \pm 1.07$ | $9.54 \pm 0.81$ | $8.41 \pm 0.70$ |
| Multiple Models | $5.24 \pm 0.51$ | $6.73 \pm 0.92$ | $9.30 \pm 0.40$ | $7.88 \pm 1.98$ | $5.82 \pm 0.77$ | $6.20 \pm 0.29$ |
| Multiple Models + COM | $8.90 \pm 0.46$ | $6.91 \pm 1.10$ | $5.00 \pm 0.32$ | $7.90 \pm 2.65$ | $9.97 \pm 0.71$ | $8.38 \pm 0.63$ |
| Multiple Models + RoMA | $10.41 \pm 1.05$ | $6.24 \pm 0.84$ | $8.20 \pm 0.40$ | $8.47 \pm 1.89$ | $9.95 \pm 0.92$ | $8.85 \pm 0.45$ |
| Multiple Models + IOM | $7.21 \pm 0.57$ | $5.66 \pm 0.80$ | $4.50 \pm 0.32$ | $8.62 \pm 0.60$ | $6.54 \pm 0.65$ | $6.59 \pm 0.46$ |
| Multiple Models + ICT | $8.62 \pm 0.61$ | $9.59 \pm 0.90$ | $8.20 \pm 2.32$ | $7.62 \pm 1.43$ | $6.45 \pm 0.85$ | $8.22 \pm 0.24$ |
| Multiple Models + Tri-Mentoring | $9.54 \pm 1.16$ | $11.10 \pm 0.54$ | $8.60 \pm 1.98$ | $9.78 \pm 1.49$ | $7.34 \pm 0.55$ | $9.38 \pm 0.46$ |
| MOBO | $12.40 \pm 0.95$ | $\mathbf{4.21 \pm 0.46}$ | N/A | $12.38 \pm 1.33$ | $11.66 \pm 0.58$ | $9.34 \pm 0.43$ |
| MOBO-$q$ParEGO | $12.26 \pm 0.97$ | $13.19 \pm 0.64$ | N/A | $7.77 \pm 0.53$ | $10.55 \pm 0.37$ | $11.74 \pm 0.39$ |
| MOBO-JES | $15.25 \pm 0.53$ | N/A | N/A | N/A | $11.59 \pm 0.97$ | $13.38 \pm 0.51$ |
| ParetoFlow | $9.58 \pm 1.83$ | $12.04 \pm 0.70$ | $12.33 \pm 3.77$ | $9.88 \pm 1.17$ | $12.21 \pm 0.28$ | $11.02 \pm 1.02$ |
| DOMOO (**ours**) | $\mathbf{4.75 \pm 0.81}$ | $8.36 \pm 0.66$ | $3.10 \pm 2.46$ | $7.70 \pm 1.34$ | $3.25 \pm 0.74$ | $\mathbf{5.45 \pm 0.39}$ |

*Table 15.* Hypervolume results for synthetic functions with 256 solutions and $50^{th}$ percentile evaluations. For each task, algorithms within one standard deviation of having the highest performance are **bolded**.

| Methods | DTLZ1 | DTLZ2 | DTLZ3 | DTLZ4 | DTLZ5 | DTLZ6 | DTLZ7 | OmniTest | VLMOP1 | VLMOP2 | VLMOP3 | ZDT1 | ZDT2 | ZDT3 | ZDT4 | ZDT6 |
|---|---|---|---|---|---|---|---|---|---|---|---|---|---|---|---|---|
| $\mathcal{D}(\text{best})$ | 10.6 | 9.91 | 10 | **10.76** | **9.35** | 8.88 | 8.56 | 4.53 | 0.08 | 1.78 | 45.65 | 4.17 | 4.68 | 5.15 | **5.46** | 4.61 |
| End-to-End | $10.56 \pm 0.07$ | $7.80 \pm 0.99$ | $9.84 \pm 0.41$ | $6.66 \pm 0.97$ | $7.02 \pm 1.04$ | $8.03 \pm 1.38$ | $10.64 \pm 0.03$ | $4.77 \pm 0.01$ | $\mathbf{0.32 \pm 0.00}$ | $4.20 \pm 0.03$ | $45.90 \pm 0.04$ | $\mathbf{4.84 \pm 0.01}$ | $5.65 \pm 0.01$ | $5.07 \pm 0.75$ | $4.48 \pm 0.20$ | $\mathbf{4.77 \pm 0.00}$ |
| End-to-End + GradNorm | $10.60 \pm 0.02$ | $7.18 \pm 1.37$ | $10.37 \pm 0.29$ | $7.37 \pm 1.70$ | $6.01 \pm 1.58$ | $\mathbf{9.83 \pm 0.45}$ | $10.01 \pm 0.33$ | $3.29 \pm 0.55$ | $0.03 \pm 0.06$ | $1.44 \pm 0.00$ | $41.29 \pm 5.03$ | $4.32 \pm 0.77$ | $3.94 \pm 0.82$ | $4.01 \pm 0.98$ | $3.76 \pm 0.69$ | $2.74 \pm 0.70$ |
| End-to-End + PcGrad | $10.63 \pm 0.01$ | $8.77 \pm 1.34$ | $10.21 \pm 0.33$ | $7.91 \pm 0.86$ | $8.17 \pm 1.23$ | $9.29 \pm 0.35$ | $10.60 \pm 0.08$ | $4.77 \pm 0.01$ | $0.19 \pm 0.05$ | $4.20 \pm 0.04$ | $45.92 \pm 0.01$ | $4.83 \pm 0.03$ | $5.63 \pm 0.05$ | $5.22 \pm 0.34$ | $3.44 \pm 0.17$ | $3.87 \pm 1.03$ |
| Multi Head | $\mathbf{10.64 \pm 0.01}$ | $7.64 \pm 1.28$ | $10.23 \pm 0.31$ | $6.62 \pm 0.77$ | $7.18 \pm 0.86$ | $7.97 \pm 1.22$ | $10.47 \pm 0.20$ | $\mathbf{4.78 \pm 0.00}$ | $\mathbf{0.32 \pm 0.00}$ | $4.19 \pm 0.06$ | $\mathbf{45.93 \pm 0.01}$ | $4.79 \pm 0.04$ | $5.52 \pm 0.12$ | $5.27 \pm 0.39$ | $4.28 \pm 0.30$ | $4.76 \pm 0.00$ |
| Multi Head + GradNorm | $10.60 \pm 0.01$ | $7.36 \pm 1.72$ | $10.33 \pm 0.23$ | $7.27 \pm 1.30$ | $7.86 \pm 0.54$ | $8.52 \pm 1.05$ | $9.92 \pm 0.45$ | $3.10 \pm 0.44$ | $0.00 \pm 0.01$ | $3.32 \pm 0.49$ | $38.74 \pm 6.20$ | $4.57 \pm 0.18$ | $4.99 \pm 0.44$ | $5.28 \pm 0.45$ | $3.21 \pm 0.59$ | $3.58 \pm 1.35$ |
| Multi Head + PcGrad | $10.62 \pm 0.01$ | $8.58 \pm 0.97$ | $10.06 \pm 0.31$ | $6.09 \pm 1.37$ | $8.41 \pm 0.70$ | $9.04 \pm 0.32$ | $10.47 \pm 0.21$ | $\mathbf{4.78 \pm 0.00}$ | $0.18 \pm 0.10$ | $4.19 \pm 0.04$ | $45.93 \pm 0.00$ | $4.79 \pm 0.09$ | $5.51 \pm 0.11$ | $5.04 \pm 0.48$ | $3.87 \pm 0.43$ | $3.60 \pm 1.35$ |
| Multiple Models | $\mathbf{10.64 \pm 0.01}$ | $7.87 \pm 1.51$ | $\mathbf{10.48 \pm 0.10}$ | $7.55 \pm 1.11$ | $7.69 \pm 1.04$ | $8.30 \pm 1.45$ | $\mathbf{10.68 \pm 0.12}$ | $\mathbf{4.78 \pm 0.00}$ | $0.31 \pm 0.01$ | $\mathbf{4.22 \pm 0.01}$ | $\mathbf{45.93 \pm 0.01}$ | $4.78 \pm 0.03$ | $5.51 \pm 0.09$ | $5.43 \pm 0.16$ | $4.79 \pm 0.21$ | $4.71 \pm 0.05$ |
| Multiple Models + COMs | $10.62 \pm 0.01$ | $8.59 \pm 0.61$ | $9.80 \pm 0.18$ | $7.45 \pm 0.33$ | $8.32 \pm 0.65$ | $8.70 \pm 0.24$ | $9.55 \pm 0.09$ | $4.77 \pm 0.01$ | $0.31 \pm 0.01$ | $4.12 \pm 0.03$ | $45.92 \pm 0.02$ | $4.52 \pm 0.04$ | $5.06 \pm 0.04$ | $5.40 \pm 0.04$ | $4.52 \pm 0.16$ | $2.28 \pm 0.55$ |
| Multiple Models + RoMA | $10.59 \pm 0.01$ | $9.30 \pm 0.53$ | $9.99 \pm 0.48$ | $8.11 \pm 0.94$ | $6.91 \pm 0.23$ | $9.29 \pm 0.20$ | $10.20 \pm 0.11$ | $3.04 \pm 0.05$ | $0.15 \pm 0.01$ | $1.44 \pm 0.00$ | $37.17 \pm 2.52$ | $4.82 \pm 0.02$ | $5.36 \pm 0.07$ | $\mathbf{5.54 \pm 0.07}$ | $3.56 \pm 0.21$ | $1.74 \pm 0.17$ |
| Multiple Models + IOM | $10.62 \pm 0.00$ | $9.13 \pm 0.38$ | $9.76 \pm 0.31$ | $8.38 \pm 0.11$ | $8.52 \pm 0.86$ | $8.76 \pm 0.63$ | $10.45 \pm 0.16$ | $4.77 \pm 0.01$ | $0.26 \pm 0.06$ | $3.84 \pm 0.36$ | $45.92 \pm 0.00$ | $4.56 \pm 0.05$ | $5.51 \pm 0.07$ | $5.41 \pm 0.27$ | $4.52 \pm 0.37$ | $4.72 \pm 0.03$ |
| Multiple Models + ICT | $10.61 \pm 0.01$ | $9.21 \pm 0.54$ | $9.57 \pm 0.54$ | $8.80 \pm 1.07$ | $7.54 \pm 0.94$ | $8.43 \pm 1.17$ | $9.82 \pm 0.25$ | $4.75 \pm 0.03$ | $0.30 \pm 0.04$ | $3.89 \pm 0.25$ | $45.91 \pm 0.01$ | $4.80 \pm 0.03$ | $5.30 \pm 0.18$ | $5.31 \pm 0.18$ | $3.92 \pm 0.14$ | $3.58 \pm 1.01$ |
| Multiple Models + Tri-Mentoring | $10.37 \pm 0.47$ | $8.76 \pm 0.84$ | $9.77 \pm 0.55$ | $8.31 \pm 0.95$ | $6.14 \pm 0.24$ | $8.23 \pm 1.42$ | $9.71 \pm 0.17$ | $4.74 \pm 0.03$ | $\mathbf{0.32 \pm 0.00}$ | $3.74 \pm 0.60$ | $44.65 \pm 2.41$ | $4.76 \pm 0.01$ | $5.46 \pm 0.21$ | $4.90 \pm 0.10$ | $4.49 \pm 0.18$ | $2.34 \pm 0.25$ |
| MOBO | $10.64 \pm 0.00$ | $\mathbf{9.96 \pm 0.21}$ | $9.09 \pm 0.24$ | $8.49 \pm 0.02$ | $8.56 \pm 0.00$ | $8.75 \pm 0.07$ | $7.76 \pm 0.01$ | $4.72 \pm 0.03$ | $0.18 \pm 0.01$ | $1.44 \pm 0.00$ | N/A | $4.26 \pm 0.02$ | $4.28 \pm 0.01$ | $5.02 \pm 0.04$ | $3.86 \pm 0.02$ | $2.63 \pm 0.18$ |
| MOBO-$q$ParEGO | $10.60 \pm 0.01$ | $9.50 \pm 0.16$ | $8.22 \pm 0.54$ | $8.59 \pm 0.01$ | $7.89 \pm 0.09$ | $8.04 \pm 1.00$ | $9.26 \pm 0.08$ | $4.03 \pm 0.11$ | $0.18 \pm 0.08$ | $1.44 \pm 0.00$ | $45.79 \pm 0.00$ | $4.22 \pm 0.01$ | $4.62 \pm 0.02$ | $5.10 \pm 0.02$ | $4.36 \pm 0.01$ | $2.51 \pm 0.60$ |
| MOBO-JES | N/A | N/A | N/A | N/A | N/A | N/A | N/A | $4.30 \pm 0.05$ | $0.06 \pm 0.00$ | N/A | N/A | $3.86 \pm 0.07$ | $4.55 \pm 0.10$ | $4.93 \pm 0.07$ | $3.99 \pm 0.07$ | $1.91 \pm 0.18$ |
| ParetoFlow | $10.38 \pm 0.04$ | $9.83 \pm 0.22$ | $9.41 \pm 0.44$ | $8.64 \pm 0.80$ | $8.45 \pm 0.85$ | $9.41 \pm 0.12$ | $8.81 \pm 0.04$ | $\mathbf{4.78 \pm 0.00}$ | N/A | $4.21 \pm 0.00$ | N/A | $4.13 \pm 0.09$ | $5.36 \pm 0.25$ | $5.10 \pm 0.12$ | $4.85 \pm 0.13$ | $4.36 \pm 0.06$ |
| DOMOO (**ours**) | $\mathbf{10.64 \pm 0.01}$ | $9.75 \pm 0.25$ | $10.41 \pm 0.12$ | $6.91 \pm 1.32$ | $8.78 \pm 0.52$ | $9.09 \pm 0.93$ | $10.63 \pm 0.10$ | $4.77 \pm 0.01$ | $\mathbf{0.32 \pm 0.00}$ | $4.11 \pm 0.16$ | $45.93 \pm 0.00$ | $4.80 \pm 0.04$ | $\mathbf{5.67 \pm 0.02}$ | $5.37 \pm 0.20$ | $4.04 \pm 0.34$ | $4.71 \pm 0.04$ |

*Table 16.* Hypervolume results for MO-NAS with 256 solutions and $50^{th}$ percentile evaluations. For each task, algorithms within one standard deviation of having the highest performance are **bolded**.

| Methods | C-10/MOP1 | C-10/MOP2 | C-10/MOP3 | C-10/MOP8 | C-10/MOP9 | IN-1K/MOP1 | IN-1K/MOP2 | IN-1K/MOP3 | IN-1K/MOP4 | IN-1K/MOP5 | IN-1K/MOP6 | IN-1K/MOP7 | IN-1K/MOP8 | NasBench201-Test |
|---|---|---|---|---|---|---|---|---|---|---|---|---|---|---|
| $\mathcal{D}(\text{best})$ | 4.72 | 10.42 | 9.21 | 4.38 | 9.64 | 4.36 | 4.45 | 9.86 | 4.15 | 4.3 | 9.15 | 3.7 | 9.13 | 9.89 |
| End-to-End | $4.68 \pm 0.04$ | $10.41 \pm 0.02$ | $9.99 \pm 0.06$ | $4.42 \pm 0.15$ | $9.78 \pm 0.14$ | $4.43 \pm 0.17$ | $4.48 \pm 0.06$ | $9.89 \pm 0.04$ | $\mathbf{4.44 \pm 0.10}$ | $4.50 \pm 0.06$ | $9.57 \pm 0.17$ | $3.63 \pm 0.28$ | $9.24 \pm 0.10$ | $9.88 \pm 0.19$ |
| End-to-End + GradNorm | $4.44 \pm 0.19$ | $10.37 \pm 0.06$ | $9.01 \pm 0.11$ | $3.62 \pm 0.31$ | $8.80 \pm 0.27$ | $4.09 \pm 0.23$ | $4.32 \pm 0.07$ | $8.29 \pm 0.32$ | $3.82 \pm 0.31$ | $4.25 \pm 0.30$ | $7.43 \pm 1.44$ | $3.54 \pm 0.44$ | $8.07 \pm 0.10$ | $8.68 \pm 1.49$ |
| End-to-End + PcGrad | $4.69 \pm 0.03$ | $10.42 \pm 0.01$ | $9.95 \pm 0.07$ | $4.21 \pm 0.13$ | $9.92 \pm 0.20$ | $4.40 \pm 0.05$ | $4.39 \pm 0.12$ | $9.92 \pm 0.09$ | $4.15 \pm 0.13$ | $4.37 \pm 0.08$ | $9.34 \pm 0.13$ | $3.79 \pm 0.11$ | $9.29 \pm 0.17$ | $9.51 \pm 0.39$ |
| Multi Head | $4.71 \pm 0.01$ | $10.34 \pm 0.11$ | $9.86 \pm 0.03$ | $4.39 \pm 0.10$ | $9.47 \pm 0.28$ | $4.53 \pm 0.07$ | $4.37 \pm 0.09$ | $9.98 \pm 0.04$ | $4.41 \pm 0.06$ | $\mathbf{4.52 \pm 0.09}$ | $\mathbf{9.58 \pm 0.23}$ | $3.94 \pm 0.33$ | $9.38 \pm 0.11$ | $9.73 \pm 0.37$ |
| Multi Head + GradNorm | $3.51 \pm 1.78$ | $10.13 \pm 0.23$ | $8.60 \pm 0.33$ | $3.65 \pm 0.19$ | $7.64 \pm 1.26$ | $3.97 \pm 0.52$ | $3.63 \pm 0.47$ | $8.20 \pm 1.33$ | $4.25 \pm 0.11$ | $4.32 \pm 0.14$ | $9.10 \pm 0.28$ | $2.65 \pm 0.59$ | $5.59 \pm 1.85$ | $9.52 \pm 0.10$ |
| Multi Head + PcGrad | $4.68 \pm 0.05$ | $\mathbf{10.43 \pm 0.01}$ | $9.59 \pm 0.31$ | $4.03 \pm 0.17$ | $9.78 \pm 0.17$ | $4.39 \pm 0.02$ | $4.48 \pm 0.04$ | $9.98 \pm 0.02$ | $4.15 \pm 0.03$ | $4.43 \pm 0.05$ | $9.32 \pm 0.11$ | $3.81 \pm 0.07$ | $9.05 \pm 0.29$ | $9.86 \pm 0.27$ |
| Multiple Models | $4.68 \pm 0.06$ | $10.08 \pm 0.42$ | $9.65 \pm 0.40$ | $4.47 \pm 0.05$ | $9.54 \pm 0.15$ | $4.31 \pm 0.32$ | $4.47 \pm 0.03$ | $9.91 \pm 0.08$ | $4.37 \pm 0.05$ | $4.49 \pm 0.04$ | $9.44 \pm 0.31$ | $3.96 \pm 0.25$ | $9.35 \pm 0.09$ | $9.76 \pm 0.44$ |
| Multiple Models + COMs | $\mathbf{4.73 \pm 0.01}$ | $10.41 \pm 0.02$ | $9.85 \pm 0.06$ | $4.24 \pm 0.06$ | $9.38 \pm 0.21$ | $4.52 \pm 0.04$ | $4.51 \pm 0.02$ | $9.93 \pm 0.07$ | $4.24 \pm 0.10$ | $4.39 \pm 0.06$ | $9.37 \pm 0.17$ | $3.75 \pm 0.12$ | $9.28 \pm 0.21$ | $9.80 \pm 0.33$ |
| Multiple Models + RoMA | $4.70 \pm 0.04$ | $\mathbf{10.43 \pm 0.01}$ | $9.91 \pm 0.09$ | $4.11 \pm 0.09$ | $9.09 \pm 0.21$ | $4.49 \pm 0.05$ | $4.49 \pm 0.03$ | $9.82 \pm 0.03$ | $4.41 \pm 0.07$ | $4.50 \pm 0.05$ | $9.50 \pm 0.13$ | $4.10 \pm 0.12$ | $9.18 \pm 0.11$ | $9.69 \pm 0.32$ |
| Multiple Models + IOM | $4.69 \pm 0.04$ | $10.34 \pm 0.05$ | $9.94 \pm 0.02$ | $4.51 \pm 0.06$ | $9.88 \pm 0.11$ | $4.51 \pm 0.06$ | $\mathbf{4.56 \pm 0.04}$ | $\mathbf{10.02 \pm 0.01}$ | $4.25 \pm 0.05$ | $4.44 \pm 0.05$ | $9.35 \pm 0.08$ | $3.74 \pm 0.12$ | $9.54 \pm 0.06$ | $9.92 \pm 0.15$ |
| Multiple Models + ICT | $4.69 \pm 0.02$ | $10.17 \pm 0.51$ | $9.61 \pm 0.26$ | $4.00 \pm 0.25$ | $8.94 \pm 0.32$ | $4.28 \pm 0.08$ | $4.31 \pm 0.07$ | $9.60 \pm 0.22$ | $4.36 \pm 0.11$ | $4.36 \pm 0.04$ | $9.49 \pm 0.07$ | $3.77 \pm 0.18$ | $8.72 \pm 0.55$ | $9.85 \pm 0.19$ |
| Multiple Models + Tri-Mentoring | $4.67 \pm 0.03$ | $10.35 \pm 0.10$ | $9.95 \pm 0.05$ | $3.88 \pm 0.18$ | $8.06 \pm 0.25$ | $4.29 \pm 0.13$ | $4.19 \pm 0.10$ | $9.15 \pm 0.30$ | $4.15 \pm 0.04$ | $4.29 \pm 0.04$ | $9.29 \pm 0.12$ | $3.75 \pm 0.20$ | $9.12 \pm 0.30$ | $8.86 \pm 0.14$ |
| MOBO | $4.71 \pm 0.03$ | $\mathbf{10.43 \pm 0.01}$ | $\mathbf{10.07 \pm 0.01}$ | $4.52 \pm 0.02$ | $\mathbf{10.02 \pm 0.04}$ | $\mathbf{4.58 \pm 0.02}$ | $4.51 \pm 0.01$ | $9.99 \pm 0.04$ | $4.17 \pm 0.05$ | $4.47 \pm 0.02$ | $9.15 \pm 0.03$ | $4.07 \pm 0.02$ | $\mathbf{9.56 \pm 0.01}$ | N/A |
| MOBO-$q$ParEGO | $4.70 \pm 0.01$ | $10.30 \pm 0.06$ | $8.46 \pm 0.11$ | $4.02 \pm 0.03$ | $9.43 \pm 0.22$ | $3.90 \pm 0.07$ | $3.64 \pm 0.05$ | $9.14 \pm 0.10$ | $3.98 \pm 0.12$ | $4.14 \pm 0.36$ | $9.14 \pm 0.12$ | $3.67 \pm 0.03$ | $8.66 \pm 0.16$ | N/A |
| MOBO-JES | N/A | N/A | N/A | N/A | N/A | N/A | N/A | N/A | N/A | N/A | N/A | N/A | N/A | N/A |
| ParetoFlow | $4.66 \pm 0.09$ | $10.41 \pm 0.00$ | $9.15 \pm 0.17$ | $4.02 \pm 0.21$ | $9.25 \pm 0.00$ | $4.18 \pm 0.01$ | $4.20 \pm 0.10$ | $9.41 \pm 0.16$ | $4.17 \pm 0.02$ | N/A | N/A | $3.40 \pm 0.22$ | $9.01 \pm 0.00$ | N/A |
| DOMOO (**ours**) | $4.67 \pm 0.04$ | $10.37 \pm 0.02$ | $9.83 \pm 0.13$ | $\mathbf{4.53 \pm 0.06}$ | $9.72 \pm 0.08$ | $4.42 \pm 0.17$ | $4.48 \pm 0.05$ | $9.29 \pm 0.42$ | $3.14 \pm 0.43$ | $2.82 \pm 0.45$ | $7.16 \pm 0.21$ | $\mathbf{4.29 \pm 0.06}$ | $9.29 \pm 0.20$ | $\mathbf{10.09 \pm 0.05}$ |

*Table 17.* Hypervolume results for MORL with 256 solutions and $50^{th}$ percentile evaluations. For each task, algorithms within one standard deviation of having the highest performance are **bolded**.

| Methods | MO-Swimmer | MO-Hopper |
|---|---|---|
| $\mathcal{D}$(best) | **3.64** | **5.67** |
| End-to-End | $2.57 \pm 0.00$ | $4.80 \pm 0.00$ |
| End-to-End + GradNorm | $2.45 \pm 0.00$ | $4.78 \pm 0.00$ |
| End-to-End + PcGrad | $2.52 \pm 0.00$ | $4.98 \pm 0.00$ |
| Multi Head | $2.73 \pm 0.00$ | $4.93 \pm 0.00$ |
| Multi Head + GradNorm | $2.47 \pm 0.00$ | $4.76 \pm 0.00$ |
| Multi Head + PcGrad | $2.51 \pm 0.00$ | $4.92 \pm 0.00$ |
| Multiple Models | $2.54 \pm 0.00$ | $4.81 \pm 0.00$ |
| Multiple Models + COMs | $2.87 \pm 0.00$ | $4.86 \pm 0.00$ |
| Multiple Models + RoMA | $2.57 \pm 0.00$ | $4.85 \pm 0.00$ |
| Multiple Models + IOM | $2.62 \pm 0.00$ | $5.15 \pm 0.00$ |
| Multiple Models + ICT | $2.68 \pm 0.18$ | $4.93 \pm 0.27$ |
| Multiple Models + Tri-Mentoring | $2.67 \pm 0.14$ | $4.79 \pm 0.01$ |
| MOBO | N/A | N/A |
| MOBO-$q$ParEGO | N/A | N/A |
| MOBO-JES | N/A | N/A |
| ParetoFlow | $2.18 \pm 0.29$ | $4.71 \pm 0.00$ |
| DOMOO **(ours)** | $3.12 \pm 0.05$ | $5.35 \pm 0.44$ |

*Table 18.* Hypervolume results for RE with 256 solutions and $50^{th}$ percentile evaluations. For each task, algorithms within one standard deviation of having the highest performance are **bolded**.

| Methods | RE21 | RE22 | RE23 | RE24 | RE25 | RE31 | RE32 | RE33 | RE34 | RE35 | RE36 | RE37 | MO-Portfolio |
|---|---|---|---|---|---|---|---|---|---|---|---|---|---|
| $\mathcal{D}$(best) | 4.1 | 4.78 | 4.75 | 4.6 | 4.79 | 10.6 | 10.56 | 10.56 | 9.3 | 10.08 | 7.61 | 5.57 | 4.24 |
| End-to-End | $4.59 \pm 0.00$ | $\mathbf{4.84 \pm 0.00}$ | $\mathbf{4.84 \pm 0.01}$ | $4.65 \pm 0.22$ | $4.78 \pm 0.08$ | $10.55 \pm 0.20$ | $\mathbf{10.65 \pm 0.00}$ | $10.52 \pm 0.16$ | $10.07 \pm 0.01$ | $10.37 \pm 0.04$ | $9.73 \pm 0.23$ | $6.55 \pm 0.08$ | $4.40 \pm 0.03$ |
| End-to-End + GradNorm | $4.54 \pm 0.04$ | $4.81 \pm 0.04$ | $4.05 \pm 0.75$ | $3.19 \pm 0.82$ | $4.74 \pm 0.07$ | $8.52 \pm 4.26$ | $10.62 \pm 0.03$ | $10.27 \pm 0.08$ | $9.41 \pm 0.25$ | $10.34 \pm 0.01$ | $0.02 \pm 0.00$ | $6.53 \pm 0.03$ | $4.17 \pm 0.14$ |
| End-to-End + PcGrad | $4.59 \pm 0.00$ | $4.08 \pm 1.52$ | $4.83 \pm 0.02$ | $4.16 \pm 0.17$ | $4.81 \pm 0.01$ | $10.64 \pm 0.00$ | $10.61 \pm 0.02$ | $10.43 \pm 0.13$ | $10.04 \pm 0.04$ | $10.54 \pm 0.02$ | $9.68 \pm 0.17$ | $6.60 \pm 0.05$ | $4.41 \pm 0.05$ |
| Multi Head | $4.59 \pm 0.00$ | $\mathbf{4.84 \pm 0.00}$ | $4.84 \pm 0.00$ | $4.57 \pm 0.53$ | $4.80 \pm 0.06$ | $10.58 \pm 0.13$ | $10.64 \pm 0.01$ | $10.58 \pm 0.06$ | $10.02 \pm 0.05$ | $10.32 \pm 0.20$ | $9.76 \pm 0.17$ | $6.63 \pm 0.05$ | $4.33 \pm 0.08$ |
| Multi Head + GradNorm | $4.28 \pm 0.42$ | $2.07 \pm 1.86$ | $3.08 \pm 0.75$ | $3.10 \pm 0.43$ | $3.95 \pm 0.86$ | $10.07 \pm 0.49$ | $10.47 \pm 0.24$ | $8.77 \pm 1.49$ | $9.69 \pm 0.63$ | $10.19 \pm 0.47$ | $6.53 \pm 3.42$ | $6.05 \pm 0.96$ | $4.10 \pm 0.21$ |
| Multi Head + PcGrad | $4.52 \pm 0.08$ | $3.02 \pm 1.54$ | $4.83 \pm 0.01$ | $2.75 \pm 0.17$ | $4.69 \pm 0.17$ | $10.55 \pm 0.20$ | $9.00 \pm 1.58$ | $10.13 \pm 0.68$ | $10.04 \pm 0.03$ | $10.51 \pm 0.04$ | $9.48 \pm 0.18$ | $6.62 \pm 0.07$ | $4.27 \pm 0.07$ |
| Multiple Models | $4.59 \pm 0.01$ | $4.76 \pm 0.15$ | $4.77 \pm 0.10$ | $4.82 \pm 0.03$ | $4.64 \pm 0.24$ | $10.64 \pm 0.01$ | $10.58 \pm 0.03$ | $10.62 \pm 0.01$ | $10.08 \pm 0.02$ | $\mathbf{10.55 \pm 0.01}$ | $\mathbf{9.83 \pm 0.20}$ | $6.67 \pm 0.02$ | $4.53 \pm 0.29$ |
| Multiple Models + COMs | $4.35 \pm 0.05$ | $4.78 \pm 0.11$ | $4.81 \pm 0.02$ | $4.41 \pm 0.62$ | $4.73 \pm 0.10$ | $10.62 \pm 0.02$ | $10.63 \pm 0.01$ | $10.19 \pm 0.81$ | $9.84 \pm 0.19$ | $10.45 \pm 0.06$ | $8.83 \pm 0.27$ | $6.28 \pm 0.08$ | $3.55 \pm 0.63$ |
| Multiple Models + RoMA | $4.54 \pm 0.01$ | $4.56 \pm 0.49$ | $4.42 \pm 0.81$ | $3.32 \pm 0.89$ | $4.73 \pm 0.17$ | $10.57 \pm 0.08$ | $10.64 \pm 0.00$ | $10.34 \pm 0.18$ | $9.28 \pm 0.05$ | $10.51 \pm 0.04$ | $7.57 \pm 0.77$ | $6.57 \pm 0.08$ | $4.24 \pm 0.08$ |
| Multiple Models + IOM | $4.58 \pm 0.01$ | $4.82 \pm 0.04$ | $4.80 \pm 0.03$ | $4.80 \pm 0.02$ | $\mathbf{4.83 \pm 0.01}$ | $10.63 \pm 0.02$ | $10.64 \pm 0.01$ | $10.58 \pm 0.03$ | $10.03 \pm 0.01$ | $10.51 \pm 0.04$ | $9.53 \pm 0.16$ | $6.56 \pm 0.08$ | $4.44 \pm 0.29$ |
| Multiple Models + ICT | $4.58 \pm 0.01$ | $4.75 \pm 0.19$ | $4.72 \pm 0.17$ | $4.56 \pm 0.20$ | $4.82 \pm 0.03$ | $\mathbf{10.65 \pm 0.00}$ | $10.63 \pm 0.01$ | $10.58 \pm 0.02$ | $10.03 \pm 0.02$ | $10.42 \pm 0.22$ | $9.71 \pm 0.17$ | $6.65 \pm 0.05$ | $4.50 \pm 0.22$ |
| Multiple Models + Tri-Mentoring | $4.59 \pm 0.00$ | $\mathbf{4.84 \pm 0.00}$ | $4.34 \pm 0.41$ | $4.74 \pm 0.18$ | $4.76 \pm 0.11$ | $\mathbf{10.65 \pm 0.00}$ | $10.63 \pm 0.01$ | $10.58 \pm 0.05$ | $9.97 \pm 0.05$ | $10.44 \pm 0.25$ | $7.13 \pm 1.75$ | $6.64 \pm 0.03$ | $4.39 \pm 0.11$ |
| MOBO | $3.99 \pm 0.06$ | $\mathbf{4.84 \pm 0.00}$ | $4.18 \pm 0.01$ | $3.35 \pm 0.09$ | $\mathbf{4.83 \pm 0.01}$ | $9.61 \pm 0.00$ | $10.64 \pm 0.00$ | $10.36 \pm 0.07$ | $7.27 \pm 0.19$ | $10.32 \pm 0.09$ | $8.51 \pm 0.00$ | $6.47 \pm 0.00$ | $0.39 \pm 0.48$ |
| MOBO-$q$ParEGO | $4.14 \pm 0.11$ | $\mathbf{4.84 \pm 0.00}$ | $4.71 \pm 0.15$ | $3.20 \pm 0.21$ | $4.83 \pm 0.00$ | $10.63 \pm 0.00$ | $10.64 \pm 0.00$ | $10.52 \pm 0.07$ | $7.28 \pm 0.16$ | $10.28 \pm 0.03$ | $8.19 \pm 0.00$ | $6.22 \pm 0.21$ | $1.82 \pm 2.23$ |
| MOBO-JES | $4.33 \pm 0.08$ | $\mathbf{4.84 \pm 0.00}$ | $4.75 \pm 0.00$ | $4.59 \pm 0.00$ | $4.81 \pm 0.01$ | N/A | N/A | $10.34 \pm 0.24$ | $9.06 \pm 0.00$ | $10.44 \pm 0.00$ | N/A | N/A | $0.00 \pm 0.00$ |
| ParetoFlow | $4.23 \pm 0.12$ | $4.63 \pm 0.04$ | N/A | N/A | N/A | $10.16 \pm 0.16$ | $10.59 \pm 0.00$ | $\mathbf{10.72 \pm 0.17}$ | $9.30 \pm 0.12$ | N/A | $7.52 \pm 0.19$ | $6.12 \pm 0.45$ | $4.03 \pm 0.07$ |
| DOMOO **(ours)** | $\mathbf{4.60 \pm 0.00}$ | $\mathbf{4.84 \pm 0.00}$ | $4.84 \pm 0.00$ | $\mathbf{4.83 \pm 0.01}$ | $4.64 \pm 0.24$ | $\mathbf{10.65 \pm 0.00}$ | $10.64 \pm 0.01$ | $10.62 \pm 0.00$ | $\mathbf{10.10 \pm 0.01}$ | $10.54 \pm 0.08$ | $9.72 \pm 0.18$ | $\mathbf{6.72 \pm 0.00}$ | $\mathbf{5.55 \pm 0.52}$ |

*Table 19.* Hypervolume results for scientific design with 256 solutions and $50^{th}$ percentile evaluations. For each task, algorithms within one standard deviation of having the highest performance are **bolded**.

| Methods | Molecule | Regex | RFP | ZINC |
|---|---|---|---|---|
| $\mathcal{D}$(best) | **2.91** | 3.96 | 4.06 | 4.52 |
| End-to-End | $1.67 \pm 1.00$ | $2.99 \pm 0.00$ | $4.02 \pm 0.02$ | $4.45 \pm 0.04$ |
| End-to-End + GradNorm | $0.00 \pm 0.00$ | $2.99 \pm 0.00$ | $3.99 \pm 0.02$ | $4.36 \pm 0.07$ |
| End-to-End + PcGrad | $2.03 \pm 0.63$ | $2.99 \pm 0.00$ | $4.02 \pm 0.07$ | $4.37 \pm 0.06$ |
| Multi Head | $0.61 \pm 0.75$ | $2.99 \pm 0.00$ | $4.05 \pm 0.04$ | $4.46 \pm 0.04$ |
| Multi Head + GradNorm | $2.10 \pm 0.58$ | $3.67 \pm 0.34$ | $\mathbf{4.06 \pm 0.02}$ | $4.27 \pm 0.05$ |
| Multi Head + PcGrad | $1.70 \pm 0.37$ | $2.99 \pm 0.00$ | $3.92 \pm 0.15$ | $4.40 \pm 0.04$ |
| Multiple Models | $1.99 \pm 0.59$ | $2.99 \pm 0.00$ | $4.02 \pm 0.03$ | $4.42 \pm 0.02$ |
| Multiple Models + COMs | $2.53 \pm 0.52$ | $3.16 \pm 0.34$ | $4.03 \pm 0.03$ | $4.33 \pm 0.10$ |
| Multiple Models + RoMA | $1.96 \pm 0.61$ | $2.99 \pm 0.00$ | $4.03 \pm 0.02$ | $4.33 \pm 0.06$ |
| Multiple Models + IOM | $2.32 \pm 0.43$ | $2.99 \pm 0.00$ | $4.04 \pm 0.04$ | $4.33 \pm 0.08$ |
| Multiple Models + ICT | $2.53 \pm 0.54$ | $3.25 \pm 0.52$ | $3.98 \pm 0.05$ | $4.40 \pm 0.04$ |
| Multiple Models + Tri-Mentoring | $1.54 \pm 0.05$ | $2.99 \pm 0.00$ | $4.03 \pm 0.08$ | $4.31 \pm 0.11$ |
| MOBO | $0.00 \pm 0.00$ | $4.54 \pm 0.11$ | $3.98 \pm 0.01$ | $4.34 \pm 0.01$ |
| MOBO-$q$ParEGO | N/A | $4.75 \pm 0.19$ | $3.67 \pm 0.03$ | $\mathbf{4.57 \pm 0.09}$ |
| MOBO-JES | N/A | N/A | N/A | N/A |
| ParetoFlow | $1.58 \pm 0.05$ | $3.26 \pm 0.00$ | N/A | $4.05 \pm 0.25$ |
| DOMOO **(ours)** | $1.74 \pm 0.36$ | $\mathbf{4.78 \pm 0.26}$ | $3.95 \pm 0.06$ | $4.46 \pm 0.06$ |

# H. IGD$_{\text{offline}}$ Experiment Results

## H.1. The $100^{th}$ Percentile Results

As shown in Table 20, Table 21, Table 22, Table 23, and Table 24, we report the $100^{th}$ percentile IGD$_{\text{offline}}$ results with 256 solutions. DOMOO consistently performs well across tasks. Methods within one standard deviation of the best are highlighted in **bold**.

*Table 20.* IGD$_{\text{offline}}$ results for synthetic functions with 256 solutions and $100^{th}$ percentile evaluations. For each task, algorithms within one standard deviation of having the highest performance are **bolded**.

| Methods | DTLZ1 | DTLZ2 | DTLZ3 | DTLZ4 | DTLZ5 | DTLZ6 | DTLZ7 | OmniTest | VLMOP1 | VLMOP2 | VLMOP3 | ZDT1 | ZDT2 | ZDT3 | ZDT4 | ZDT6 |
|---|---|---|---|---|---|---|---|---|---|---|---|---|---|---|---|---|
| $\mathcal{D}$(best) | 0.25 | **0.27** | 0.23 | 0.01 | **0.35** | 0.43 | 0.61 | 0.46 | 0.06 | 1.34 | 0.08 | 0.48 | 0.45 | 0.55 | **0.07** | 0.14 |
| End-to-End | $0.20 \pm 0.03$ | $0.75 \pm 0.08$ | $0.21 \pm 0.05$ | $0.87 \pm 0.10$ | $0.85 \pm 0.03$ | $0.58 \pm 0.11$ | $0.29 \pm 0.01$ | $0.22 \pm 0.00$ | $0.11 \pm 0.08$ | $0.91 \pm 0.00$ | $0.08 \pm 0.05$ | $\mathbf{0.14 \pm 0.00}$ | $0.21 \pm 0.00$ | $0.38 \pm 0.04$ | $0.38 \pm 0.07$ | $\mathbf{0.11 \pm 0.00}$ |
| End-to-End + GradNorm | $0.17 \pm 0.00$ | $0.85 \pm 0.05$ | $0.22 \pm 0.05$ | $0.87 \pm 0.22$ | $0.98 \pm 0.03$ | $\mathbf{0.42 \pm 0.05}$ | $0.29 \pm 0.01$ | $0.29 \pm 0.08$ | $0.05 \pm 0.02$ | $1.24 \pm 0.26$ | $0.34 \pm 0.21$ | $0.29 \pm 0.03$ | $0.27 \pm 0.02$ | $0.47 \pm 0.02$ | $0.48 \pm 0.27$ | $0.18 \pm 0.04$ |
| End-to-End + PcGrad | $0.17 \pm 0.01$ | $0.58 \pm 0.08$ | $0.18 \pm 0.01$ | $0.66 \pm 0.14$ | $0.62 \pm 0.06$ | $0.54 \pm 0.05$ | $0.28 \pm 0.01$ | $0.22 \pm 0.00$ | $\mathbf{0.03 \pm 0.00}$ | $0.91 \pm 0.00$ | $0.07 \pm 0.01$ | $\mathbf{0.14 \pm 0.00}$ | $0.21 \pm 0.00$ | $0.37 \pm 0.03$ | $0.85 \pm 0.12$ | $0.41 \pm 0.36$ |
| Multi Head | $0.17 \pm 0.00$ | $0.74 \pm 0.09$ | $0.18 \pm 0.03$ | $0.96 \pm 0.11$ | $0.77 \pm 0.03$ | $0.61 \pm 0.10$ | $0.31 \pm 0.03$ | $0.22 \pm 0.00$ | $0.09 \pm 0.05$ | $0.91 \pm 0.00$ | $\mathbf{0.05 \pm 0.00}$ | $0.16 \pm 0.02$ | $0.24 \pm 0.05$ | $0.32 \pm 0.03$ | $0.48 \pm 0.12$ | $\mathbf{0.11 \pm 0.00}$ |
| Multi Head + GradNorm | $0.17 \pm 0.00$ | $0.78 \pm 0.17$ | $0.18 \pm 0.06$ | $0.94 \pm 0.04$ | $0.86 \pm 0.11$ | $0.52 \pm 0.07$ | $0.41 \pm 0.04$ | $0.46 \pm 0.25$ | $0.82 \pm 0.24$ | $0.91 \pm 0.00$ | $0.19 \pm 0.18$ | $0.29 \pm 0.09$ | $0.29 \pm 0.06$ | $0.32 \pm 0.05$ | $0.79 \pm 0.14$ | $0.43 \pm 0.36$ |
| Multi Head + PcGrad | $0.17 \pm 0.00$ | $0.57 \pm 0.05$ | $0.18 \pm 0.02$ | $1.02 \pm 0.08$ | $0.60 \pm 0.04$ | $0.52 \pm 0.03$ | $0.31 \pm 0.03$ | $0.22 \pm 0.00$ | $0.11 \pm 0.13$ | $0.91 \pm 0.00$ | $0.06 \pm 0.01$ | $0.18 \pm 0.07$ | $0.23 \pm 0.05$ | $0.36 \pm 0.02$ | $0.66 \pm 0.24$ | $0.48 \pm 0.44$ |
| Multiple Models | $0.17 \pm 0.00$ | $0.70 \pm 0.01$ | $0.16 \pm 0.03$ | $0.76 \pm 0.01$ | $0.76 \pm 0.08$ | $0.62 \pm 0.11$ | $0.29 \pm 0.03$ | $0.22 \pm 0.00$ | $0.09 \pm 0.09$ | $0.91 \pm 0.00$ | $\mathbf{0.05 \pm 0.00}$ | $0.16 \pm 0.01$ | $\mathbf{0.21 \pm 0.01}$ | $0.35 \pm 0.06$ | $0.22 \pm 0.10$ | $0.13 \pm 0.03$ |
| Multiple Models + COMs | $0.17 \pm 0.00$ | $0.44 \pm 0.04$ | $0.21 \pm 0.02$ | $0.67 \pm 0.04$ | $0.51 \pm 0.04$ | $0.59 \pm 0.02$ | $0.35 \pm 0.02$ | $0.22 \pm 0.00$ | $0.11 \pm 0.11$ | $0.91 \pm 0.00$ | $0.16 \pm 0.09$ | $0.23 \pm 0.02$ | $0.29 \pm 0.02$ | $0.38 \pm 0.01$ | $0.32 \pm 0.04$ | $0.90 \pm 0.19$ |
| Multiple Models + RoMA | $0.18 \pm 0.00$ | $0.66 \pm 0.01$ | $0.21 \pm 0.04$ | $0.95 \pm 0.02$ | $0.90 \pm 0.04$ | $0.48 \pm 0.01$ | $0.28 \pm 0.00$ | $0.47 \pm 0.10$ | $\mathbf{0.03 \pm 0.00}$ | $0.91 \pm 0.00$ | $0.19 \pm 0.15$ | $\mathbf{0.14 \pm 0.00}$ | $0.21 \pm 0.00$ | $\mathbf{0.22 \pm 0.00}$ | $0.75 \pm 0.09$ | $1.08 \pm 0.17$ |
| Multiple Models + IOM | $0.18 \pm 0.00$ | $0.38 \pm 0.04$ | $0.18 \pm 0.02$ | $0.48 \pm 0.01$ | $0.42 \pm 0.04$ | $0.56 \pm 0.02$ | $0.27 \pm 0.01$ | $0.22 \pm 0.00$ | $0.12 \pm 0.09$ | $0.91 \pm 0.00$ | $0.18 \pm 0.04$ | $0.23 \pm 0.04$ | $\mathbf{0.21 \pm 0.01}$ | $0.29 \pm 0.01$ | $0.30 \pm 0.08$ | $0.12 \pm 0.00$ |
| Multiple Models + ICT | $0.17 \pm 0.01$ | $0.50 \pm 0.04$ | $0.22 \pm 0.04$ | $0.70 \pm 0.08$ | $0.61 \pm 0.10$ | $0.59 \pm 0.06$ | $0.31 \pm 0.00$ | $0.23 \pm 0.00$ | $0.07 \pm 0.02$ | $0.91 \pm 0.00$ | $0.09 \pm 0.05$ | $0.16 \pm 0.01$ | $0.23 \pm 0.01$ | $0.43 \pm 0.06$ | $0.59 \pm 0.06$ | $0.32 \pm 0.18$ |
| Multiple Models + Tri-Mentoring | $0.21 \pm 0.06$ | $0.65 \pm 0.08$ | $0.20 \pm 0.03$ | $0.72 \pm 0.13$ | $0.74 \pm 0.11$ | $0.57 \pm 0.07$ | $0.35 \pm 0.02$ | $0.23 \pm 0.00$ | $0.06 \pm 0.02$ | $0.91 \pm 0.01$ | $0.06 \pm 0.00$ | $0.19 \pm 0.01$ | $0.24 \pm 0.05$ | $0.48 \pm 0.06$ | $0.33 \pm 0.05$ | $0.57 \pm 0.35$ |
| MOBO | $\mathbf{0.16 \pm 0.00}$ | $0.31 \pm 0.00$ | $0.20 \pm 0.00$ | $0.42 \pm 0.00$ | $0.39 \pm 0.00$ | $0.56 \pm 0.01$ | $0.29 \pm 0.01$ | $0.22 \pm 0.00$ | $\mathbf{0.03 \pm 0.00}$ | $0.99 \pm 0.02$ | N/A | $0.33 \pm 0.00$ | $0.32 \pm 0.00$ | $0.41 \pm 0.01$ | $0.52 \pm 0.03$ | $0.81 \pm 0.00$ |
| MOBO-$q$ParEGO | $0.17 \pm 0.00$ | $0.31 \pm 0.00$ | $0.19 \pm 0.00$ | $0.43 \pm 0.01$ | $0.38 \pm 0.00$ | $0.59 \pm 0.02$ | $0.29 \pm 0.00$ | $0.22 \pm 0.00$ | $\mathbf{0.03 \pm 0.00}$ | $0.95 \pm 0.02$ | $0.12 \pm 0.00$ | $0.33 \pm 0.02$ | $0.31 \pm 0.02$ | $0.42 \pm 0.00$ | $0.33 \pm 0.03$ | $0.75 \pm 0.02$ |
| MOBO-JES | N/A | N/A | N/A | N/A | N/A | N/A | N/A | $0.23 \pm 0.00$ | $0.08 \pm 0.00$ | N/A | N/A | $0.51 \pm 0.02$ | $0.39 \pm 0.02$ | $0.56 \pm 0.01$ | $0.56 \pm 0.02$ | $0.95 \pm 0.00$ |
| ParetoFlow | $\mathbf{0.11 \pm 0.02}$ | $0.39 \pm 0.10$ | $0.20 \pm 0.04$ | $\mathbf{0.00 \pm 0.00}$ | $0.48 \pm 0.03$ | $0.77 \pm 0.06$ | $0.52 \pm 0.04$ | $\mathbf{0.19 \pm 0.00}$ | N/A | $\mathbf{0.84 \pm 0.00}$ | N/A | $0.46 \pm 0.02$ | $0.38 \pm 0.03$ | $0.53 \pm 0.01$ | $\mathbf{0.05 \pm 0.03}$ | $0.09 \pm 0.08$ |
| DOMOO **(ours)** | $0.16 \pm 0.00$ | $0.40 \pm 0.03$ | $\mathbf{0.14 \pm 0.01}$ | $0.77 \pm 0.01$ | $0.46 \pm 0.02$ | $0.51 \pm 0.09$ | $\mathbf{0.26 \pm 0.01}$ | $0.22 \pm 0.00$ | $\mathbf{0.03 \pm 0.00}$ | $0.91 \pm 0.00$ | $\mathbf{0.05 \pm 0.00}$ | $0.15 \pm 0.01$ | $0.21 \pm 0.00$ | $0.33 \pm 0.03$ | $0.21 \pm 0.07$ | $0.12 \pm 0.01$ |

*Table 21.* IGD$_{\text{offline}}$ results for MO-NAS with 256 solutions and $100^{th}$ percentile evaluations. For each task, algorithms within one standard deviation of having the highest performance are **bolded**.

| Methods | C-10/MOP1 | C-10/MOP2 | C-10/MOP3 | C-10/MOP8 | C-10/MOP9 | IN-1K/MOP1 | IN-1K/MOP2 | IN-1K/MOP3 | IN-1K/MOP4 | IN-1K/MOP5 | IN-1K/MOP6 | IN-1K/MOP7 | IN-1K/MOP8 | NasBench201-Test |
|---|---|---|---|---|---|---|---|---|---|---|---|---|---|---|
| $\mathcal{D}$(best) | 0.11 | 0.1 | 0.34 | 0.36 | 0.33 | 0.34 | 0.32 | 0.37 | 0.35 | 0.29 | 0.37 | 0.64 | 0.62 | 0.32 |
| End-to-End | 0.10 ± 0.00 | 0.08 ± 0.00 | 0.29 ± 0.00 | 0.25 ± 0.01 | 0.25 ± 0.02 | 0.26 ± 0.01 | **0.28 ± 0.00** | **0.33 ± 0.00** | **0.24 ± 0.01** | 0.22 ± 0.01 | **0.31 ± 0.03** | 0.43 ± 0.08 | 0.55 ± 0.01 | **0.25 ± 0.00** |
| End-to-End + GradNorm | 0.15 ± 0.02 | 0.10 ± 0.00 | 0.40 ± 0.02 | 0.39 ± 0.06 | 0.27 ± 0.01 | 0.32 ± 0.01 | 0.31 ± 0.01 | 0.48 ± 0.02 | 0.28 ± 0.04 | 0.24 ± 0.02 | 0.39 ± 0.05 | 0.42 ± 0.04 | 0.63 ± 0.01 | 0.49 ± 0.34 |
| End-to-End + PcGrad | 0.11 ± 0.00 | 0.08 ± 0.01 | 0.29 ± 0.00 | 0.27 ± 0.01 | 0.25 ± 0.02 | 0.28 ± 0.01 | 0.29 ± 0.00 | **0.33 ± 0.00** | 0.30 ± 0.03 | 0.23 ± 0.02 | 0.33 ± 0.02 | 0.41 ± 0.04 | 0.55 ± 0.00 | 0.26 ± 0.01 |
| Multi Head | 0.10 ± 0.00 | 0.08 ± 0.00 | **0.28 ± 0.00** | 0.26 ± 0.01 | 0.27 ± 0.02 | **0.24 ± 0.01** | 0.29 ± 0.00 | **0.33 ± 0.00** | 0.25 ± 0.01 | 0.22 ± 0.02 | 0.33 ± 0.03 | **0.37 ± 0.03** | 0.55 ± 0.00 | 0.26 ± 0.00 |
| Multi Head + GradNorm | 0.21 ± 0.12 | 0.12 ± 0.01 | 0.35 ± 0.02 | 0.38 ± 0.06 | 0.36 ± 0.05 | 0.35 ± 0.08 | 0.48 ± 0.17 | 0.43 ± 0.02 | 0.30 ± 0.05 | 0.25 ± 0.03 | 0.36 ± 0.05 | 0.80 ± 0.21 | 0.68 ± 0.00 | 0.27 ± 0.01 |
| Multi Head + PcGrad | 0.11 ± 0.00 | 0.08 ± 0.00 | 0.29 ± 0.01 | 0.27 ± 0.02 | 0.28 ± 0.02 | 0.28 ± 0.01 | 0.29 ± 0.01 | **0.33 ± 0.00** | 0.28 ± 0.01 | 0.22 ± 0.01 | 0.33 ± 0.02 | 0.42 ± 0.02 | 0.55 ± 0.00 | 0.26 ± 0.00 |
| Multiple Models | 0.10 ± 0.01 | 0.09 ± 0.00 | **0.28 ± 0.00** | 0.25 ± 0.01 | 0.25 ± 0.01 | 0.26 ± 0.03 | **0.28 ± 0.00** | **0.33 ± 0.00** | 0.25 ± 0.02 | 0.22 ± 0.01 | 0.34 ± 0.03 | 0.39 ± 0.05 | 0.56 ± 0.01 | 0.26 ± 0.00 |
| Multiple Models + COMs | 0.10 ± 0.00 | 0.09 ± 0.01 | **0.28 ± 0.00** | 0.27 ± 0.01 | 0.27 ± 0.02 | 0.25 ± 0.01 | 0.29 ± 0.00 | **0.33 ± 0.00** | 0.27 ± 0.03 | 0.23 ± 0.01 | 0.31 ± 0.01 | 0.40 ± 0.01 | 0.56 ± 0.00 | 0.27 ± 0.01 |
| Multiple Models + RoMA | 0.10 ± 0.01 | 0.09 ± 0.00 | 0.29 ± 0.00 | 0.29 ± 0.02 | 0.27 ± 0.01 | **0.24 ± 0.01** | 0.30 ± 0.02 | 0.35 ± 0.00 | 0.27 ± 0.01 | 0.22 ± 0.01 | 0.31 ± 0.01 | 0.40 ± 0.03 | 0.58 ± 0.01 | 0.27 ± 0.02 |
| Multiple Models + IOM | 0.11 ± 0.00 | 0.11 ± 0.01 | 0.29 ± 0.00 | **0.24 ± 0.02** | 0.26 ± 0.02 | 0.25 ± 0.01 | **0.28 ± 0.00** | **0.33 ± 0.00** | 0.28 ± 0.01 | 0.22 ± 0.01 | 0.32 ± 0.02 | 0.43 ± 0.02 | **0.54 ± 0.00** | 0.27 ± 0.01 |
| Multiple Models + ICT | 0.11 ± 0.00 | 0.09 ± 0.02 | 0.32 ± 0.02 | 0.37 ± 0.11 | 0.26 ± 0.03 | 0.29 ± 0.01 | 0.30 ± 0.01 | 0.34 ± 0.01 | 0.26 ± 0.02 | 0.26 ± 0.01 | 0.31 ± 0.01 | 0.45 ± 0.05 | 0.57 ± 0.01 | 0.27 ± 0.02 |
| Multiple Models + Tri-Mentoring | 0.10 ± 0.01 | 0.08 ± 0.01 | 0.30 ± 0.00 | 0.34 ± 0.04 | 0.29 ± 0.02 | 0.29 ± 0.03 | 0.33 ± 0.01 | 0.37 ± 0.01 | 0.28 ± 0.01 | 0.25 ± 0.02 | 0.32 ± 0.01 | 0.41 ± 0.06 | 0.55 ± 0.00 | 0.30 ± 0.02 |
| MOBO | 0.10 ± 0.00 | 0.08 ± 0.00 | 0.29 ± 0.00 | 0.28 ± 0.01 | 0.29 ± 0.02 | 0.25 ± 0.01 | 0.29 ± 0.00 | **0.33 ± 0.00** | 0.32 ± 0.01 | **0.20 ± 0.01** | 0.35 ± 0.01 | 0.46 ± 0.02 | **0.54 ± 0.00** | N/A |
| MOBO-qParEGO | 0.11 ± 0.00 | 0.11 ± 0.00 | 0.37 ± 0.00 | 0.27 ± 0.01 | 0.28 ± 0.00 | 0.34 ± 0.01 | 0.33 ± 0.01 | 0.37 ± 0.00 | 0.35 ± 0.01 | 0.30 ± 0.03 | 0.36 ± 0.02 | 0.43 ± 0.01 | 0.56 ± 0.00 | N/A |
| MOBO-JES | N/A | N/A | N/A | N/A | N/A | N/A | N/A | N/A | N/A | N/A | N/A | N/A | N/A | N/A |
| ParetoFlow | **0.09 ± 0.02** | **0.05 ± 0.02** | 0.32 ± 0.01 | 0.31 ± 0.02 | **0.24 ± 0.00** | 0.30 ± 0.02 | 0.33 ± 0.02 | 0.36 ± 0.01 | 0.30 ± 0.01 | N/A | N/A | 0.59 ± 0.03 | 0.58 ± 0.00 | N/A |
| DOMOO (ours) | 0.11 ± 0.00 | 0.10 ± 0.00 | 0.29 ± 0.00 | 0.25 ± 0.01 | 0.26 ± 0.01 | 0.25 ± 0.01 | 0.29 ± 0.00 | 0.34 ± 0.00 | 0.26 ± 0.02 | 0.23 ± 0.01 | 0.35 ± 0.04 | 0.38 ± 0.03 | 0.57 ± 0.01 | 0.27 ± 0.01 |

*Table 22.* IGD$_{\text{offline}}$ results for MORL with 256 solutions and $100^{th}$ percentile evaluations. For each task, algorithms within one standard deviation of having the highest performance are **bolded**.

| Methods | MO-Swimmer | MO-Hopper |
|---|---|---|
| $\mathcal{D}$(best) | **0.43** | 0.8 |
| End-to-End | 0.47 ± 0.00 | 0.64 ± 0.00 |
| End-to-End + GradNorm | 0.59 ± 0.00 | 0.76 ± 0.00 |
| End-to-End + PcGrad | 0.49 ± 0.00 | 0.77 ± 0.00 |
| Multi Head | 0.48 ± 0.00 | 0.70 ± 0.00 |
| Multi Head + GradNorm | 0.50 ± 0.00 | 0.91 ± 0.00 |
| Multi Head + PcGrad | 0.53 ± 0.00 | 0.67 ± 0.00 |
| Multiple Models | 0.48 ± 0.00 | 0.65 ± 0.00 |
| Multiple Models + COMs | 0.45 ± 0.00 | 0.68 ± 0.00 |
| Multiple Models + RoMA | 0.45 ± 0.00 | 0.64 ± 0.00 |
| Multiple Models + IOM | 0.54 ± 0.00 | 0.59 ± 0.00 |
| Multiple Models + ICT | 0.49 ± 0.03 | 0.70 ± 0.08 |
| Multiple Models + Tri-Mentoring | 0.49 ± 0.04 | 0.73 ± 0.05 |
| MOBO | N/A | N/A |
| MOBO-qParEGO | N/A | N/A |
| MOBO-JES | N/A | N/A |
| ParetoFlow | 0.45 ± 0.00 | 0.80 ± 0.00 |
| DOMOO (ours) | 0.49 ± 0.00 | **0.58 ± 0.07** |

*Table 23.* IGD$_{\text{offline}}$ results for RE with 256 solutions and $100^{th}$ percentile evaluations. For each task, algorithms within one standard deviation of having the highest performance are **bolded**.

| Methods | RE21 | RE22 | RE23 | RE24 | RE25 | RE31 | RE32 | RE33 | RE34 | RE35 | RE36 | RE37 | MO-Portfolio |
|---|---|---|---|---|---|---|---|---|---|---|---|---|---|
| $\mathcal{D}$(best) | 0.56 | **0.00** | 0.00 | **0.00** | 0.03 | 0.01 | 0.02 | 0.04 | 0.34 | 0.09 | 0.69 | 0.65 | 0.47 |
| End-to-End | 0.45 ± 0.00 | 0.21 ± 0.02 | 0.03 ± 0.02 | 0.11 ± 0.10 | 0.09 ± 0.05 | 0.27 ± 0.00 | 0.09 ± 0.02 | 0.05 ± 0.00 | 0.30 ± 0.00 | 0.33 ± 0.05 | 0.36 ± 0.02 | 0.52 ± 0.00 | 0.55 ± 0.00 |
| End-to-End + GradNorm | 0.46 ± 0.00 | 0.15 ± 0.06 | 0.32 ± 0.35 | 0.36 ± 0.43 | 0.07 ± 0.00 | 0.61 ± 1.18 | 0.06 ± 0.01 | 0.07 ± 0.01 | 0.32 ± 0.00 | 0.34 ± 0.03 | 3.08 ± 0.00 | 0.52 ± 0.00 | 0.56 ± 0.00 |
| End-to-End + PcGrad | 0.45 ± 0.00 | 0.15 ± 0.08 | 0.03 ± 0.02 | 0.23 ± 0.05 | 0.07 ± 0.00 | 0.22 ± 0.03 | 0.11 ± 0.01 | 0.07 ± 0.02 | 0.30 ± 0.00 | 0.16 ± 0.04 | 0.39 ± 0.03 | 0.52 ± 0.00 | 0.55 ± 0.01 |
| Multi Head | 0.45 ± 0.00 | 0.17 ± 0.04 | 0.03 ± 0.01 | 0.01 ± 0.01 | 0.09 ± 0.05 | 0.26 ± 0.02 | 0.09 ± 0.02 | 0.07 ± 0.00 | 0.30 ± 0.00 | 0.18 ± 0.08 | 0.36 ± 0.02 | 0.51 ± 0.00 | 0.56 ± 0.01 |
| Multi Head + GradNorm | 0.47 ± 0.03 | 0.25 ± 0.23 | 0.41 ± 0.48 | 0.37 ± 0.42 | 0.11 ± 0.08 | 0.17 ± 0.04 | 0.04 ± 0.00 | 0.20 ± 0.20 | 0.31 ± 0.01 | 0.27 ± 0.13 | 0.74 ± 0.68 | 0.61 ± 0.17 | 0.60 ± 0.04 |
| Multi Head + PcGrad | 0.45 ± 0.00 | 9.99 ± 19.92 | 0.02 ± 0.02 | 0.69 ± 0.24 | 0.10 ± 0.05 | 0.19 ± 0.06 | 0.08 ± 0.03 | 0.06 ± 0.01 | 0.30 ± 0.00 | 0.12 ± 0.05 | 0.47 ± 0.02 | 0.52 ± 0.00 | 0.58 ± 0.02 |
| Multiple Models | 0.45 ± 0.00 | 0.07 ± 0.00 | 0.04 ± 0.00 | 0.04 ± 0.03 | 0.12 ± 0.03 | 0.26 ± 0.01 | 0.11 ± 0.00 | 0.08 ± 0.02 | 0.30 ± 0.00 | 0.14 ± 0.05 | 0.35 ± 0.01 | 0.52 ± 0.00 | 0.57 ± 0.02 |
| Multiple Models + COMs | 0.45 ± 0.00 | 0.10 ± 0.06 | 0.01 ± 0.01 | 0.04 ± 0.04 | 0.11 ± 0.07 | 0.19 ± 0.05 | 0.11 ± 0.01 | 0.06 ± 0.00 | 0.30 ± 0.00 | **0.06 ± 0.00** | 0.40 ± 0.02 | 0.53 ± 0.01 | 1.07 ± 0.20 |
| Multiple Models + RoMA | 0.47 ± 0.00 | 0.09 ± 0.07 | 0.02 ± 0.02 | 0.43 ± 0.38 | 0.11 ± 0.08 | 0.01 ± 0.00 | 0.02 ± 0.00 | 0.05 ± 0.00 | 0.31 ± 0.00 | 0.08 ± 0.01 | 0.40 ± 0.03 | 0.52 ± 0.00 | 0.57 ± 0.01 |
| Multiple Models + IOM | 0.45 ± 0.00 | 0.02 ± 0.02 | **0.00 ± 0.01** | **0.00 ± 0.00** | 0.02 ± 0.02 | 0.21 ± 0.04 | 0.06 ± 0.01 | 0.05 ± 0.00 | 0.30 ± 0.00 | **0.06 ± 0.00** | 0.37 ± 0.04 | 0.52 ± 0.00 | 0.59 ± 0.01 |
| Multiple Models + ICT | 0.45 ± 0.00 | 0.13 ± 0.09 | 0.08 ± 0.06 | 0.10 ± 0.12 | 0.03 ± 0.02 | 0.04 ± 0.03 | 0.07 ± 0.02 | 0.06 ± 0.01 | 0.30 ± 0.00 | 0.09 ± 0.02 | 0.36 ± 0.02 | 0.52 ± 0.00 | 0.57 ± 0.01 |
| Multiple Models + Tri-Mentoring | 0.45 ± 0.00 | 0.20 ± 0.02 | 0.06 ± 0.03 | **0.00 ± 0.00** | 0.06 ± 0.02 | 0.02 ± 0.01 | 0.09 ± 0.01 | 0.07 ± 0.02 | 0.30 ± 0.00 | 0.10 ± 0.07 | 0.39 ± 0.03 | 0.52 ± 0.00 | 0.55 ± 0.00 |
| MOBO | 0.48 ± 0.01 | **0.00 ± 0.00** | 0.00 ± 0.00 | 0.03 ± 0.01 | **0.01 ± 0.00** | 0.04 ± 0.00 | 0.02 ± 0.00 | 0.07 ± 0.00 | 0.32 ± 0.00 | **0.06 ± 0.00** | 0.33 ± 0.00 | 0.51 ± 0.00 | 1.33 ± 0.18 |
| MOBO-qParEGO | 0.45 ± 0.00 | **0.00 ± 0.00** | 0.00 ± 0.00 | 0.01 ± 0.01 | **0.01 ± 0.00** | 0.04 ± 0.00 | 0.04 ± 0.00 | 0.04 ± 0.00 | 0.35 ± 0.00 | 0.07 ± 0.00 | 0.35 ± 0.00 | 0.51 ± 0.00 | **0.38 ± 0.01** |
| MOBO-JES | 0.45 ± 0.00 | 0.02 ± 0.00 | 0.00 ± 0.00 | **0.00 ± 0.00** | 0.07 ± 0.02 | N/A | N/A | 0.07 ± 0.01 | 0.36 ± 0.00 | 0.09 ± 0.00 | N/A | N/A | N/A |
| ParetoFlow | **0.37 ± 0.07** | **0.00 ± 0.00** | N/A | N/A | N/A | **0.00 ± 0.00** | **0.00 ± 0.00** | 0.03 ± 0.01 | 0.22 ± 0.05 | N/A | 0.47 ± 0.12 | **0.44 ± 0.06** | 0.41 ± 0.04 |
| DOMOO (ours) | 0.45 ± 0.00 | 0.06 ± 0.01 | 0.04 ± 0.00 | 0.01 ± 0.02 | 0.06 ± 0.01 | 0.22 ± 0.01 | 0.05 ± 0.02 | 0.06 ± 0.01 | 0.30 ± 0.00 | 0.08 ± 0.00 | 0.35 ± 0.01 | 0.52 ± 0.00 | 0.39 ± 0.01 |

*Table 24.* IGD$_{\text{offline}}$ results for scientific design with 256 solutions and $100^{th}$ percentile evaluations. For each task, algorithms within one standard deviation of having the highest performance are **bolded**.

| Methods | Molecule | Regex | RFP | ZINC |
|---|---|---|---|---|
| $\mathcal{D}$(best) | 0.84 | 1.05 | 0.39 | 0.2 |
| End-to-End | $0.94 \pm 0.01$ | $1.09 \pm 0.02$ | $0.31 \pm 0.05$ | $0.17 \pm 0.01$ |
| End-to-End + GradNorm | $1.44 \pm 0.01$ | $1.04 \pm 0.00$ | $0.31 \pm 0.07$ | $0.17 \pm 0.01$ |
| End-to-End + PcGrad | $1.10 \pm 0.24$ | $1.04 \pm 0.00$ | $0.33 \pm 0.05$ | $0.16 \pm 0.00$ |
| Multi Head | $0.94 \pm 0.01$ | $1.08 \pm 0.00$ | $0.30 \pm 0.07$ | $0.17 \pm 0.01$ |
| Multi Head + GradNorm | $0.94 \pm 0.10$ | $1.06 \pm 0.01$ | $0.30 \pm 0.06$ | $0.18 \pm 0.00$ |
| Multi Head + PcGrad | $1.02 \pm 0.17$ | $1.04 \pm 0.00$ | $\mathbf{0.28 \pm 0.04}$ | $0.16 \pm 0.01$ |
| Multiple Models | $0.94 \pm 0.01$ | $1.08 \pm 0.00$ | $0.39 \pm 0.01$ | $0.17 \pm 0.01$ |
| Multiple Models + COMs | $0.79 \pm 0.06$ | $1.04 \pm 0.00$ | $0.33 \pm 0.06$ | $0.16 \pm 0.00$ |
| Multiple Models + RoMA | $0.87 \pm 0.26$ | $1.04 \pm 0.00$ | $0.35 \pm 0.07$ | $0.17 \pm 0.00$ |
| Multiple Models + IOM | $0.75 \pm 0.08$ | $1.04 \pm 0.00$ | $0.35 \pm 0.06$ | $0.18 \pm 0.00$ |
| Multiple Models + ICT | $0.85 \pm 0.00$ | $1.04 \pm 0.00$ | $0.29 \pm 0.05$ | $0.18 \pm 0.00$ |
| Multiple Models + Tri-Mentoring | $1.24 \pm 0.15$ | $1.04 \pm 0.00$ | $0.37 \pm 0.06$ | $0.18 \pm 0.00$ |
| MOBO | $0.76 \pm 0.22$ | $\mathbf{0.75 \pm 0.01}$ | $0.38 \pm 0.01$ | $\mathbf{0.14 \pm 0.00}$ |
| MOBO-$q$ParEGO | N/A | $0.88 \pm 0.00$ | $0.39 \pm 0.01$ | $0.16 \pm 0.01$ |
| MOBO-JES | N/A | N/A | N/A | N/A |
| ParetoFlow | $\mathbf{0.64 \pm 0.47}$ | $0.87 \pm 0.00$ | N/A | $0.15 \pm 0.01$ |
| DOMOO (**ours**) | $0.86 \pm 0.02$ | $0.88 \pm 0.04$ | $0.35 \pm 0.06$ | $0.17 \pm 0.01$ |

## H.2. The $50^{th}$ Percentile Results

As shown in Table 25, we report the $50^{th}$ percentile IGD$_{\text{offline}}$ average ranks with 256 solutions. As shown in Table 26, Table 27, Table 28, Table 29, and Table 30, we report the $50^{th}$ percentile IGD$_{\text{offline}}$ results with 256 solutions. DOMOO consistently performs well across tasks. Methods within one standard deviation of the best are highlighted in **bold**.

*Table 25.* Comparison of average IGD$_{\text{offline}}$ ranks at the $50^{th}$ percentile achieved by different offline MOO methods across different tasks in Off-MOO-Bench (Xue et al., 2024). Details are the same as Table 14.

| Methods | Synthetic | MO-NAS | MORL | Sci-Design | RE | Average Rank |
|---|---|---|---|---|---|---|
| $\mathcal{D}$(best) | $8.11 \pm 0.55$ | $9.31 \pm 1.28$ | $\mathbf{1.20 \pm 0.24}$ | $\mathbf{1.80 \pm 0.43}$ | $\underline{5.63 \pm 0.60}$ | $7.00 \pm 0.68$ |
| End-to-End | $7.38 \pm 1.31$ | $\mathbf{4.64 \pm 1.07}$ | $10.50 \pm 0.55$ | $8.97 \pm 0.53$ | $9.06 \pm 1.08$ | $7.30 \pm 0.89$ |
| End-to-End + GradNorm | $12.01 \pm 1.47$ | $12.10 \pm 2.39$ | $11.40 \pm 0.66$ | $9.57 \pm 1.41$ | $11.18 \pm 0.81$ | $11.59 \pm 1.36$ |
| End-to-End + PcGrad | $\underline{6.40 \pm 0.87}$ | $6.99 \pm 1.13$ | $8.90 \pm 0.58$ | $7.47 \pm 1.72$ | $9.26 \pm 0.65$ | $7.52 \pm 0.65$ |
| Multi Head | $6.88 \pm 1.15$ | $\underline{5.01 \pm 0.52}$ | $4.90 \pm 0.58$ | $7.17 \pm 0.19$ | $7.71 \pm 0.41$ | $\underline{6.62 \pm 0.45}$ |
| Multi Head + GradNorm | $10.56 \pm 1.28$ | $12.58 \pm 1.29$ | $13.00 \pm 0.63$ | $\underline{6.20 \pm 1.50}$ | $11.75 \pm 1.32$ | $11.10 \pm 1.22$ |
| Multi Head + PcGrad | $7.83 \pm 1.23$ | $7.42 \pm 1.06$ | $6.20 \pm 0.68$ | $8.35 \pm 2.09$ | $9.71 \pm 1.03$ | $8.18 \pm 0.85$ |
| Multiple Models | $\underline{5.75 \pm 0.80}$ | $\underline{5.28 \pm 1.02}$ | $10.40 \pm 0.80$ | $9.57 \pm 2.30$ | $7.96 \pm 0.51$ | $\underline{6.74 \pm 0.32}$ |
| MultipleModels + COMs | $8.57 \pm 0.49$ | $6.56 \pm 0.87$ | $6.00 \pm 0.77$ | $6.72 \pm 2.72$ | $9.12 \pm 0.95$ | $7.96 \pm 0.48$ |
| Multiple Models + RoMA | $10.45 \pm 0.53$ | $7.12 \pm 1.27$ | $7.20 \pm 0.40$ | $8.15 \pm 0.56$ | $9.43 \pm 0.90$ | $8.93 \pm 0.68$ |
| Multiple Models + IOM | $7.02 \pm 0.73$ | $7.85 \pm 2.72$ | $\underline{4.60 \pm 0.49}$ | $9.78 \pm 1.59$ | $\mathbf{5.32 \pm 0.79}$ | $\mathbf{6.51 \pm 0.54}$ |
| Multiple Models + ICT | $8.81 \pm 0.69$ | $8.80 \pm 1.05$ | $8.00 \pm 2.63$ | $9.55 \pm 2.04$ | $7.62 \pm 0.69$ | $8.50 \pm 0.26$ |
| Multiple Models + Tri-Mentoring | $9.77 \pm 0.71$ | $9.71 \pm 0.92$ | $8.40 \pm 1.59$ | $10.30 \pm 1.93$ | $8.49 \pm 0.50$ | $9.36 \pm 0.45$ |
| MOBO | $10.85 \pm 1.13$ | $6.26 \pm 0.68$ | N/A | $10.27 \pm 1.29$ | $10.57 \pm 0.83$ | $9.08 \pm 0.91$ |
| MOBO-$q$ParEGO | $10.82 \pm 1.15$ | $11.27 \pm 0.71$ | N/A | $8.00 \pm 1.14$ | $7.56 \pm 0.53$ | $9.92 \pm 0.32$ |
| MOBO-JES | $14.58 \pm 0.63$ | N/A | N/A | N/A | $9.51 \pm 1.88$ | $11.08 \pm 2.03$ |
| ParetoFlow | $8.63 \pm 2.25$ | $9.19 \pm 0.65$ | $10.83 \pm 5.54$ | $\underline{6.21 \pm 1.74}$ | $\underline{5.35 \pm 0.43}$ | $8.55 \pm 2.23$ |
| DOMOO (**ours**) | $\mathbf{5.66 \pm 0.94}$ | $8.70 \pm 0.62$ | $\underline{3.10 \pm 2.26}$ | $8.60 \pm 0.87$ | $6.21 \pm 0.31$ | $6.77 \pm 0.57$ |

*Table 26.* IGD$_{\text{offline}}$ results for synthetic functions with 256 solutions and $50^{th}$ percentile evaluations. For each task, algorithms within one standard deviation of having the highest performance are **bolded**.

| Methods | DTLZ1 | DTLZ2 | DTLZ3 | DTLZ4 | DTLZ5 | DTLZ6 | DTLZ7 | OmniTest | VLMOP1 | VLMOP2 | VLMOP3 | ZDT1 | ZDT2 | ZDT3 | ZDT4 | ZDT6 |
|---|---|---|---|---|---|---|---|---|---|---|---|---|---|---|---|---|
| $\mathcal{D}$(best) | 0.25 | **0.27** | 0.23 | **0.01** | **0.35** | **0.43** | 0.61 | 0.46 | 0.06 | 1.34 | 0.08 | 0.48 | 0.45 | 0.55 | **0.07** | 0.14 |
| End-to-End | 0.24 ± 0.03 | 0.83 ± 0.06 | 0.28 ± 0.03 | 0.89 ± 0.10 | 0.93 ± 0.05 | 0.65 ± 0.12 | 0.30 ± 0.01 | 0.22 ± 0.00 | 0.11 ± 0.08 | 0.91 ± 0.00 | 0.09 ± 0.05 | **0.14 ± 0.00** | 0.21 ± 0.00 | 0.48 ± 0.16 | 0.55 ± 0.08 | 0.13 ± 0.00 |
| End-to-End + GradNorm | 0.23 ± 0.01 | 0.93 ± 0.07 | 0.23 ± 0.05 | 0.95 ± 0.14 | 1.04 ± 0.05 | 0.45 ± 0.05 | 0.48 ± 0.08 | 0.81 ± 0.13 | 0.38 ± 0.18 | 1.45 ± 0.00 | 0.40 ± 0.20 | 0.74 ± 0.25 | 0.76 ± 0.21 | 0.83 ± 0.24 | 0.78 ± 0.25 | |
| End-to-End + PcGrad | 0.21 ± 0.02 | 0.67 ± 0.11 | 0.26 ± 0.01 | 0.72 ± 0.15 | 0.73 ± 0.08 | 0.58 ± 0.05 | **0.29 ± 0.03** | 0.22 ± 0.00 | **0.03 ± 0.00** | 0.91 ± 0.00 | 0.07 ± 0.01 | **0.14 ± 0.00** | 0.21 ± 0.00 | 0.45 ± 0.03 | 0.97 ± 0.07 | 0.42 ± 0.35 |
| Multi Head | 0.18 ± 0.01 | 0.82 ± 0.08 | 0.25 ± 0.03 | 0.97 ± 0.12 | 0.86 ± 0.04 | 0.67 ± 0.09 | 0.35 ± 0.04 | 0.22 ± 0.00 | 0.09 ± 0.05 | 0.91 ± 0.00 | **0.06 ± 0.01** | 0.16 ± 0.02 | 0.26 ± 0.06 | 0.38 ± 0.08 | 0.63 ± 0.12 | 0.12 ± 0.00 |
| Multi Head + GradNorm | 0.22 ± 0.01 | 0.82 ± 0.15 | 0.22 ± 0.05 | 0.97 ± 0.03 | 0.91 ± 0.09 | 0.59 ± 0.06 | 0.47 ± 0.05 | 0.79 ± 0.12 | 0.86 ± 0.16 | 0.91 ± 0.00 | 0.34 ± 0.22 | 0.31 ± 0.09 | 0.38 ± 0.14 | 0.39 ± 0.07 | 0.97 ± 0.16 | 0.52 ± 0.47 |
| Multi Head + PcGrad | 0.22 ± 0.01 | 0.69 ± 0.05 | 0.29 ± 0.02 | 1.03 ± 0.08 | 0.69 ± 0.04 | 0.57 ± 0.03 | 0.34 ± 0.06 | 0.22 ± 0.00 | 0.14 ± 0.11 | 0.91 ± 0.00 | **0.06 ± 0.01** | 0.18 ± 0.07 | 0.24 ± 0.05 | 0.45 ± 0.05 | 0.82 ± 0.19 | 0.51 ± 0.45 |
| Multiple Models | 0.18 ± 0.01 | 0.77 ± 0.03 | **0.20 ± 0.04** | 0.82 ± 0.04 | 0.81 ± 0.08 | 0.66 ± 0.11 | 0.30 ± 0.03 | 0.22 ± 0.00 | 0.10 ± 0.10 | 0.91 ± 0.00 | 0.06 ± 0.00 | 0.16 ± 0.01 | 0.22 ± 0.02 | 0.38 ± 0.06 | 0.40 ± 0.10 | 0.14 ± 0.02 |
| Multiple Models + COMs | 0.22 ± 0.01 | 0.52 ± 0.06 | 0.33 ± 0.02 | 0.72 ± 0.07 | 0.60 ± 0.06 | 0.61 ± 0.01 | 0.42 ± 0.01 | 0.22 ± 0.00 | 0.12 ± 0.11 | 0.92 ± 0.00 | 0.17 ± 0.09 | 0.23 ± 0.02 | 0.31 ± 0.01 | 0.41 ± 0.01 | 0.49 ± 0.05 | 0.99 ± 0.16 |
| Multiple Models + RoMA | 0.25 ± 0.01 | 0.75 ± 0.02 | 0.27 ± 0.05 | 0.99 ± 0.02 | 0.94 ± 0.03 | 0.54 ± 0.01 | 0.39 ± 0.02 | 0.82 ± 0.02 | **0.03 ± 0.00** | 1.45 ± 0.00 | 0.34 ± 0.09 | **0.14 ± 0.00** | 0.32 ± 0.03 | **0.28 ± 0.01** | 0.95 ± 0.08 | 1.18 ± 0.08 |
| Multiple Models + IOM | 0.22 ± 0.00 | 0.47 ± 0.11 | 0.31 ± 0.05 | 0.52 ± 0.01 | 0.49 ± 0.02 | 0.64 ± 0.06 | 0.32 ± 0.04 | 0.23 ± 0.00 | 0.13 ± 0.09 | 0.91 ± 0.01 | 0.20 ± 0.04 | 0.26 ± 0.04 | **0.21 ± 0.01** | 0.35 ± 0.02 | 0.52 ± 0.16 | **0.12 ± 0.01** |
| Multiple Models + ICT | 0.22 ± 0.01 | 0.59 ± 0.04 | 0.33 ± 0.04 | 0.73 ± 0.07 | 0.76 ± 0.08 | 0.65 ± 0.09 | 0.43 ± 0.07 | 0.23 ± 0.01 | 0.08 ± 0.03 | 0.91 ± 0.00 | 0.11 ± 0.07 | 0.17 ± 0.01 | 0.31 ± 0.06 | 0.47 ± 0.05 | 0.78 ± 0.06 | 0.52 ± 0.31 |
| Multiple Models + Tri-Mentoring | 0.27 ± 0.06 | 0.71 ± 0.06 | 0.30 ± 0.04 | 0.77 ± 0.13 | 0.89 ± 0.05 | 0.71 ± 0.09 | 0.37 ± 0.02 | 0.23 ± 0.01 | 0.06 ± 0.03 | 0.93 ± 0.04 | 0.10 ± 0.07 | 0.19 ± 0.01 | 0.27 ± 0.07 | 0.56 ± 0.05 | 0.54 ± 0.08 | 0.90 ± 0.08 |
| MOBO | **0.17 ± 0.00** | 0.35 ± 0.00 | 0.36 ± 0.03 | 0.49 ± 0.01 | 0.43 ± 0.00 | 0.61 ± 0.02 | 0.60 ± 0.00 | 0.27 ± 0.02 | **0.03 ± 0.00** | 1.45 ± 0.00 | N/A | 0.38 ± 0.01 | 0.59 ± 0.00 | 0.53 ± 0.03 | 0.81 ± 0.01 | 0.93 ± 0.03 |
| MOBO-qParEGO | 0.22 ± 0.00 | 0.36 ± 0.00 | 0.39 ± 0.01 | 0.46 ± 0.00 | 0.45 ± 0.00 | 0.64 ± 0.06 | 0.43 ± 0.01 | 0.50 ± 0.07 | 0.04 ± 0.01 | 1.45 ± 0.00 | 0.13 ± 0.00 | 0.38 ± 0.01 | 0.46 ± 0.00 | 0.48 ± 0.01 | 0.59 ± 0.01 | 1.00 ± 0.16 |
| MOBO-JES | N/A | N/A | N/A | N/A | N/A | N/A | N/A | 0.48 ± 0.01 | 0.08 ± 0.00 | N/A | N/A | 0.58 ± 0.01 | 0.49 ± 0.06 | 0.65 ± 0.02 | 0.74 ± 0.02 | 1.21 ± 0.00 |
| ParetoFlow | 0.18 ± 0.02 | 0.45 ± 0.07 | 0.35 ± 0.03 | 0.16 ± 0.07 | 0.55 ± 0.06 | 0.79 ± 0.06 | 0.56 ± 0.03 | **0.19 ± 0.00** | N/A | **0.84 ± 0.00** | N/A | 0.46 ± 0.02 | 0.46 ± 0.04 | 0.55 ± 0.05 | 0.10 ± 0.01 | 0.12 ± 0.09 |
| DOMOO (**ours**) | 0.18 ± 0.01 | 0.42 ± 0.05 | 0.22 ± 0.05 | 0.95 ± 0.07 | 0.52 ± 0.07 | 0.55 ± 0.08 | 0.29 ± 0.02 | 0.22 ± 0.00 | 0.05 ± 0.03 | 0.91 ± 0.00 | 0.11 ± 0.04 | 0.18 ± 0.03 | 0.22 ± 0.01 | 0.40 ± 0.04 | 0.74 ± 0.14 | 0.13 ± 0.01 |

*Table 27.* IGD$_{\text{offline}}$ results for MO-NAS with 256 solutions and $50^{th}$ percentile evaluations. For each task, algorithms within one standard deviation of having the highest performance are **bolded**.

| Methods | C-10/MOP1 | C-10/MOP2 | C-10/MOP3 | C-10/MOP8 | C-10/MOP9 | IN-1K/MOP1 | IN-1K/MOP2 | IN-1K/MOP3 | IN-1K/MOP4 | IN-1K/MOP5 | IN-1K/MOP6 | IN-1K/MOP7 | IN-1K/MOP8 | NasBench201-Test |
|---|---|---|---|---|---|---|---|---|---|---|---|---|---|---|
| $\mathcal{D}$(best) | 0.11 | 0.1 | 0.34 | 0.36 | 0.33 | 0.34 | 0.32 | 0.37 | 0.35 | 0.29 | 0.37 | 0.64 | 0.62 | 0.32 |
| End-to-End | 0.12 ± 0.01 | 0.10 ± 0.01 | **0.30 ± 0.01** | 0.32 ± 0.02 | 0.31 ± 0.02 | 0.28 ± 0.03 | **0.29 ± 0.00** | 0.34 ± 0.00 | **0.27 ± 0.02** | **0.25 ± 0.02** | 0.36 ± 0.04 | 0.54 ± 0.09 | 0.57 ± 0.02 | 0.29 ± 0.02 |
| End-to-End + GradNorm | 0.27 ± 0.10 | 0.11 ± 0.01 | 0.42 ± 0.02 | 0.59 ± 0.13 | 0.40 ± 0.04 | 0.34 ± 0.02 | 0.32 ± 0.01 | 0.50 ± 0.02 | 0.48 ± 0.11 | 0.35 ± 0.07 | 0.65 ± 0.19 | 0.60 ± 0.14 | 0.68 ± 0.02 | 0.54 ± 0.32 |
| End-to-End + PcGrad | 0.13 ± 0.02 | 0.10 ± 0.01 | 0.31 ± 0.00 | 0.39 ± 0.03 | 0.31 ± 0.02 | 0.31 ± 0.01 | 0.30 ± 0.01 | 0.34 ± 0.01 | 0.35 ± 0.04 | 0.28 ± 0.02 | 0.37 ± 0.02 | 0.50 ± 0.04 | 0.56 ± 0.01 | 0.32 ± 0.04 |
| Multi Head | 0.11 ± 0.00 | 0.11 ± 0.01 | 0.30 ± 0.00 | 0.32 ± 0.02 | 0.34 ± 0.03 | 0.26 ± 0.01 | 0.33 ± 0.00 | 0.33 ± 0.00 | 0.27 ± 0.01 | 0.26 ± 0.02 | 0.38 ± 0.04 | 0.46 ± 0.10 | 0.55 ± 0.00 | 0.31 ± 0.03 |
| Multi Head + GradNorm | inf ± nan | 0.12 ± 0.01 | 0.39 ± 0.03 | 0.48 ± 0.07 | 0.52 ± 0.02 | 0.45 ± 0.14 | 0.60 ± 0.22 | 0.48 ± 0.03 | 0.35 ± 0.05 | 0.30 ± 0.04 | 0.42 ± 0.06 | 0.92 ± 0.21 | 0.73 ± 0.00 | 0.30 ± 0.01 |
| Multi Head + PcGrad | 0.14 ± 0.05 | 0.09 ± 0.01 | 0.31 ± 0.01 | 0.48 ± 0.06 | 0.34 ± 0.03 | 0.30 ± 0.01 | 0.30 ± 0.00 | 0.33 ± 0.00 | 0.34 ± 0.02 | 0.27 ± 0.01 | 0.36 ± 0.02 | 0.47 ± 0.02 | 0.56 ± 0.00 | 0.29 ± 0.02 |
| Multiple Models | 0.12 ± 0.01 | 0.15 ± 0.07 | **0.30 ± 0.01** | **0.30 ± 0.02** | 0.33 ± 0.02 | 0.28 ± 0.06 | **0.29 ± 0.00** | 0.33 ± 0.00 | 0.28 ± 0.01 | 0.25 ± 0.01 | 0.38 ± 0.04 | 0.45 ± 0.06 | 0.56 ± 0.01 | 0.30 ± 0.02 |
| Multiple Models + COMs | 0.11 ± 0.00 | 0.10 ± 0.01 | 0.31 ± 0.00 | 0.38 ± 0.03 | 0.34 ± 0.02 | 0.28 ± 0.01 | 0.30 ± 0.00 | 0.34 ± 0.00 | 0.32 ± 0.03 | 0.27 ± 0.01 | **0.35 ± 0.01** | 0.51 ± 0.05 | 0.57 ± 0.01 | 0.34 ± 0.04 |
| Multiple Models + RoMA | 0.12 ± 0.01 | 0.10 ± 0.01 | 0.32 ± 0.01 | 0.34 ± 0.02 | 0.36 ± 0.01 | **0.25 ± 0.01** | 0.32 ± 0.02 | 0.37 ± 0.01 | 0.31 ± 0.03 | 0.25 ± 0.01 | **0.35 ± 0.01** | 0.46 ± 0.05 | 0.60 ± 0.02 | 0.32 ± 0.02 |
| Multiple Models + IOM | 0.12 ± 0.01 | 0.12 ± 0.01 | 0.31 ± 0.00 | 0.32 ± 0.03 | 0.32 ± 0.02 | 0.27 ± 0.01 | **0.29 ± 0.00** | **0.33 ± 0.01** | 0.33 ± 0.03 | 0.26 ± 0.00 | 0.36 ± 0.03 | 0.51 ± 0.04 | 0.55 ± 0.00 | 0.33 ± 0.03 |
| Multiple Models + ICT | 0.12 ± 0.01 | 0.13 ± 0.06 | 0.34 ± 0.03 | 0.46 ± 0.11 | 0.33 ± 0.03 | 0.32 ± 0.02 | 0.32 ± 0.01 | 0.36 ± 0.01 | 0.31 ± 0.02 | 0.29 ± 0.01 | **0.35 ± 0.01** | 0.51 ± 0.07 | 0.59 ± 0.01 | 0.31 ± 0.01 |
| Multiple Models + Tri-Mentoring | 0.11 ± 0.01 | 0.09 ± 0.01 | 0.32 ± 0.01 | 0.48 ± 0.07 | 0.36 ± 0.02 | 0.32 ± 0.04 | 0.35 ± 0.01 | 0.39 ± 0.01 | 0.33 ± 0.01 | 0.29 ± 0.01 | **0.35 ± 0.01** | 0.49 ± 0.06 | 0.56 ± 0.01 | 0.36 ± 0.01 |
| MOBO | 0.11 ± 0.00 | 0.09 ± 0.01 | **0.30 ± 0.01** | 0.33 ± 0.01 | 0.33 ± 0.02 | 0.28 ± 0.00 | 0.31 ± 0.00 | 0.34 ± 0.00 | 0.35 ± 0.01 | **0.25 ± 0.02** | 0.38 ± 0.01 | 0.49 ± 0.01 | **0.55 ± 0.01** | N/A |
| MOBO-qParEGO | 0.11 ± 0.00 | 0.11 ± 0.00 | 0.37 ± 0.00 | 0.33 ± 0.01 | 0.37 ± 0.01 | 0.39 ± 0.00 | 0.45 ± 0.01 | 0.39 ± 0.00 | 0.38 ± 0.02 | 0.35 ± 0.11 | 0.37 ± 0.01 | 0.51 ± 0.00 | 0.59 ± 0.00 | N/A |
| MOBO-JES | N/A | N/A | N/A | N/A | N/A | N/A | N/A | N/A | N/A | N/A | N/A | N/A | N/A | N/A |
| ParetoFlow | **0.10 ± 0.02** | **0.06 ± 0.02** | 0.33 ± 0.00 | 0.38 ± 0.03 | **0.29 ± 0.00** | 0.33 ± 0.01 | 0.33 ± 0.02 | 0.37 ± 0.01 | 0.34 ± 0.02 | N/A | N/A | 0.65 ± 0.01 | 0.58 ± 0.00 | N/A |
| DOMOO (**ours**) | 0.12 ± 0.00 | 0.11 ± 0.00 | **0.30 ± 0.01** | 0.31 ± 0.01 | 0.34 ± 0.01 | 0.33 ± 0.06 | 0.31 ± 0.00 | 0.41 ± 0.04 | 0.66 ± 0.14 | 0.78 ± 0.12 | 0.62 ± 0.06 | **0.43 ± 0.03** | 0.59 ± 0.02 | **0.28 ± 0.01** |

*Table 28.* IGD$_{\text{offline}}$ results for MORL with 256 solutions and $50^{th}$ percentile evaluations. For each task, algorithms within one standard deviation of having the highest performance are **bolded**.

| Methods | MO-Swimmer | MO-Hopper |
|---|---|---|
| $\mathcal{D}$(best) | **0.43** | **0.8** |
| End-to-End | 0.81 ± 0.00 | 0.90 ± 0.00 |
| End-to-End + GradNorm | 0.77 ± 0.00 | 0.91 ± 0.00 |
| End-to-End + PcGrad | 0.78 ± 0.00 | 0.89 ± 0.00 |
| Multi Head | 0.71 ± 0.00 | 0.89 ± 0.00 |
| Multi Head + GradNorm | 0.82 ± 0.00 | 0.91 ± 0.00 |
| Multi Head + PcGrad | 0.77 ± 0.00 | 0.88 ± 0.00 |
| Multiple Models | 0.78 ± 0.00 | 0.90 ± 0.00 |
| Multiple Models + COMs | 0.66 ± 0.00 | 0.90 ± 0.00 |
| Multiple Models + RoMA | 0.76 ± 0.00 | 0.90 ± 0.00 |
| Multiple Models + IOM | 0.75 ± 0.00 | 0.85 ± 0.00 |
| Multiple Models + ICT | 0.74 ± 0.03 | 0.89 ± 0.04 |
| Multiple Models + Tri-Mentoring | 0.72 ± 0.05 | 0.91 ± 0.00 |
| MOBO | N/A | N/A |
| MOBO-qParEGO | N/A | N/A |
| MOBO-JES | N/A | N/A |
| ParetoFlow | 0.87 ± 0.18 | 0.93 ± 0.00 |
| DOMOO (**ours**) | 0.62 ± 0.04 | 0.81 ± 0.09 |

*Table 29.* IGD$_{\text{offline}}$ results for RE with 256 solutions and $50^{th}$ percentile evaluations. For each task, algorithms within one standard deviation of having the highest performance are **bolded**.

| Methods | RE21 | RE22 | RE23 | RE24 | RE25 | RE31 | RE32 | RE33 | RE34 | RE35 | RE36 | RE37 | MO-Portfolio |
|---|---|---|---|---|---|---|---|---|---|---|---|---|---|
| $\mathcal{D}$(best) | 0.56 | **0.00** | **0.00** | **0.00** | 0.03 | 0.34 | 0.02 | 0.04 | 0.34 | 0.09 | 0.69 | 0.65 | 0.47 |
| End-to-End | 0.45 ± 0.00 | 0.22 ± 0.01 | 0.04 ± 0.00 | 0.12 ± 0.10 | 0.10 ± 0.05 | 0.27 ± 0.00 | 0.09 ± 0.02 | 0.07 ± 0.03 | **0.30 ± 0.00** | 0.36 ± 0.05 | 0.41 ± 0.03 | 0.53 ± 0.01 | 0.56 ± 0.01 |
| End-to-End + GradNorm | 0.47 ± 0.02 | 0.15 ± 0.06 | 0.34 ± 0.34 | 0.66 ± 0.33 | 0.12 ± 0.09 | 0.68 ± 1.17 | 0.06 ± 0.02 | 0.15 ± 0.01 | 0.35 ± 0.02 | 0.34 ± 0.04 | 3.08 ± 0.00 | 0.53 ± 0.01 | 0.56 ± 0.01 |
| End-to-End + PcGrad | 0.45 ± 0.00 | 0.49 ± 0.63 | 0.03 ± 0.02 | 0.30 ± 0.08 | 0.07 ± 0.00 | 0.26 ± 0.04 | 0.11 ± 0.01 | 0.12 ± 0.04 | 0.31 ± 0.00 | 0.18 ± 0.03 | 0.44 ± 0.04 | 0.52 ± 0.00 | 0.56 ± 0.01 |
| Multi Head | 0.45 ± 0.00 | 0.18 ± 0.04 | 0.03 ± 0.01 | 0.11 ± 0.20 | 0.09 ± 0.05 | 0.27 ± 0.02 | 0.09 ± 0.02 | 0.08 ± 0.01 | **0.30 ± 0.00** | 0.19 ± 0.09 | **0.38 ± 0.02** | 0.52 ± 0.00 | 0.58 ± 0.02 |
| Multi Head + GradNorm | 0.47 ± 0.03 | 2.65 ± 3.53 | 0.79 ± 0.36 | 0.68 ± 0.19 | 0.45 ± 0.39 | 0.20 ± 0.02 | 0.05 ± 0.02 | 0.30 ± 0.16 | 0.33 ± 0.04 | 0.28 ± 0.12 | 0.96 ± 0.89 | 0.62 ± 0.17 | 0.65 ± 0.07 |
| Multi Head + PcGrad | 0.45 ± 0.00 | 10.21 ± 19.81 | 0.03 ± 0.02 | 0.84 ± 0.08 | 0.13 ± 0.09 | 0.21 ± 0.05 | 0.32 ± 0.26 | 0.13 ± 0.10 | **0.30 ± 0.00** | 0.13 ± 0.05 | 0.48 ± 0.02 | 0.52 ± 0.00 | 0.59 ± 0.02 |
| Multiple Models | 0.45 ± 0.00 | 0.08 ± 0.01 | 0.06 ± 0.03 | 0.04 ± 0.03 | 0.15 ± 0.05 | 0.26 ± 0.01 | 0.11 ± 0.00 | 0.08 ± 0.02 | **0.30 ± 0.00** | 0.15 ± 0.05 | **0.38 ± 0.02** | 0.52 ± 0.00 | 0.59 ± 0.03 |
| Multiple Models + COMs | 0.45 ± 0.00 | 0.10 ± 0.05 | 0.02 ± 0.01 | 0.16 ± 0.24 | 0.16 ± 0.06 | 0.20 ± 0.06 | 0.11 ± 0.01 | 0.11 ± 0.06 | 0.31 ± 0.02 | **0.06 ± 0.00** | 0.47 ± 0.02 | 0.54 ± 0.01 | 1.08 ± 0.20 |
| Multiple Models + RoMA | 0.48 ± 0.00 | 0.16 ± 0.13 | 0.21 ± 0.36 | 0.63 ± 0.35 | 0.14 ± 0.10 | 0.02 ± 0.00 | 0.03 ± 0.00 | 0.12 ± 0.04 | 0.33 ± 0.01 | 0.08 ± 0.00 | 0.70 ± 0.06 | 0.53 ± 0.00 | 0.58 ± 0.01 |
| Multiple Models + IOM | 0.45 ± 0.00 | 0.07 ± 0.06 | 0.01 ± 0.01 | **0.00 ± 0.00** | 0.02 ± 0.02 | 0.21 ± 0.04 | 0.08 ± 0.02 | 0.05 ± 0.00 | **0.30 ± 0.00** | 0.07 ± 0.00 | 0.42 ± 0.04 | 0.52 ± 0.00 | 0.60 ± 0.02 |
| Multiple Models + ICT | 0.45 ± 0.00 | 0.14 ± 0.07 | 0.08 ± 0.06 | 0.11 ± 0.11 | 0.06 ± 0.05 | 0.04 ± 0.02 | 0.08 ± 0.03 | 0.11 ± 0.04 | 0.31 ± 0.00 | 0.10 ± 0.02 | 0.43 ± 0.04 | 0.52 ± 0.00 | 0.58 ± 0.01 |
| Multiple Models + Tri-Mentoring | 0.45 ± 0.00 | 0.20 ± 0.02 | 0.23 ± 0.18 | 0.02 ± 0.04 | 0.11 ± 0.05 | 0.03 ± 0.01 | 0.09 ± 0.01 | 0.09 ± 0.00 | 0.31 ± 0.00 | 0.11 ± 0.07 | 0.70 ± 0.22 | 0.52 ± 0.00 | 0.57 ± 0.01 |
| MOBO | 0.59 ± 0.01 | 0.01 ± 0.00 | 0.26 ± 0.00 | 0.57 ± 0.04 | 0.03 ± 0.02 | 0.07 ± 0.00 | 0.03 ± 0.00 | 0.20 ± 0.02 | 0.53 ± 0.03 | 0.07 ± 0.00 | 0.54 ± 0.00 | 0.51 ± 0.00 | 1.96 ± 0.02 |
| MOBO-$q$ParEGO | 0.48 ± 0.01 | 0.01 ± 0.00 | 0.04 ± 0.06 | 0.63 ± 0.10 | **0.01 ± 0.00** | 0.06 ± 0.00 | 0.04 ± 0.00 | 0.05 ± 0.00 | 0.52 ± 0.01 | 0.08 ± 0.00 | 0.54 ± 0.00 | 0.52 ± 0.00 | 0.54 ± 0.02 |
| MOBO-JES | 0.46 ± 0.01 | 0.03 ± 0.00 | **0.00 ± 0.00** | **0.00 ± 0.00** | 0.12 ± 0.01 | N/A | N/A | 0.09 ± 0.02 | 0.37 ± 0.00 | 0.10 ± 0.00 | N/A | N/A | N/A |
| ParetoFlow | **0.37 ± 0.06** | 0.03 ± 0.02 | N/A | N/A | N/A | 0.07 ± 0.04 | **0.02 ± 0.00** | **0.04 ± 0.01** | 0.35 ± 0.07 | N/A | 0.56 ± 0.12 | **0.48 ± 0.06** | **0.42 ± 0.03** |
| DOMOO (ours) | 0.45 ± 0.00 | 0.07 ± 0.01 | 0.04 ± 0.00 | 0.03 ± 0.02 | 0.14 ± 0.04 | 0.22 ± 0.01 | 0.05 ± 0.01 | 0.07 ± 0.01 | **0.30 ± 0.00** | 0.10 ± 0.02 | 0.39 ± 0.02 | 0.52 ± 0.00 | 0.49 ± 0.08 |

*Table 30.* IGD$_{\text{offline}}$ results for scientific design with 256 solutions and $50^{th}$ percentile evaluations. For each task, algorithms within one standard deviation of having the highest performance are **bolded**.

| Methods | Molecule | Regex | RFP | ZINC |
|---|---|---|---|---|
| $\mathcal{D}$(best) | **0.84** | 1.05 | **0.39** | 0.2 |
| End-to-End | 1.23 ± 0.24 | 1.19 ± 0.00 | 0.41 ± 0.00 | 0.27 ± 0.01 |
| End-to-End + GradNorm | 1.44 ± 0.00 | 1.19 ± 0.00 | 0.40 ± 0.00 | 0.24 ± 0.01 |
| End-to-End + PcGrad | 1.20 ± 0.25 | 1.19 ± 0.00 | 0.40 ± 0.00 | 0.24 ± 0.01 |
| Multi Head | 1.42 ± 0.03 | 1.19 ± 0.00 | 0.40 ± 0.00 | 0.23 ± 0.02 |
| Multi Head + GradNorm | 1.15 ± 0.22 | 1.09 ± 0.05 | 0.40 ± 0.00 | 0.27 ± 0.00 |
| Multi Head + PcGrad | 1.31 ± 0.19 | 1.19 ± 0.00 | 0.42 ± 0.03 | 0.25 ± 0.02 |
| Multiple Models | 1.21 ± 0.23 | 1.19 ± 0.00 | 0.41 ± 0.00 | 0.28 ± 0.03 |
| Multiple Models + COMs | 0.97 ± 0.21 | 1.19 ± 0.00 | 0.40 ± 0.00 | 0.28 ± 0.02 |
| Multiple Models + RoMA | 1.22 ± 0.24 | 1.19 ± 0.00 | 0.40 ± 0.00 | 0.25 ± 0.01 |
| Multiple Models + IOM | 1.03 ± 0.20 | 1.19 ± 0.00 | 0.41 ± 0.00 | 0.30 ± 0.01 |
| Multiple Models + ICT | 0.98 ± 0.23 | 1.14 ± 0.07 | 0.42 ± 0.01 | 0.28 ± 0.02 |
| Multiple Models + Tri-Mentoring | 1.39 ± 0.03 | 1.19 ± 0.00 | 0.41 ± 0.01 | 0.28 ± 0.03 |
| MOBO | 1.44 ± 0.00 | 1.05 ± 0.00 | 0.40 ± 0.00 | 0.30 ± 0.01 |
| MOBO-$q$ParEGO | N/A | 1.05 ± 0.00 | 0.43 ± 0.01 | 0.18 ± 0.02 |
| MOBO-JES | N/A | N/A | N/A | N/A |
| ParetoFlow | 1.19 ± 0.19 | **0.87 ± 0.00** | N/A | 0.26 ± 0.08 |
| DOMOO (ours) | 1.29 ± 0.18 | 1.03 ± 0.06 | 0.42 ± 0.02 | **0.17 ± 0.01** |

## I. Results of Selection Indicators for Diversity

**About the Impact of the Selection Indicator on Diversity.** The core issue with HV-based selection in offline MOO arises from surrogate-induced spurious Pareto fronts in OOD regions: due to limited offline data, surrogates often extrapolate a much wider front than the true one. Candidates far beyond the true front typically correspond to poorly calibrated regions, i.e., they have low true quality and tend to cluster tightly in objective space. When using HV for selection, its marginal-volume mechanism tends to uniformly pick solutions along this wide spurious front. Consequently, HV frequently selects many tightly clustered, low-quality solutions in reality, harming true diversity despite acceptable surrogate-based scores. In contrast, IGD$_{\text{offline}}$ uses the offline Pareto front as reference and computes average distances to it. This inherently penalizes solutions deviating from reliable data regions, acting as a conservative filter against OOD artifacts where no active queries can correct surrogate errors.

The resulting solution distributions are shown in Figure 4. The results clearly demonstrate that the HV selection leads to a poorly distributed set with solutions clustered in a narrow region. The IGD$_{\text{offline}}$ selection produces a well-distributed front

that covers the entire spectrum of known trade-offs, underscoring its effectiveness in preserving diversity.

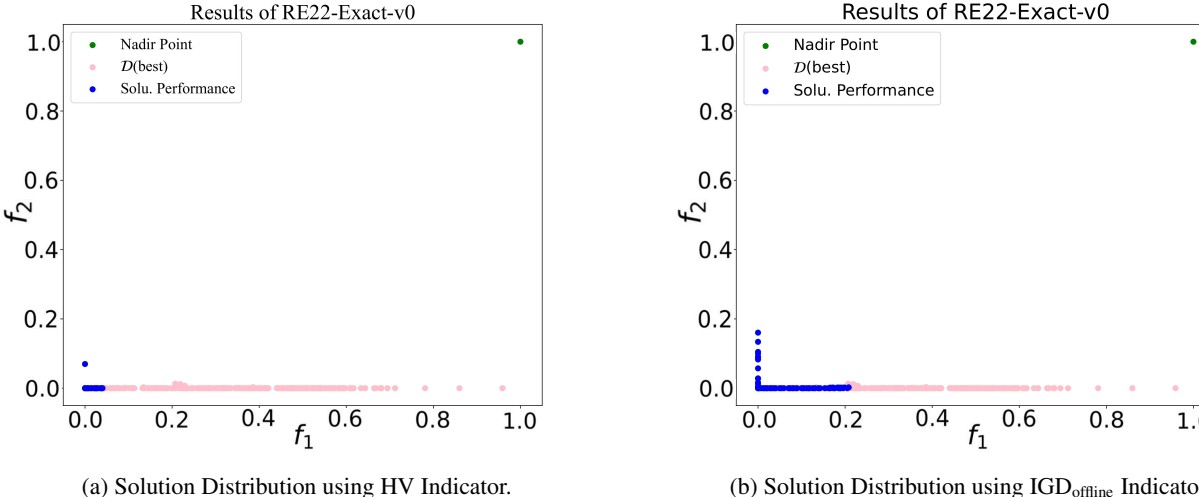

(a) Solution Distribution using HV Indicator.  (b) Solution Distribution using IGD$_{\text{offline}}$ Indicator.

*Figure 4.* Comparison of final solution sets selected by different indicators.

## J. Additional Details for Ablation Study

### J.1. Detailed Settings of Ablation Variants

In Section 5.3 of the main paper, we evaluate the contribution of each essential module in DOMOO by comparing the full version with five ablated variants. The detailed implementation settings for these variants are defined as follows:

• **Without Accumulative Risk Control (w.o. ARC):** In this version, we replace the accumulative risk control (as shown in Equation 2) with learning rate in gradient descent.

• **Without Nested Pareto Set Learning (w.o. NPSL):** In this version, we remove the "Preference Update" part and randomly sample $\boldsymbol{\lambda}$ from all valid preferences $\Lambda$ at the begin of each iteration.

• **Without Pareto Set Model Generation (w.o. PSMG):** In this version, we remove the candidate generation step of the Pareto set model $h_{\phi^*}$ and use the surrogate model alone to generate all candidate solutions.

• **Without Surrogate Model Generation (w.o. SMG):** In this version, we remove candidate generated by surrogate model and rely solely on the Pareto set model to generate all candidate solutions.

• **Without Diversity-Driven Solution Selection (w.o. DDSS):** In this version, we omit the proposed solution selection strategy and select all 256 solutions by HV. All other settings in the five versions are kept similar to the original version.

### J.2. Effectiveness of Diversity-Driven Selection Mechanism

**About the Effectiveness of Diversity-Driven Selection.** To demonstrate the necessity of our proposed DDSS, we compare the solution distributions with and without this mechanism on a representative benchmark task, as shown in Figure 5. Without DDSS, solution distribution shows a poorly diversified front along the $f_2$ axis, while our DDSS effectively produces a well-distributed Pareto front. This contrast highlights DDSS's crucial role in balancing diversity and convergence under OOD constraints.

## K. Hyper-Parameter Analysis

**About the Impact of Hyper-Parameters.** To explore the sensitivity of DOMOO to different hyper-parameters, we analyze the exploration steps in nested Pareto set learning $T_{\text{exp}}$ on three representative tasks, with results shown in in Tables 31 and 32. DOMOO is robust on continuous and sequence-based tasks, but shows higher sensitivity on discrete tasks, likely due to the difficulty of optimizing over high-cardinality categorical spaces. Nonetheless, performance remains stable when

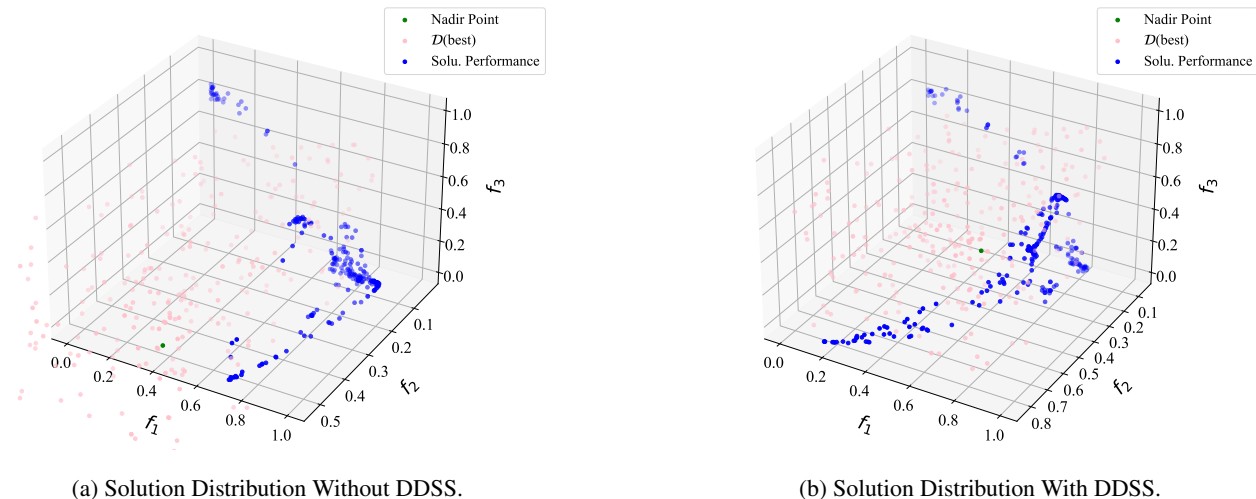

(a) Solution Distribution Without DDSS.        (b) Solution Distribution With DDSS.

*Figure 5.* Comparison of solution distributions with and without the DDSS mechanism.

$T_{\exp}$ is set within a reasonable range.

**About the $K$ in Diversity-Driven Solution Selection.** To examine the effect of the DDSS selection budget on performance, we analyze the maximum number of solutions selected by the $\text{IGD}_{\text{offline}}$-based stage before HV filling. As described in Section 4.3, DOMOO first selects at most 128 solutions from $\mathbf{X}_{\text{cand}}$ using $\text{IGD}_{\text{offline}}$, and then uses HV to fill the remaining slots to obtain 256 solutions for final evaluation. This maximum number therefore plays a key role in balancing diversity and convergence. We evaluate different settings of this hyper-parameter on several representative tasks, with results reported in Tables 33 and 34. The results show that DOMOO remains stable across a broad range of values, and setting the maximum number to 128 provides a good balance between convergence quality and front coverage.

**About the Robustness to Scaling Factor in the $\text{IGD}_{\text{offline}}$ Indicator.** As shown in Table 35, we further investigate the sensitivity of $\text{IGD}_{\text{offline}}$ to the scaling factor $\beta$. When $\beta$ is increased from 0.5 to 5.0, the average ranks of all methods exhibit only minor fluctuations, and their relative order remains largely unchanged. Within a reasonable range, the choice of the scaling value does not substantially affect the comparative evaluation results under $\text{IGD}_{\text{offline}}$, verifying the robustness of this indicator with respect to the scaling hyper-parameter. In addition, DOMOO consistently achieves the best average rank across all choices of $\beta$.

**About the Robustness of DOMOO to the Energy Model Risk-Ratio Hyper-parameter in Energy-Based Tasks.** As shown in Tables 36 and 37, we further analyze the sensitivity of DOMOO to the risk ratio used in the construction of the energy models across different tasks. When the risk ratio varies from 0.2 to 1.6, most tasks (e.g., `re24`, `re25`, `re34`, `dtlz4`) exhibit almost unchanged HV and $\text{IGD}_{\text{offline}}$, indicating very low sensitivity and strong robustness to this hyper-parameter. For tasks such as `in1kmop7` and `mo_hopper_v2`, the performance shows only mild and smooth variation without any abrupt degradation, suggesting controlled and predictable sensitivity rather than instability. Overall, these results demonstrate that DOMOO maintains stable performance under a wide range of risk ratios, verifying the robustness of the algorithm with respect to the risk-ratio hyper-parameter in energy-based tasks.

## L. How DOMOO Performance Varies with Different Training Set Sizes

To further examine the robustness of DOMOO with respect to the amount of available training data, we conduct an additional sensitivity analysis in which the dataset is randomly subsampled to 25%, 50%, 75%, and 100% of its original size. As reported in Tables 38 and 39, DOMOO maintains highly stable performance across all data scales. For most tasks (e.g., `in1kmop7`, `regex`, `re24`), both HV and $\text{IGD}_{\text{offline}}$ vary only marginally as the amount of training data changes, indicating that the method does not rely on large datasets to achieve strong performance.

Interestingly, the HV metric for `mo_hopper_v2` exhibits a slight downward trend as data size increases, while its $\text{IGD}_{\text{offline}}$ values remain consistent across all subsampling ratios. This suggests that the convergence behavior of DOMOO is not significantly affected by the available data volume. Overall, these results demonstrate that DOMOO is robust and

*Table 31.* HV results under different $T_{\exp}/T$ values.

| Tasks | 0% | 12.5% | 25% | 37.5% | 50% | 62.5% | 75% | 87.5% | 100% |
|---|---|---|---|---|---|---|---|---|---|
| dtlz3 | 10.6026 | 10.5864 | 10.6024 | 10.5850 | 10.5912 | 10.5857 | 10.6026 | 10.5850 | 10.5946 |
| in1kmop7 | 4.4498 | 4.4221 | 4.4399 | 4.4068 | 4.4693 | 4.3824 | 4.4477 | 4.3824 | 4.4300 |
| re21 | 4.6001 | 4.5974 | 4.6001 | 4.5976 | 4.6000 | 4.5974 | 4.6000 | 4.5976 | 4.6000 |
| zdt1 | 4.8207 | 4.8198 | 4.8191 | 4.8184 | 4.8205 | 4.8186 | 4.8191 | 4.8199 | 4.8195 |
| c10mop1 | 4.7270 | 4.7574 | 4.7455 | 4.7484 | 4.7245 | 4.7553 | 4.7269 | 4.7579 | 4.7473 |
| re31 | 10.6481 | 10.6481 | 10.6481 | 10.6481 | 10.6481 | 10.6481 | 10.6481 | 10.6481 | 10.6481 |
| vlmop1 | 0.3168 | 0.3168 | 0.3168 | 0.3168 | 0.3168 | 0.3168 | 0.3168 | 0.3168 | 0.3168 |

*Table 32.* $\text{IGD}_{\text{offline}}$ results under different $T_{\exp}/T$ values.

| Tasks | 0% | 12.5% | 25% | 37.5% | 50% | 62.5% | 75% | 87.5% | 100% |
|---|---|---|---|---|---|---|---|---|---|
| dtlz3 | 0.1662 | 0.1893 | 0.1690 | 0.1898 | 0.1855 | 0.1898 | 0.1736 | 0.1896 | 0.1784 |
| in1kmop7 | 0.3385 | 0.3413 | 0.3465 | 0.3560 | 0.3470 | 0.3663 | 0.3436 | 0.3663 | 0.3458 |
| re21 | 0.4449 | 0.4454 | 0.4449 | 0.4453 | 0.4449 | 0.4454 | 0.4449 | 0.4453 | 0.4449 |
| zdt1 | 0.1399 | 0.1399 | 0.1388 | 0.1412 | 0.1401 | 0.1410 | 0.1409 | 0.1404 | 0.1400 |
| c10mop1 | 0.1064 | 0.1067 | 0.1105 | 0.1151 | 0.1167 | 0.1097 | 0.1092 | 0.1084 | 0.1094 |
| re31 | 0.0278 | 0.0252 | 0.0296 | 0.0313 | 0.0309 | 0.0363 | 0.0296 | 0.0338 | 0.0364 |
| vlmop1 | 0.0289 | 0.0290 | 0.0291 | 0.0289 | 0.0292 | 0.0289 | 0.0289 | 0.0290 | 0.0289 |

sample-efficient, and its effectiveness persists even when the training data is substantially reduced.

To further investigate the performance of DOMOO under varying levels of OOD severity, we prune the dataset by removing some high-quality data to simulate different OOD levels. The experimental results are shown in Tables 40–45. The experimental results show that DOMOO can effectively balance diversity and quality across different OOD levels. Notably, even under severe OOD conditions (Tables 40 and 41), DOMOO still maintains strong performance.

## M. Performance of Online Pareto Set Learning Methods under Offline Optimization

As shown in Table 46, online Pareto set learning methods, namely EPS (Ye et al., 2024) and PSL-MOBO (Lin et al., 2022), are not well-suited for offline optimization. When applied to offline optimization, they often encounter severe out-of-distribution (OOD) issues (Lu et al., 2023; Brookes et al., 2019) , i.e., they yield solutions that are overconfident on the surrogate model, leading to significant deterioration or even invalidation of the solutions.

*Table 33.* HV results under different maximum numbers.

| Tasks | 0 | 32 | 64 | 128 | 160 | 192 | 224 | 256 |
|---|---|---|---|---|---|---|---|---|
| re22 | 4.8399 | 4.8399 | 4.8399 | 4.8399 | 4.8399 | 4.8399 | 4.8399 | 4.8399 |
| dtlz1 | 10.6462 | 10.6457 | 10.6462 | 10.6460 | 10.6456 | 10.6456 | 10.6456 | 10.6456 |
| in1kmop1 | 4.5600 | 4.5844 | 4.6191 | 4.6191 | 4.6191 | 4.6191 | 4.6191 | 4.6191 |
| in1kmop2 | 4.3242 | 4.4885 | 4.4885 | 4.4987 | 4.4987 | 4.4987 | 4.4987 | 4.4987 |
| in1kmop3 | 9.6707 | 9.7966 | 9.7967 | 9.8696 | 9.8696 | 9.8696 | 9.8696 | 9.8696 |

*Table 34.* $\mathbf{IGD_{offline}}$ results under different maximum numbers.

| Tasks | 0 | 32 | 64 | 128 | 160 | 192 | 224 | 256 |
|---|---|---|---|---|---|---|---|---|
| re22 | 0.0599 | 0.0457 | 0.0457 | 0.0457 | 0.0457 | 0.0457 | 0.0457 | 0.0457 |
| dtlz1 | 0.1681 | 0.1687 | 0.1675 | 0.1687 | 0.1696 | 0.1696 | 0.1696 | 0.1696 |
| in1kmop1 | 0.2406 | 0.2450 | 0.2328 | 0.2328 | 0.2328 | 0.2328 | 0.2328 | 0.2328 |
| in1kmop2 | 0.3089 | 0.3149 | 0.3149 | 0.3103 | 0.3103 | 0.3103 | 0.3103 | 0.3103 |
| in1kmop | 0.3712 | 0.3678 | 0.3678 | 0.3618 | 0.3618 | 0.3618 | 0.3618 | 0.3618 |

*Table 35.* Comparison of average $IGD_{offline}$ ranks under different $\beta$.

| Methods | 0.5 | 1.0 | 1.5 | 2.0 | 2.5 | 3.0 | 3.5 | 4.0 | 4.5 | 5.0 |
|---|---|---|---|---|---|---|---|---|---|---|
| End2End + GradNorm | 9.93 | 9.95 | 9.69 | 9.45 | 9.28 | 9.19 | 9.09 | 9.03 | 9.10 | 9.07 |
| End2End + PcGrad | 7.47 | 7.04 | 7.07 | 7.04 | 7.08 | 7.01 | 6.98 | 6.94 | 6.98 | 7.00 |
| End2End + Vallina | 6.96 | 6.42 | 6.23 | 6.23 | 6.32 | 6.40 | 6.45 | 6.44 | 6.44 | 6.49 |
| MOBO + JES | 9.74 | 10.46 | 10.55 | 10.36 | 10.45 | 10.35 | 10.52 | 10.51 | 10.61 | 10.62 |
| MOBO + ParEGO | 7.36 | 8.35 | 8.65 | 8.67 | 8.70 | 8.68 | 8.66 | 8.67 | 8.61 | 8.57 |
| MOBO + Vallina | 6.31 | 6.89 | 7.26 | 7.00 | 6.89 | 6.74 | 6.82 | 6.78 | 6.89 | 6.86 |
| MultiHead + GradNorm | 10.15 | 10.14 | 9.96 | 9.83 | 9.76 | 9.72 | 9.70 | 9.67 | 9.67 | 9.66 |
| MultiHead + PcGrad | 7.36 | 7.21 | 7.35 | 7.39 | 7.37 | 7.37 | 7.40 | 7.39 | 7.41 | 7.33 |
| MultiHead + Vallina | 6.89 | 6.52 | 6.43 | 6.29 | 6.21 | 6.28 | 6.27 | 6.30 | 6.34 | 6.31 |
| MultipleModels + COM | 6.80 | 7.72 | 7.93 | 8.01 | 8.09 | 8.14 | 8.08 | 8.13 | 8.16 | 8.13 |
| MultipleModels + ICT | 7.55 | 7.82 | 7.83 | 7.92 | 7.90 | 7.89 | 7.84 | 7.81 | 7.79 | 7.75 |
| MultipleModels + IOM | 5.56 | 5.96 | 6.33 | 6.41 | 6.46 | 6.47 | 6.53 | 6.47 | 6.48 | 6.46 |
| MultipleModels + RoMA | 7.98 | 7.72 | 7.59 | 7.79 | 7.85 | 7.94 | 8.05 | 8.11 | 8.12 | 8.19 |
| MultipleModels + TriMentoring | 8.29 | 8.56 | 8.30 | 8.35 | 8.43 | 8.34 | 8.31 | 8.30 | 8.24 | 8.23 |
| MultipleModels + Vallina | 7.23 | 6.52 | 6.46 | 6.52 | 6.45 | 6.48 | 6.46 | 6.54 | 6.48 | 6.53 |
| DOMOO | **6.19** | **5.66** | **5.62** | **5.79** | **5.87** | **5.95** | **5.97** | **6.03** | **5.91** | **5.99** |

*Table 36.* Comparison of average HV ranks across different energy model risk ratios in Off-MOO-Bench.

| Tasks | 0.2 | 0.4 | 0.6 | 0.8 | 1.0 | 1.2 | 1.4 | 1.6 |
|---|---|---|---|---|---|---|---|---|
| dtlz4 | 9.617±0.083 | 9.742±0.046 | 9.729±0.050 | 9.740±0.045 | 9.742±0.046 | 9.437±0.005 | 9.712±0.037 | 9.718±0.048 |
| in1kmop7 | 4.519±0.002 | 4.501±0.002 | 4.434±0.004 | 4.432±0.002 | 4.444±0.002 | 4.429±0.000 | 4.509±0.003 | 4.428±0.001 |
| mo_hopper | 6.449±0.035 | 6.396±0.020 | 6.474±0.126 | 6.448±0.069 | 6.396±0.071 | 6.358±0.057 | 6.396±0.055 | 6.440±0.024 |
| re24 | 4.835±0.000 | 4.835±0.000 | 4.835±0.000 | 4.835±0.000 | 4.835±0.000 | 4.835±0.000 | 4.835±0.000 | 4.835±0.000 |
| re25 | 4.840±0.000 | 4.840±0.000 | 4.840±0.000 | 4.840±0.000 | 4.840±0.000 | 4.840±0.000 | 4.840±0.000 | 4.840±0.000 |
| re34 | 10.122±0.000 | 10.122±0.000 | 10.122±0.000 | 10.122±0.000 | 10.122±0.000 | 10.122±0.000 | 10.122±0.000 | 10.122±0.000 |
| regex | 6.189±0.138 | 6.198±0.147 | 6.034±0.006 | 6.034±0.006 | 6.034±0.006 | 6.034±0.006 | 6.254±0.090 | 6.254±0.090 |
| vlmop1 | 0.317±0.000 | 0.317±0.000 | 0.317±0.000 | 0.317±0.000 | 0.317±0.000 | 0.317±0.000 | 0.317±0.000 | 0.317±0.000 |

*Table 37.* Comparison of average IGD$_{\text{offline}}$ ranks across different energy model risk ratios in Off-MOO-Bench.

| Tasks | 0.2 | 0.4 | 0.6 | 0.8 | 1.0 | 1.2 | 1.4 | 1.6 |
|---|---|---|---|---|---|---|---|---|
| dtlz4 | 0.640±0.035 | 0.739±0.001 | 0.739±0.001 | 0.738±0.001 | 0.738±0.001 | 0.744±0.002 | 0.738±0.002 | 0.743±0.002 |
| in1kmop7 | 0.361±0.000 | 0.372±0.000 | 0.361±0.001 | 0.365±0.001 | 0.357±0.001 | 0.363±0.000 | 0.366±0.000 | 0.364±0.001 |
| mo_hopper | 0.565±0.005 | 0.559±0.005 | 0.495±0.015 | 0.569±0.002 | 0.581±0.005 | 0.610±0.002 | 0.603±0.000 | 0.600±0.000 |
| re24 | 0.016±0.000 | 0.016±0.000 | 0.024±0.000 | 0.024±0.000 | 0.024±0.000 | 0.024±0.000 | 0.023±0.001 | 0.024±0.000 |
| re25 | 0.083±0.000 | 0.091±0.001 | 0.091±0.001 | 0.091±0.001 | 0.091±0.001 | 0.091±0.001 | 0.091±0.001 | 0.091±0.001 |
| re34 | 0.297±0.000 | 0.297±0.000 | 0.297±0.000 | 0.297±0.000 | 0.297±0.000 | 0.297±0.000 | 0.297±0.000 | 0.297±0.000 |
| regex | 0.896±0.002 | 0.893±0.003 | 0.897±0.000 | 0.897±0.000 | 0.897±0.000 | 0.897±0.000 | 0.897±0.000 | 0.897±0.000 |
| vlmop1 | 0.029±0.000 | 0.030±0.000 | 0.032±0.000 | 0.030±0.000 | 0.030±0.000 | 0.031±0.000 | 0.030±0.000 | 0.032±0.000 |

*Table 38.* Comparison of average HV ranks across different tasks in Off-MOO-Bench under varying training dataset sizes (25%, 50%, 75%, and 100% of the full training data).

| Tasks | 25% | 50% | 75% | 100% |
|---|---|---|---|---|
| in1kmop7 | 4.458±0.003 | 4.414±0.012 | 4.486±0.004 | 4.480±0.080 |
| mo_hopper_v2 | 5.168±0.212 | 5.451±1.100 | 4.881±0.394 | 6.530±0.240 |
| re24 | 4.682±0.046 | 4.789±0.009 | 4.749±0.022 | 4.840±0.000 |
| regex | 6.383±0.115 | 6.440±0.034 | 6.449±0.119 | 6.520±0.110 |

*Table 39.* Comparison of average IGD$_{\text{offline}}$ ranks across different tasks in Off-MOO-Bench under varying training dataset sizes (25%, 50%, 75%, and 100% of the full training data).

| Tasks | 25% | 50% | 75% | 100% |
|---|---|---|---|---|
| in1kmop7 | 0.357±0.000 | 0.389±0.001 | 0.403±0.001 | 0.380±0.030 |
| mo_hopper_v2 | 0.845±0.009 | 0.785±0.041 | 0.921±0.022 | 0.580±0.070 |
| re24 | 0.084±0.012 | 0.034±0.003 | 0.049±0.049 | 0.010±0.020 |
| regex | 0.878±0.000 | 0.873±0.000 | 0.866±0.000 | 0.900±0.010 |

*Table 40.* Results on the subset of data with quality scores between the 0th and 50th percentiles. HV values are reported and higher HV indicates better performance.

| Methods | DTLZ1 | DTLZ3 | IN1KMOP7 | MO_HOPPER_V2 | OMNITEST | RE24 | RE32 | RE35 | REGEX | VLMOP3 |
|---|---|---|---|---|---|---|---|---|---|---|
| End2End | 10.64 | 10.61 | 3.60 | 5.82 | 4.57 | 4.49 | 10.64 | 10.57 | 3.98 | 45.65 |
| MultiHead | 10.64 | 10.50 | 3.92 | 5.45 | 4.42 | 3.25 | 10.61 | 10.58 | 3.83 | 38.74 |
| MultipleModels | 10.64 | 10.61 | 3.74 | **5.95** | 4.64 | 4.16 | 10.64 | 10.57 | 3.87 | 45.62 |
| DOMOO | **10.64** | **10.63** | **4.27** | 4.95 | **4.66** | **4.73** | 10.64 | 10.59 | 5.58 | **45.88** |

*Table 41.* Results on the subset of data with quality scores between the 0th and 50th percentiles. IGD$_{\text{offline}}$ are reported and lower IGD$_{\text{offline}}$ indicates better performance.

| Methods | DTLZ1 | DTLZ3 | IN1KMOP7 | MO_HOPPER_V2 | OMNITEST | RE24 | RE32 | RE35 | REGEX | VLMOP3 |
|---|---|---|---|---|---|---|---|---|---|---|
| End2End | 0.18 | 0.16 | 0.57 | 0.78 | **0.28** | **0.12** | 0.03 | **0.08** | 1.07 | 0.07 |
| MultiHead | 0.17 | 0.15 | 0.54 | 0.76 | 0.30 | 0.61 | 0.05 | 0.18 | 1.08 | 0.32 |
| MultipleModels | 0.17 | 0.14 | 0.53 | 0.76 | 0.29 | 0.25 | 0.04 | 0.14 | 1.08 | 0.07 |
| DOMOO | **0.12** | **0.12** | **0.47** | **0.69** | 0.38 | 0.20 | **0.03** | 0.10 | **0.89** | **0.03** |

*Table 42.* Results on the subset of data with quality scores between the 0th and 75th percentiles. HV values are reported and higher HV indicates better performance.

| Methods | DTLZ1 | DTLZ3 | IN1KMOP7 | MO_HOPPER_V2 | OMNITEST | RE24 | RE32 | RE35 | REGEX | VLMOP3 |
|---|---|---|---|---|---|---|---|---|---|---|
| End2End | 10.64 | 10.54 | 3.76 | 5.54 | 4.60 | 4.48 | 10.64 | 10.58 | 3.98 | 45.85 |
| MultiHead | 10.64 | 10.25 | 3.99 | 4.82 | 4.35 | 2.78 | 10.60 | 10.56 | 3.54 | 41.00 |
| MultipleModels | 10.64 | 10.41 | 3.67 | **5.90** | 4.40 | 3.89 | 10.64 | 10.58 | 3.76 | 44.79 |
| DOMOO | **10.64** | **10.61** | **4.33** | 5.34 | **4.62** | **4.69** | 10.65 | 10.58 | **4.77** | **45.52** |

*Table 43.* Results on the subset of data with quality scores between the 0th and 75th percentiles. IGD$_{offline}$ are reported and lower IGD$_{offline}$ indicates better performance.

| Methods | DTLZ1 | DTLZ3 | IN1KMOP7 | MO_HOPPER_V2 | OMNITEST | RE24 | RE32 | RE35 | REGEX | VLMOP3 |
|---|---|---|---|---|---|---|---|---|---|---|
| End2End | 0.17 | 0.19 | 0.53 | 0.78 | 0.29 | **0.13** | 0.03 | 0.10 | 1.06 | 0.07 |
| MultiHead | 0.17 | 0.21 | **0.46** | 0.90 | 0.33 | 0.83 | 0.04 | 0.25 | 1.07 | 0.27 |
| MultipleModels | 0.17 | 0.18 | 0.56 | **0.63** | 0.33 | 0.37 | 0.03 | 0.09 | 1.09 | 0.08 |
| DOMOO | **0.15** | **0.14** | 0.50 | 0.72 | **0.29** | 0.21 | **0.03** | **0.08** | **0.90** | **0.04** |

*Table 44.* Results on the full dataset (quality scores from 0th to 100th percentile). HV values are reported and higher HV indicates better performance.

| Methods | DTLZ1 | DTLZ3 | IN1KMOP7 | MO_HOPPER_V2 | OMNITEST | RE24 | RE32 | RE35 | REGEX | VLMOP3 |
|---|---|---|---|---|---|---|---|---|---|---|
| End2End | 10.64 | **10.58** | 3.67 | **5.89** | **4.68** | 4.45 | 10.64 | 10.57 | 3.80 | **45.70** |
| MultiHead | 10.64 | 10.41 | 4.14 | 5.41 | 4.67 | 2.85 | 10.64 | 10.50 | 3.98 | 43.36 |
| MultipleModels | 10.64 | 10.57 | 3.60 | 5.65 | 4.10 | 4.05 | 10.64 | 10.58 | 3.98 | 44.20 |
| DOMOO | **10.65** | 10.46 | **4.25** | 5.32 | 4.63 | **4.71** | 10.64 | 10.58 | **6.06** | 44.91 |

*Table 45.* Results on the full dataset (quality scores from 0th to 100th percentile). IGD$_{offline}$ are reported and lower IGD$_{offline}$ indicates better performance.

| Methods | DTLZ1 | DTLZ3 | IN1KMOP7 | MO_HOPPER_V2 | OMNITEST | RE24 | RE32 | RE35 | REGEX | VLMOP3 |
|---|---|---|---|---|---|---|---|---|---|---|
| End2End | 0.17 | 0.17 | 0.55 | **0.64** | 0.26 | **0.14** | 0.04 | 0.11 | 1.08 | 0.13 |
| MultiHead | 0.17 | 0.21 | **0.43** | 0.78 | 0.25 | 0.79 | 0.04 | 0.27 | 1.06 | 0.19 |
| MultipleModels | 0.17 | **0.16** | 0.58 | 0.73 | 0.41 | 0.30 | 0.03 | 0.08 | 1.06 | 0.09 |
| DOMOO | **0.15** | 0.19 | 0.51 | 0.72 | 0.28 | 0.21 | **0.01** | **0.07** | **0.91** | **0.05** |

*Table 46.* Hypervolume results of online Pareto set learning methods under the offline optimization.

| Methods | DTLZ1 | DTLZ3 | DTLZ4 | DTLZ5 | DTLZ6 | DTLZ7 | MO-Hopper | MO-Swimmer | OmniTest | MO-Portfolio | RE21 | RE22 | RE23 | RE24 | RE25 | RE31 |
|---|---|---|---|---|---|---|---|---|---|---|---|---|---|---|---|---|
| $\mathcal{D}_{best}$ | **10.6** | **10** | **10.76** | **9.35** | **8.88** | 8.56 | **5.67** | **3.64** | **4.53** | **4.24** | 4.1 | **4.78** | **4.75** | 4.6 | 4.79 | **10.6** |
| EPS | 4.81 | N/A | N/A | N/A | N/A | N/A | 4.75 | 2.086 | 2.003 | 1.004 | 4.182 | 2.643 | 2.583 | **10.639** | **10.599** | N/A |
| PSL-MOBO | 7.15 | 6.84 | 8.77 | 7.47 | 8.87 | **10.36** | 4.75 | 3.086 | 3.754 | N/A | **4.84** | 2.63 | 2.84 | 0.84 | 0.84 | 9.00 |

