# OpenReview forum: "Diversity-Driven Offline Multi-Objective Optimization via Nested Pareto Set Learning"
_ICML.cc/2026/Conference — ICML 2026 regular_

### Official Review · Reviewer_e45Y · 2026-03-07

**Soundness:** 3
**Presentation:** 3
**Significance:** 2
**Originality:** 3
**Overall Recommendation:** 4
**Confidence:** 4

**Summary:**

This paper studies offline multi-objective optimization (MOO), where optimization must be performed using a fixed dataset without access to true objective functions, which introduces out-of-distribution (OOD) errors in surrogate models. To address this issue, the authors propose DOMOO, a framework that integrates risk control, nested Pareto set learning, and a diversity-driven selection strategy with a new indicator (IGDoffline) to obtain diverse and high-quality solutions.

**Compliance With Llm Reviewing Policy:**

Affirmed.

**Ethical Review Flag:**

Flag this paper for an ethics review.

**Key Questions For Authors:**

1. The proposed framework integrates several existing techniques such as surrogate modeling, Pareto set learning, and diversity-based selection. Could the authors clarify the key methodological novelty of DOMOO and explain how it fundamentally differs from existing offline MOO approaches?

2. The experimental results show relatively modest improvements over baseline methods. Could the authors provide additional analysis, such as more comprehensive comparisons or ablation studies, to better demonstrate the advantages of the proposed method?

3. While the paper includes benchmark experiments, the discussion of real-world applications is limited. Could the authors elaborate on potential practical scenarios where DOMOO could be effectively applied and discuss its deployment feasibility in real-world settings?

**Limitations:**

The method is not equally convincing across all task types. The paper itself notes weaker performance on highly discrete sparse tasks with high-dimensional one-hot encodings. That limitation is important, because it means the main message should be stated more carefully than the current version does.

**Strengths And Weaknesses:**

1. The paper is generally well organized and clearly written. The motivation, methodology, and experimental sections are structured logically and are easy to follow.

2. The overall novelty of the work appears somewhat limited. Many components of the framework build upon existing ideas in surrogate modeling, Pareto set learning, and diversity-based selection, making the methodological contribution somewhat incremental.

3. The experimental results do not clearly demonstrate a strong advantage of the proposed approach. The reported improvements over baseline methods appear relatively modest, and the superiority of the method is not sufficiently convincing.

4. The paper would benefit from stronger discussion of real-world applications. Although benchmarks are included, the practical impact and deployment scenarios of the proposed approach are not sufficiently elaborated.

---

> ### Author Rebuttal · Authors · 2026-03-31
>
> We really appreciate Reviewer e45Y for the positive comments on our paper. We address the raised concerns below.
>
> **[W1] Clarification of the methodological novelty of DOMOO.**
>
> We agree it is crucial to clearly clarifying the methodological novelty.
>
> The key novelty of DOMOO lies not in individual components, but in how they are **jointly formulated and optimized in a unified framework**, which fundamentally differs from existing offline MOO approaches. Specifically, our contributions can be summarized in two aspects:
>
> - **Pareto set learning with adaptive preference optimization**
> We propose a nested optimization framework where preferences are **adaptively updated to explore different regions** of the Pareto front. The inner loop performs preference exploration, encouraging coverage of regions where the Pareto set model has insufficient exploration and improving diversity, while the outer loop optimizes the Pareto set model under OOD risk control to enhance solution quality. This joint design of adaptive preference learning + risk-aware model optimization is, to our knowledge, **novel in offline MOO and enables an effective diversity–quality trade-off**.
>
> - **A tailored diversity indicator for offline settings ($\text{IGD}_\text{offline}$)**
> We propose $\text{IGD}_\text{offline}$, which replaces the unavailable true Pareto front with a shifted and normalized offline reference, allowing a more reliable evaluation of diversity in offline scenarios. This metric also directly guides our diversity-driven selection strategy.
>
> Compared with existing methods, DOMOO differs in several fundamental aspects. For example, methods such as ParetoFlow rely on fixed preference sampling (e.g., Das–Dennis) and evolutionary refinement, without adaptive preference learning or explicit OOD risk control. Preference-guided diffusion methods improve diversity via heuristics such as crowding distance, but their generation process is unidirectional and does not involve learning or updating preferences, **nor do they explicitly address OOD issues**.
>
> In contrast, DOMOO introduces a nested optimization framework that jointly adapts preferences and optimizes the Pareto set model under OOD risk, which is the core methodological difference.
>
> **[W2] Concerns about the strength of empirical improvements.**
>
> While the improvements over some compared methods may appear modest in certain individual tasks, we would like to emphasize that **DOMOO achieves consistently strong performance across a wide range of benchmarks**, including both synthetic functions and real-world tasks. In particular, it attains the best average ranking across tasks, indicating superior overall performance and stability. We believe this is more meaningful than improvements on a single dataset, as offline MOO requires robustness across diverse problem settings.
>
> We have already included comprehensive ablation studies in Section 5.3, which demonstrate the necessity and contribution of each component. In addition, Appendices K and L provide extensive sensitivity analyses under different hyperparameters, varying training data sizes, and manually constructed OOD conditions, further validating the robustness of DOMOO.
>
> To further strengthen the empirical evaluation, we have **additionally compared our method with a recent work (PGD [1])**. The results further support the advantage of our method and we will include these additional comparisons in the final version of the paper.  **All the additional experiments are shown `in this link`** ([https://anonymous.4open.science/r/DOMOO-0388/rebuttal_table/To_Reviewer_e45Y.md](https://anonymous.4open.science/r/DOMOO-0388/rebuttal_table/To_Reviewer_e45Y.md)).
>
> **[W3] Discussion on real-world applicability and deployment feasibility.**
>
> Our method is designed for **offline multi-objective optimization scenarios** where evaluating true objectives is **extremely expensive or infeasible**, which are common in real-world applications. In these settings, only historical datasets are available, and DOMOO can be directly applied to generate a diverse set of high-quality candidate solutions without requiring additional evaluations, making it suitable for practical deployment.
>
> In our experiments, we have already evaluated DOMOO on a wide range of real-world benchmarks, including those adopted by related works such as PGD [1] and ParetoFlow [2], which cover representative application domains.
>
> If there are additional real-world datasets or application scenarios that we have not covered, we would greatly appreciate your suggestions and are happy to further evaluate our method, as this would help strengthen the practical relevance of our work.
>
> [1] Preference-Guided Diffusion for Multi-Objective Offline Optimization[C]. The 39th Annual Conference on Neural Information Processing Systems.
>
> [2] ParetoFlow: Guided Flows in Multi-Objective Optimization[C]. The 13th International Conference on Learning Representations.

---

> > ### Author Rebuttal · Reviewer_e45Y · 2026-04-04
> >
> > Thank you for your response and the effort you have made to address the reviewers’ comments.
> >
> > However, I believe that the current score appropriately reflects the overall quality and contribution level of the manuscript.
> >
> > Therefore, I will maintain my original score.

---

> > > ### Author Response · Authors · 2026-04-05
> > >
> > > Thank you so much for your positive feedback! We deeply appreciate the time and effort you invested in reviewing our paper. Your constructive comments have been incredibly helpful in improving the quality and clarity of our manuscript. Thank you again for your support!
> > >
> > > **We would like to restate the main contributions of this work:**
> > >
> > > * **Addressing diversity degradation under OOD generalization in offline MOO**: Prior offline multi-objective optimization methods largely overlook how OOD generalization errors of surrogate models can bias solutions toward extreme regions, leading to a loss of diversity. We explicitly identify and study this issue as a core challenge in offline MOO.
> > > * **Risk-aware offline MOO framework**: We propose DOMOO, which explicitly models OOD uncertainty via accumulative risk control and integrates it into the optimization process.
> > > * **Bi-level Pareto set learning with adaptive preferences**: Our nested PSL framework jointly learns preference and solution mappings, enabling better adaptation to diverse Pareto front geometries.
> > > * **Diversity-driven solution selection for offline settings**: We introduce IGD_offline and a corresponding selection strategy to balance convergence and diversity under surrogate uncertainty.

---

### Official Review · Reviewer_vubi · 2026-03-10

**Soundness:** 3
**Presentation:** 3
**Significance:** 3
**Originality:** 3
**Overall Recommendation:** 4
**Confidence:** 3

**Summary:**

This paper focuses on the out-of-distribution (OOD) issue in offline multi-objective optimization. The authors propose a framework called DOMOO, which includes an accumulative risk control module, a nested Pareto set learning strategy, and a diversity-driven selection strategy. Experimental results on both synthetic and real-world benchmarks demonstrate the superiority of the proposed method.

**Compliance With Llm Reviewing Policy:**

Affirmed.

**Final Justification:**

The rebuttal has addressed my main concerns. The clarification regarding the risk control mechanism improves the overall clarity of the paper. Therefore, I keep my original positive evaluation of the paper (weak accept).

**Key Questions For Authors:**

1. Could the authors clarify the inconsistency between Eq. (2) and the accompanying explanation regarding the preference gradient update phase?

2. How would the performance of the compared algorithms change under $\text{IGD}_{\text{offline}}$ if objective normalization were applied (e.g., normalizing each objective based on its range in the offline dataset)?

3. In DOMOO, the energy model appears to be mainly used within the nested Pareto set learning module. However, risk control may also be important during the solution selection stage. Could the authors clarify what the risk distribution of the candidate solutions looks like after the nested Pareto set learning stage? In particular, if some candidate solutions are still associated with high risk, would it be beneficial or necessary to filter such solutions before the final solution selection stage?

If the authors can clearly address the above questions and provide sufficient clarification, I would be willing to reconsider my evaluation.

**Limitations:**

yes

**Strengths And Weaknesses:**

**Strengths:**
1. The paper addresses an important issue (the OOD problem) in offline multi-objective optimization, which may inspire further research in this direction.
2. The paper is well organized and generally easy to follow.
3. The proposed method for handling the OOD issue appears technically sound, although a few details are not clearly explained.
4. The experimental evaluation is comprehensive, and several ablation studies are conducted to analyze the effectiveness of different components of the proposed method.

**Weaknesses:**
1. The inner loop of preference updating consists of three phases. Among them, the preference gradient update phase is somewhat confusing. The paper states that the model is guided to focus on regions where its performance is lacking. However, according to Eq. (2), the preferences appear to be updated in a way that minimizes the scalarization function, which corresponds to moving toward regions where the current performance is already good. Consequently, in the outer loop, the Pareto set model is updated to further improve these already well-performing preferences. This seems inconsistent with the explanation provided in the paper.
2. Some explanations regarding the risk control mechanism could be further clarified. In particular, it would be helpful if the paper could provide more intuition on why sampling with Langevin dynamics is expected to generate high-risk samples. In addition, the role and importance of the risk suppression factor in Eqs. (2) and (3) could be discussed in more detail. Providing brief explanations for these design choices may help readers better understand the motivation and effectiveness of the proposed risk control module.
3. In the proposed $\text{IGD}_{\text{offline}}$ metric, it appears that the objective values are not normalized. This may affect the fairness of performance comparisons, especially since the objectives in the RE tasks can have significantly different scales.

---

> ### Author Rebuttal · Authors · 2026-03-31
>
> We really appreciate Reviewer vubi for the positive comments on our paper. We address the raised concerns below.
>
> **[W1] Misalignment between Eq. (2) and the claimed exploration behavior.**
>
> Thank you for the insightful comment, and we apologize for the confusion caused by our expressions.
>
> We clarify that the inner-loop preference optimization is designed to **adaptively explore the preference space** to improve the **diversity of generated solutions**, rather than simply focusing on already well-performing regions.
>
> Although Eq. (2) minimizes the scalarization function, it is evaluated on the solution generated by the current Pareto set model, i.e., $x = h_\phi(\lambda)$. Therefore, the update reflects **how well the model performs under each preference**: if a preference leads to unpromising solutions (i.e., the model performs poorly under that preference), it will produce larger gradients and thus be updated more significantly. In this way, **the optimization implicitly shifts toward poorly learned regions**.
>
> The outer loop then optimizes the Pareto set model to improve the **quality of solutions** under these preferences.
>
> Together, this nested design balances diversity (via preference exploration) and quality (via model optimization). We will revise the paper to clarify this point and avoid misunderstanding.
>
>
> **[W2] Clarification of the risk control mechanism.**
>
> Thank you for the helpful suggestion. We agree that providing more intuition on the risk control mechanism would improve clarity.
>
> **(1) Why Langevin dynamics generates high-risk samples**
> The energy model assigns higher energy values to out-of-distribution (OOD) or unreliable regions. Langevin dynamics performs gradient-based sampling guided by the energy landscape, which tends to move samples toward regions with higher energy. Therefore, starting from in-distribution samples, Langevin dynamics can progressively generate samples that lie in higher-risk (i.e., more OOD) regions. **These samples serve as an effective approximation of the high-risk distribution**.
>
> **(2) Role of the risk suppression factor**
> The risk suppression factor acts as a **weighting term** in Eqs. (2) and (3), controlling the optimization updates based on the estimated risk of the generated solutions. Specifically, solutions with **higher estimated risk receive smaller effective updates**, thereby **reducing the influence of unreliable (OOD) regions** during both preference optimization and Pareto set model training. In contrast, low-risk (in-distribution) solutions are emphasized, leading to more stable and reliable optimization.
>
> Overall, this mechanism enables DOMOO to **balance exploration and reliability** by discouraging the optimizer from over-exploiting regions where the surrogate predictions are likely to be inaccurate.
> We will include additional intuitive explanations in the final version to clarify these design choices.
>
>
> **[W3] Potential lack of normalization in $\text{IGD}_\text{offline}$.**
>
> Thank you for your concern. Following the setup in OfflineMOO, we have **already applied normalization** when computing both the HV and $\text{IGD}_\text{offline}$ indicator. Therefore, the issue of inconsistent scales across different objectives does not arise, and the comparisons remain fair.

---

> > ### Author Rebuttal · Reviewer_vubi · 2026-04-02
> >
> > Thank you for your rebuttal. However, one of my previous question was not addressed. I would like to restate it more clearly:
> >
> > As mentioned in the paper, it is necessary to select an optimal subset from the learned Pareto solution set. In the proposed method, IGD and HV metrics are used for subset selection, but no explicit risk control mechanism is incorporated. In my view, handling out-of-distribution (OOD) issues is also important at the final subset selection stage. Could the authors clarify why risk control is not considered during this stage, and whether this omission may affect the reliability of the selected solutions?

---

> > > ### Author Response · Authors · 2026-04-05
> > >
> > > Thank you for your follow-up question. DOMOO does not completely ignore the OOD issue at the final stage; rather, it addresses risk control earlier during the Pareto set learning stage. As a result, the candidate solutions that enter the final pool have already been filtered through risk-aware optimization and are therefore risk-suppressed to a certain extent.
> > >
> > > The main role of the final data selection stage is to balance diversity and convergence. If we were to apply an additional round of explicit risk-based filtering at this stage, the resulting candidate set could become overly conservative, potentially harming the coverage of the Pareto front.
> > >
> > > We also experimented with applying the energy model to suppress risk when generating solutions from the surrogate model. The results are shown in the table below. Empirically, this additional risk filtering did not lead to a consistent or stable performance improvement.
> > >
> > > | Task  | $HV$          | $IGD\_{off}$     |
> > > |-------|-------------|--------------|
> > > | dtlz4 | 9.44 ± 0.28 | 0.60 ± 0.04  |
> > > | re25  | 4.84 ± 0.00 | 0.07 ± 0.00  |
> > > | zdt3  | 5.61 ± 0.15 | 0.35 ± 0.04  |
> > >
> > > Therefore, our current design choice is to incorporate risk control primarily into the learning stage, while reserving the final selection stage for diversity–convergence trade-offs. We agree that this point was not sufficiently clarified in the paper, and we will make it more explicit in the revision. Thanks.

---

### Official Review · Reviewer_Lx2b · 2026-03-11

**Soundness:** 2
**Presentation:** 3
**Significance:** 2
**Originality:** 2
**Overall Recommendation:** 3
**Confidence:** 4

**Summary:**

This paper proposes DOMOO, a diversity-driven offline multi-objective optimization approach, for finding a diverse set of solutions to offline multi-objective optimization problems. The key contribution is a nested Pareto set learning framework, which comprises: 1) an accumulative risk control module, 2) a nested Pareto set learning approach, and 3) a diversity-driven solution selection strategy incorporating a novel IGD_offline indicator. Experimental results demonstrate that the proposed DOMOO approach achieves promising performance on several offline multi-objective optimization problems.

**Compliance With Llm Reviewing Policy:**

Affirmed.

**Final Justification:**

Some of my concerns have been properly addressed, and I appreciate the efforts the authors have made in the rebuttal. However, I still think the technical novelty over the previous work (Lu et al., 2023) is somewhat incremental, and the practical advantages of the proposed method are not clearly supported by the current experiments. Therefore, I raise my overall recommendation to weak reject, but I cannot go beyond that.

A remaining concern that does not affect my recommendation: I can understand that a half-half mix of HV and IGD_offline might lead to a solution set with a "better" quality-diversity trade-off due to the complementary strengths mentioned by the authors. However, it is still not convincing that such a half-half mix can consistently outperform both the strategy of directly maximizing HV and the strategy of directly maximizing IGD_offline. It should be noted that a solution set with the best quality-diversity trade-off (if such a thing even exists) does not necessarily yield the best HV and IGD_offline metrics.

**Key Questions For Authors:**

Please address the concerns raised in the weaknesses above.

**Limitations:**

Yes. This work has discussed the limitations of the proposed method in Section 6.

**Strengths And Weaknesses:**

**Strengths**

+ This paper is clearly written and easy to follow.

+ Offline multi-objective optimization is important for many real-world applications, yet remains relatively underexplored. This work makes a timely and valuable contribution to this important research area.

+ The proposed method can outperform other existing approaches on some offline multi-objective optimization problems.

**Weaknesses**

- **Accumulative Risk Control Module**

For the accumulative risk control module, the main component is directly adopted from existing work (Lu et al., 2023), thereby limiting its contribution.

In addition, during the training of the Pareto set model, the risk suppression factor is directly added to control the gradient of augmented Tchebycheff scalarization for updating both the preference and the model, without further justification or analysis. No theoretical analysis is provided to demonstrate that adding this factor can mitigate the out-of-distribution risk for Pareto set learning.

- **Preference Gradient Update**

Regarding the preference update, this work states that "By adaptively updating preferences in this way, the model is guided to focus on regions where its performance is lacking, thus making the training process more effective." However, no analysis is provided to show that the training process is accelerated. Moreover, it remains unclear whether the updated biased preference might actually hurt the model's overall performance.

- **Solution Selection**

This paper states that the proposed solution selection strategy (and the IGD_offline indicator) is intended to "avoid the bias of hypervolume indicator". However, this bias is not formally defined and analyzed in the work. In Figure 2, it is unclear why the hypervolume indicator fails to select the solutions in the middle of the Pareto front.

The paper also mentions that this approach is designed "to address the diversity challenge posed by the HV indicator in offline settings, which is demonstrated in Appendix I". However, Appendix I contains only a brief analysis of performance on a problem with a large number of almost weakly Pareto optimal solutions. It is well known that hypervolume-based methods may perform poorly in this specific case, as such solutions tend to have small hypervolume contributions. Therefore, it is not surprising that an IGD-based method achieves better performance in this scenario. This case study is not directly related to the offline setting and should not be used to represent "the diversity challenge posed by the HV indicator in offline settings".

A formal analysis is needed to better support the motivation for the proposed solution selection strategy.

- **Mixed IGD_offline and HV based Selection**

The proposed solution selection strategy first uses the IGD_offline indicator to select half of the candidate solutions and then selects the remaining solutions using the HV indicator. This approach appears largely heuristic and lacks theoretical support.

If hypervolume or IGD_offline is the actual performance metric of interest (e.g., in Tables 1 and 2), it is unclear why that same indicator is not used to select all solutions. It is difficult to understand how a half-half mix of HV and IGD_offline could consistently yield the best performance for both metrics.

- **Visualization of the Learned Pareto Front**

In Figure 3, did the "PSL after exploration" and "PSL after Nested PSL" results have the same computational budget? Alternatively, were the nested PSL results obtained by building upon the exploration phase? If so, please also report the "PSL after exploration" results under a computational budget equal to that of the nested PSL approach.

- **Experiment**

According to Tables 1 and 2, the proposed DOMOO method achieves the best performance only on synthetic and RE problems in terms of the HV rank, and only on synthetic problems in terms of the proposed IGD_offline indicator. On the more realistic MO-NAS, MORL, and Sci-Design problems, it is outperformed by some simpler baseline methods. These experimental results are insufficient to demonstrate a clear advantage of DOMOO over existing methods.

- **Ablation Study**

In Table 3, for the Regex problem, the w.o. SMG achieves better HV performance than DOMOO, but this result is not properly highlighted. In addition, it is worth examining why the version without SMG can outperform DOMOO in both HV and IGD_offline. This raises the question of whether the surrogate model generation step might degrade overall performance on certain problems.

In addition, according to Table 1 and Table 2, DOMOO primarily demonstrates stronger overall performance on synthetic and RE problems. However, in Table 3, the improvements of DOMOO over several simpler ablation baselines appear marginal for DTLZ3 and even sometimes trivial for RE24.

- **Citation for Pareto Solution Definition**

The citation provided for the definitions of Pareto-optimal solutions, the Pareto set, and the Pareto front is not appropriate.

---

> ### Author Rebuttal · Authors · 2026-03-31
>
> Thank you for providing valuable comments.
>
> **[W1] Limitation about the risk control.**
>
> Our key contribution lies in establishing a **unified theoretical perspective on offline risk** that consistently characterizes both single-objective and multi-objective settings under distribution shift. Building on this perspective, we extend accumulative risk control from its original single-objective formulation to the multi-objective regime, and provide corresponding analysis and empirical validation. These additions clarify how risk interacts with Pareto optimization, particularly in shaping solution diversity and preventing OOD-induced bias.
>
> Under this unified framework, we further design DOMOO as a **risk-aware offline MOO framework**, where risk is systematically integrated into both preference updating and Pareto set learning, enabling a balance between diversity and reliability.
>
> **[W2] Theory of risk suppression factor.**
>
> Although MOO involves multiple objectives, optimization is performed via scalarization. Thus, the OOD risk can be analyzed through the scalarized objective. Let $Q_T$ denote the distribution after $T$ updates. The scalarization error satisfies
> $$
> \mathbb E_{x\sim Q_T}[|\hat g-g|]
> \le
> (1+\rho)\sum_i \lambda_i \mathbb E_{x\sim Q_T}[|\hat f_i-f_i|],
> $$
> showing that it decomposes into weighted single-objective errors.
>
> Following ARCOO, each surrogate error is bounded by the in-distribution error plus a term proportional to $\mathrm{TV}(Q_T,P)$. The key difference between single- and multi-objective settings lies only in the weights, while the dependence on distribution shift remains the same, enabling a unified treatment.
>
> With risk control,
> $$
> \mathrm{TV}(Q_T,P)\le L_P \eta G \sum_{t}R(x_t),
> $$
> while without it the bound is $L_P\eta GT$. Since $\sum_t R(x_t)<T$, this leads to a strictly tighter bound on the scalarization error.
>
> This explains why incorporating the risk suppression factor into scalarization effectively mitigates OOD risk in PSL.
>
> **[W3] Preference Gradient Update.**
>
> **We do not claim theoretical acceleration**; “more effective” refers to better allocation of updates. Adaptive preference updates bias training toward underperforming regions, improving coverage of the Pareto front rather than repeatedly optimizing well-covered areas.
>
> Regarding potential bias, preferences are continuously resampled from a Dirichlet distribution, ensuring global coverage, and combined with risk control to avoid overfitting to unreliable regions. In practice, this transient bias does not harm overall performance.
>
> **[W4] Concern about solution selection.**
>
> By “bias of the HV indicator,” we refer to its tendency to favor extreme solutions with larger marginal contributions, while under-representing central regions of the Pareto front. This explains the phenomenon observed in Fig. 2.
>
> We agree that Appendix I alone is insufficient, and our intention is to highlight a issue: in offline settings with limited and uneven candidates, HV-based selection can further amplify imbalance by prioritizing extreme points.
>
> In contrast, IGD_offline encourages uniform coverage via distance to reference points, making it more suitable for such settings. We will clarify this in the revision.
>
> **[W5] Concern about Mixed IGD_offline and HV based Selection.**
>
> IGD_offline promotes coverage while HV emphasizes solution quality, and their combination balances diversity and extremity in Sec. 4.3.
>
> The 1/2–1/2 split is not ad hoc but selected via hyperparameter analysis. By varying $K$ from 0 to 256 (Tables 33–34), we find that $K=128$ consistently performs best, with stable performance across a reasonable range.
>
> **[W6] Visualization of the Learned Pareto Front.**
>
> Figure 3 is intended as a visualization of different stages in the Nested PSL training process, rather than a controlled ablation study. In particular, “PSL after Nested PSL” is obtained by further refining the model after the exploration phase, i.e., it builds upon “PSL after exploration.” Therefore, the two results do not share the same computational budget.
>
> To analyze the effect of computational allocation, we provide additional experiments in Tables 31–32, where we vary the exploration budget ratio.
>
> **[W7] About Experiment.**
>
> While DOMOO does not achieve the best score on every task, it shows consistent and robust performance across settings. On challenging real-world tasks, simpler baselines may score higher but are more prone to exploiting surrogate errors.
>
> Our method focuses on reliability under OOD conditions rather than overfitting specific tasks, leading to more stable overall performance.
>
> **[W8] About Ablation Study.**
>
> On the Regex task, the w.o. SMG variant performs slightly better because the task is relatively simple, where NPSL alone is sufficient and SMG may introduce minor noise.
>
> However, SMG is essential for improving and stabilizing performance on most tasks in Table 3.
>
> **[W9] About typos.**
>
> We will fix it.

---

> > ### Author Rebuttal · Reviewer_Lx2b · 2026-04-04
> >
> > Thank you very much for your response and the new experimental results. Some of my concerns have been properly addressed, but several of them remain:
> >
> > **[W1/W2]** It seems that the main theoretical results are obtained by directly applying the results from ARCOO (Lu et al., 2023) to the scalarized single-objective optimization problem, which makes the theoretical contribution less promising.
> >
> > **[W4]** It would be better to provide a clear and convincing example to demonstrate the bias of hypervolume and its diversity challenge in the offline setting.
> >
> > **[W5]** For solution selection, it is still hard to understand how a half-half mix of HV and IGD_offline could consistently outperform both the strategy of directly maximizing HV and the strategy of directly maximizing IGD_offline.
> >
> > **[W7]** Can you provide more explanation on "on challenging real-world tasks, simpler baselines may score higher but are more prone to exploiting surrogate errors", and its drawback for real-world applications?
> >
> > In addition, it seems that on more realistic problems, some simpler baselines (e.g., MultipleModels+COMs / MultipleModels+IOM) can significantly outperform DOMOO on MORL/Sci-Design, and achieve similar performance on MO-NAS. Since DOMOO mainly achieves promising performance on synthetic problems, its practical advantage is questionable.

---

> > > ### Author Response · Authors · 2026-04-05
> > >
> > > Thank you for your continued engagement and constructive feedback.
> > >
> > > **[W1/W2]** We agree that the scalarization step allows borrowing error bounds from single-objective analysis. However, directly applying ARCOO to offline MOO is non-trivial because:
> > > Preference-Risk Coupling: In PSL, the optimization landscape is conditioned on preference vectors. OOD risk does not act uniformly; it varies across objectives and interacts with preference gradients. Simply bounding scalarized error without modeling how risk shifts preference dynamics would not prevent Pareto front distortion.
> > > Multi-Objective Dominance Structure: Surrogate overestimation in one objective can artificially dominate others, collapsing diversity. Our theoretical extension shows how the risk suppression factor modifies the preference gradient update (Eq. 2), effectively damping steps toward unreliable regions while preserving gradient flow toward well-supported trade-offs.
> > >
> > > **[W4]** The core issue arises from surrogate-induced spurious Pareto fronts in OOD regions: due to limited offline data, surrogates often extrapolate a much wider front than the true one. Candidates far beyond the true front typically correspond to poorly calibrated regions—i.e., they have low true quality and tend to cluster tightly in objective space.
> > > When using HV for selection, its marginal-volume mechanism tends to uniformly pick solutions along this wide spurious front. Consequently, HV frequently selects many tightly clustered, low-quality solutions in reality, harming true diversity despite acceptable surrogate-based scores.
> > > In contrast, $IGD_{\text{offline}}$ uses the offline Pareto front as reference and computes average distances to it. This inherently penalizes solutions deviating from reliable data regions, acting as a conservative filter against OOD artifacts. As a result, $IGD_{\text{offline}}$ preserves well-distributed, trustworthy solutions—directly addressing the offline diversity challenge where no active queries can correct surrogate errors.
> > > You can found the schematic in https://anonymous.4open.science/r/DOMOO-0388/rebuttal_table/rebuttal_Lx2b.png.
> > >
> > > **[W5]** The hybrid selection strategy is designed to exploit the **complementary strengths** of the two indicators:
> > >
> > > | Indicator | Strength | Limitation in Offline MOO |
> > > |-----------|----------|---------------------------|
> > > | **$IGD_{\text{offline}}$** | Promotes uniform coverage via distance to reference points | May retain diverse but suboptimal solutions |
> > > | **HV** | Emphasizes convergence and solution quality | Favors extreme solutions, under-represents central trade-offs |
> > >
> > > **Why the 1/2–1/2 split works:**
> > > 1. **Stage 1 ($IGD_{\text{offline}}$)**: Selects up to 128 solutions to establish a diversity-preserving "skeleton" along the offline PF, preventing clustering in OOD regions.
> > > 2. **Stage 2 (HV)**: Fills the remaining slots by maximizing hypervolume, refining the skeleton toward optimality without sacrificing coverage.
> > >
> > > **Empirical validation**: The 128/256 budget is not heuristic but selected via hyperparameter analysis (Tables 33–34). Varying the $IGD_{\text{offline}}$ budget from 0 to 256 shows stable performance with a clear peak around 128, confirming this split balances exploration and exploitation.
> > >
> > > **[W7]** Methods like MultipleModels+COMs or End-to-End+NSGA-II aggressively optimize surrogate predictions without explicit risk awareness. In offline settings, surrogates are prone to overestimation in OOD regions. These baselines push solutions into those regions, achieving artificially high HV scores on the surrogate but poor true performance.
> > >
> > > Drawbacks for real-world applications: In domains like molecular design (Sci-Design) or robotics control (MORL), deploying a solution with falsely predicted high efficacy or stability can lead to costly experimental failures, safety risks, or wasted computational budgets. Offline MOO is explicitly used when true evaluations are expensive or hazardous; thus, reliability and deployability often outweigh nominal surrogate scores.
> > >
> > > **[Q1]**
> > >
> > > We clarify three points:
> > >
> > > 1. **Different goals**: DOMOO prioritizes *reliability under OOD conditions* over maximizing surrogate scores. Baselines like `MultipleModels+COMs/IOM` may achieve higher HV by exploiting overestimated OOD regions; DOMOO's risk control suppresses such extrapolations, trading slight surrogate HV for higher *true* deployability.
> > >
> > > 2. **Task heterogeneity**: DOMOO's gradient-based PSL suits continuous/smooth spaces better. On highly discrete tasks (e.g., certain Regex/NAS), all neural surrogate methods face challenges—we acknowledge this in Sec. 6 and plan hybrid architectures for future work.
> > >
> > > 3. **Holistic evaluation**: DOMOO achieves the *best average rank* across benchmarks (Table 1) and consistently ranks top in IGD$_{\text{offline}}$ (Table 2), indicating superior balanced performance. In offline MOO, robustness to surrogate bias often outweighs peak surrogate scores.

---

### Official Review · Reviewer_gHjy · 2026-03-11

**Soundness:** 3
**Presentation:** 2
**Significance:** 2
**Originality:** 2
**Overall Recommendation:** 4
**Confidence:** 3

**Summary:**

This paper consider a offline MOO setting where the objectives formulation is unknown but only some offline dataset ((xi, f(xi))) is known.
The method firstly learn a surrogate objective $\hat f(x_i)$ from the offline dataset, using whiche, the build a surrogate augmented Tchebycheff scalization objective, with which they can find the corresponding Pareto-set model for each weight lambda. The Pareto optimal solution’s relation to lambda is modeled by $x=h_\phi(\lambda)$. In this way, for any lambda, they can get the corresponding Pareto solution estimate through model $ h_\phi $. To obtain reliable phi, they start with a warm up using a set of lambda as data-driven weight gives larger weight to objectives with smaller gap (closer to ideal) at that offline PF point and learn phi by minimizing the surrogate scalization objective. Then, they sample many lambda following Dirichlet distribution and keep refining the surrogate objective. Furthermore, to prevent OOD estimation error from hat f, energy model $E_\omega$ is used to construct risk level R(x). In this way, preference refinement is employed through gradient update on lambda to seek improvable trade-offs under the current model and surrogate, where R(x) comes in as “stepsize tuner”. They output a set of obtained solutions that are nearly pareto optimal that are most representative under minimal IGD metric. They evaluate DOMOO on Off-MOO-Bench (Synthetic, MO-NAS, MORL, Sci-Design, RE) using HV and IGDoffline rankings, and report that DOMOO attains the best overall average rank across tasks while being weaker on some highly discrete one-hot settings.

**Compliance With Llm Reviewing Policy:**

Affirmed.

**Key Questions For Authors:**

(Minor) z, ideal vector, was not clearly introduced. I suppose there is zi = min_x fi(x). Please clarify.

What is Dirichlet(a) explicitly? Why chosen over Dirichlet distribution?

How often do improvements under f ̂translate to improvements under the true evaluator (when available), especially for OOD candidates? Can you report the correlation or rank-consistency between f ̂(x)and f(x)on generated samples?

What is the failure mode when E_ωis miscalibrated? Do you have ablations showing sensitivity to energy-model capacity, training procedure, and the choice of P ̃,Q ̃/ Langevin steps?

The paper notes weaker performance on one-hot/high-dimensional discrete tasks. Can you clarify whether the limitation comes from h_ϕparameterization, gradients through g ̂^"tch_aug" , or the risk model, and what modification would address it?

**Limitations:**

yes

**Strengths And Weaknesses:**

The paper proposes a coherent offline-MOO pipeline that combines a preference-conditioned Pareto-set model h with risk-weighted updates via R(x) and it explicitly addresses the practical “HV collapses to extremes” issue by adding an IGD-driven diversity selection stage.  The key ingredients (risk R(x) preference updates of λ, and the shifted IGD-reference) are largely heuristic and surrogate-dependent, so it is unclear when they truly reflect improvement under the unknown frather than exploiting f ̂.

---

> ### Author Rebuttal · Authors · 2026-03-31
>
> We really appreciate Reviewer gHjy for the positive comments on our paper. We address the concerns below.
>
> **[W1] Definition of the ideal vector z.**
>
> We will clarify this point in the revision by explicitly defining the ideal vector $ z^* $ in Section 3.2. In particular, each component will be defined as $z_i^* = \min_{x \in \mathcal{X}} f_i(x)$.
>
> **[W2] Definition of the Dirichlet($\alpha$).**
>
> We need to model a distribution over preference vectors $\lambda$ in DOMOO, where each $\lambda$ represents a trade-off among objectives and must satisfy $\lambda_i \ge 0$ and $\sum_i \lambda_i = 1$. Dirichlet($\alpha$) is a standard choice for this, since it is a distribution defined exactly on this simplex.
>
> We use Dirichlet not just for convenience, but because it models a distribution over preference vectors rather than just performing random sampling. By sampling $\lambda \sim \text{Dirichlet}(\alpha)$, we can generate diverse trade-offs (e.g., focusing on one objective or balancing all of them), so the model does not overfit to a small set of preferences. This is important in our setting, since we want the learned Pareto set to work well across all possible preferences.
>
> This kind of stochastic sampling of preference vectors via Dirichlet(α) is a well-validated approach in Pareto front learning and multi-objective optimization [1].
>
> **[W3] Concern about the gap between the true evaluator and the surrogate model.**
>
> We analyze the consistency between the surrogate $\hat{f}(x)$ and the true evaluator $f(x)$ on ZDT3, focusing on OOD regions.
>
> We treat offline data as ID, generate OOD samples via Langevin dynamics, and evaluate DOMOO-selected solutions. Pearson correlations show that the surrogate is highly reliable on ID data (0.9999, 0.9892), but degrades in OOD, especially for objective 2 (0.3961). In contrast, DOMOO-selected solutions improve this correlation (0.4300) while maintaining strong consistency on objective 1 (0.9985).
>
> These results indicate that DOMOO mitigates surrogate misalignment in OOD regions, leading to more reliable optimization.
>
> **[W4] Failure modes of miscalibrated $E_\omega$ and sensitivity ablations.**
>
> **Failure mode.** When the energy model $E_\omega$ is miscalibrated, its estimated density may no longer reflect the true data distribution. In such cases, two failure modes can arise: (i) *under-penalization*, where OOD regions are assigned overly low energy, causing the optimizer to exploit unreliable surrogate predictions (i.e., spurious optima), and (ii) *over-penalization*, where valid high-quality regions are incorrectly suppressed, leading to overly conservative solutions. Both cases harm the alignment between surrogate optimization and true objective improvement. **All the additional experiments are shown `in this link`**([https://anonymous.4open.science/r/DOMOO-0388/rebuttal_table/To_Reviewer_gHjy.md](https://anonymous.4open.science/r/DOMOO-0388/rebuttal_table/To_Reviewer_gHjy.md)).
>
> **Ablation on energy model design.** We conduct sensitivity analyses to evaluate robustness:
>
> - **Model capacity (hidden size).** Varying the hidden size from 128 to 1024 results in only minor performance changes across tasks (e.g., HV on dtlz4 varies within ~0.2), indicating low sensitivity to model capacity.
>
> - **Training procedure (epochs).** Increasing training epochs from 0 to 100 yields stable performance without noticeable degradation or overfitting, suggesting robustness to training schedules.
>
> - **Langevin steps / sampling design ($\tilde{Q}$).** Adjusting the number of Langevin steps (0.5–2.0×) also leads to consistently stable results, showing insensitivity to negative sample generation.
>
> - **Selection of low-risk samples ($\tilde{Q}$).** Varying the selection strategy for low-risk samples from the offline dataset similarly produces stable performance.
>
> These results show that while $E_\omega$ helps identify unreliable regions, our method does not depend on precise calibration. Instead, it only requires a coarse distinction between ID and OOD regions, making it robust in practice.
>
> **[W5] Concerns about performance on one-hot and high-dimensional discrete tasks.**
>
> We attribute this limitation to the mismatch between continuous optimization and one-hot, high-dimensional discrete spaces. Valid data lie only on sparse vertices, while intermediate regions are largely OOD. The Pareto set model may generate such points, where the surrogate becomes unreliable and yields spurious optima. Although the risk model mitigates this issue, its effectiveness is limited due to the vast OOD space.
>
> A potential solution is to adopt a more compact latent representation (e.g., VAE) instead of one-hot encoding, to better align the search space with the data distribution. We will also explore alternative approaches in future work to further address this issue.
>
> [1] Navon A, Shamsian A, Chechik G, et al. Learning the pareto front with hypernetworks[J]. arXiv preprint arXiv:2010.04104, 2020.

---

> > ### Author Rebuttal · Reviewer_gHjy · 2026-04-05
> >
> > The authors addressed my concerns.

---

> > > ### Author Response · Authors · 2026-04-07
> > >
> > > Thank you very much for taking the time to review our paper and read our rebuttal. We are glad that our responses have adequately addressed your concerns. Your constructive feedback during the review process has been highly valuable in improving the quality of our work. Thank you again for your recognition and support!

---

### Decision · Program_Chairs · 2026-04-30

**Decision:**

Accept (regular)

**Comment:**

The paper need substantially strengthen the discussion of theoretical distinctiveness from ARCOO and scalarized approaches, provide clear empirical evidence (or candid discussion) regarding performance gaps on real-world tasks versus synthetic benchmarks, and include the requested illuminating examples of HV bias in offline contexts. The camera-ready version should explicitly address the limitations regarding generalization to complex real-world scenarios highlighted by the reviewer.